# RL4CO: A Unified Reinforcement Learning for Combinatorial Optimization Library

## ABSTRACT

Deep reinforcement learning offers notable benefits in addressing combinatorial problems over traditional solvers, reducing the reliance on domain-specific knowledge and expert solutions, and improving computational efficiency. Despite the recent surge in interest in neural combinatorial optimization, practitioners often do not have access to a standardized code base. Moreover, different algorithms are frequently based on fragmentized implementations that hinder reproducibility and fair comparison. To address these challenges, we introduce RL4CO, a unified Reinforcement Learning (RL) for Combinatorial Optimization (CO) library. We employ state-of-the-art software and best practices in implementation, such as modularity and configuration management, to be flexible, easily modifiable, and extensible by researchers. Thanks to our unified codebase, we benchmark baseline RL solvers with different evaluation schemes on zero-shot performance, generalization, and adaptability on diverse tasks. Notably, we find that some recent methods may fall behind their predecessors depending on the evaluation settings. We hope RL4CO will encourage the exploration of novel solutions to complex real-world tasks, allowing the community to compare with existing methods through a unified framework that decouples the science from software engineering. We open-source our library at https://anonymous.4open.science/r/rl4co-iclr.

## 1 INTRODUCTION

Combinatorial optimization (CO) is a mathematical optimization area that encompasses a wide variety of important practical problems, such as routing problems, scheduling, and hardware design, whose solution space typically grows exponentially to the size of the problem (also often referred to as NP-hardness). As a result, CO problems can take considerable expertise to craft solvers and raw computational power to solve. Neural Combinatorial Optimization (NCO) (Bengio et al., 2021; Mazyavkina et al., 2021; Peng et al., 2021) provides breakthroughs in CO by leveraging recent advances in deep learning, especially by automating the design of solvers and considerably improving the efficiency in providing solutions. While conventional operations research (OR) approaches (Helsgaun, 2017; Vidal, 2022; David Applegate & Cook, 2023) have achieved significant progress in CO, they encounter limitations when addressing new CO tasks, as they necessitate extensive expertise. In contrast, NCO trained with reinforcement learning (RL) overcomes the limitations of OR-based approaches (i.e., manual designs) by harnessing RL's ability to learn in the absence of optimal solutions (Bello et al., 2017).[1]

NCO presents possibilities as a general problem-solving approach in CO, handling challenging problems with minimal or (almost) no dependence on problem-specific knowledge (Bello et al., 2017; Kool et al., 2019; Kwon et al., 2020; Hottung & Tierney, 2019; Barrett et al., 2020; Ahn et al., 2020a; Barrett et al., 2022). Among CO tasks, routing problems such as the Traveling Salesman Problem (TSP) and Capacitated Vehicle Routing Problem (CVRP) serve as central test suites for the capabilities of NCO due to the extensive NCO research on those types of problems (Nazari et al., 2018; Kool et al., 2019; Kwon et al., 2020; Kim et al., 2022b) and their applicability of at-hand comparison of highly dedicated heuristic solvers investigated over several decades of study by the OR

---

[1] Supervised learning (SL) approaches also offer notable improvements (Vinyals et al., 2015b; Lu et al., 2019; Hottung et al., 2020; Fu et al., 2020; Kool et al., 2021; Xin et al., 2021; Li et al., 2021b; Drakulic et al., 2023; Sun & Yang, 2023). However, their use is restricted due to the requirements of (near) optimal solutions during training.

community (Helsgaun, 2017; David Applegate & Cook, 2023). Recent advances (Fu et al., 2021; Li et al., 2021b; Jin et al., 2023) of NCO achieve comparable or superior performance to state-of-the-art (SOTA) solvers on these benchmarks, implying the potential of NCO to revolutionize the laborious manual design of CO solvers (Vidal, 2022; Ropke & Pisinger, 2006).

Recently, while NCO has significantly influenced the CO domain, there remains an absence of unified and flexible implementations of NCO solvers. For instance, two cornerstone NCO models, AM (Kool et al., 2019) and POMO (Kwon et al., 2020), rely on significantly different codebases that can hinder reproducibility and peer validation in the NCO domain. We believe a unified platform is crucial to advance the NCO field for both researchers and practitioners, by decoupling science from engineering and ensuring the transition from research ideas to practical applications. Furthermore, a consistent evaluation benchmark is needed; despite the usual assessment of NCO solvers based on in-training distributions, the necessity for out-of-training-distribution evaluations, reflecting real-world scenarios, is often overshadowed. Finally, there is also a need for evaluations that account for the volume of training data, as its importance has been substantiated in other ML sectors.

**Contributions.** In this work, we introduce RL4CO, a new reinforcement learning (RL) for combinatorial optimization (CO) benchmark. RL4CO is first and foremost a library of baselines, environments, and boilerplate from the literature implemented in a *modular*, *flexible*, and *unified* way with what we found are the best software practices and libraries, including TorchRL (Moens, 2023b), PyTorch Lightning (Falcon & The PyTorch Lightning team, 2019), Hydra (Yadan, 2019) and Tensor-Dict (Moens, 2023a). Through our thoroughly tested unified and documented codebase, we conduct experiments to explore best practices in RL for CO and benchmark our baselines. We demonstrate that existing SOTA methods may perform poorly on different evaluation metrics and sometimes even underperform their predecessors. We also analyzed the inference schemes of NCO solvers and found that simple inference data augmentation can outperform more computationally demanding schemes, such as sampling, thanks to our unified implementation. We showcase the flexibility of RL4CO by also benchmarking lesser-studied problems in the NCO community that can be used to model real-world problems, such as non-euclidean hardware device placement (Kim et al., 2023) and constrained pickup and delivery problem (Li et al., 2021a).

## 2 PRELIMINARIES

The solution space of CO problems generally grows exponentially to their size. Such solution space of CO hinders the learning of NCO solvers that generate the solution in a single shot[2]. As a way to mitigate such difficulties, *autoregressive* (AR) methods as Nazari et al. (2018); Vinyals et al. (2015a); Kool et al. (2019); Kwon et al. (2020); Kim et al. (2022b) generate solutions one step at a time in an autoregressive fashion akin to language models (Brown et al., 2020; Touvron et al., 2023; OpenAI, 2023) to handle solution feasibility and constraints; in RL4CO we focus primarily on benchmarking AR approaches for given their broad applicability.

**Solving Combinatorial Optimization with Autoregressive Sequence Generation** AR methods construct a feasible solution as follows. They first *encodes* the problem $\boldsymbol{x}$ (e.g., For TSP, node coordinates) with trainable encoder $f_\theta$ and then *decodes* the solution $\boldsymbol{a}$ by deciding the next "action" (e.g., the next city to visit) based on the current (partial) solution using decoder $g_\theta$, and repeating this until the solver generates the complete solution. Formally speaking,

$$\boldsymbol{h} = f_\theta(\boldsymbol{x}) \tag{1}$$

$$a_t \sim g_\theta(a_t | a_{t-1}, ..., a_0, \boldsymbol{h}) \tag{2}$$

$$p_\theta(\boldsymbol{a}|\boldsymbol{x}) \triangleq \prod_{t=1}^{T-1} g_\theta(a_t | a_{t-1}, ..., a_0, \boldsymbol{h}), \tag{3}$$

where $\boldsymbol{a} = (a_1, ..., a_T)$, $T$ are the solution construction steps, is a feasible (and potentially optimal) solution to CO problems, $\boldsymbol{x}$ is the problem description of CO, $p_\theta$ is a (stochastic) solver that maps

---

[2] Also known as non-autoregressive approaches (NAR) (Joshi et al., 2021; Kool et al., 2021; Qiu et al., 2022; Gagrani et al., 2022; Xiao et al., 2023). However, imposing the feasibility of NAR-generated solutions may not be straightforward, especially for CO problems with complex constraints.

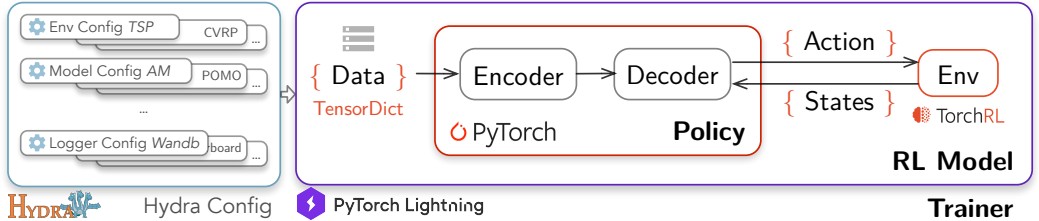

Figure 3.1: An overview of the RL4CO pipeline from configurations to training.

$x$ to a solution $a$. For example, for a 2D TSP with $N$ cities, $x = \{(x_i, y_i)\}_{i=1}^N$, where $(x_i, y_i)$ is the coordinates of $i$th city $v_i$, a solution $a = (v_1, v_2, ...v_N)$.

**Training NCO Solvers via Reinforcement Learning** The solver $p_\theta$ can be trained with SL or RL schemes. In this work, we focus on RL solvers as they can be trained without relying on the optimal (or high-quality) solutions. Under the RL formalism, the training problem of NCOs becomes as follows:

$$\theta^* = \underset{\theta}{\arg\max}\Big[\mathbb{E}_{x \sim P(x)}\big[\mathbb{E}_{a \sim p_\theta(a|x)}R(a, x)\big]\Big], \tag{4}$$

where $P(x)$ is problem distribution, $R(a, x)$ is reward (i.e., the negative cost) of $a$ given $x$.

Eq. (4) can be solved with various RL methods including value-based (Khalil et al., 2017), policy gradient (PG) (Kool et al., 2019; Kwon et al., 2020; Kim et al., 2022b; Park et al., 2023), and actor-critic (AC) methods (Zhang et al., 2020; Park et al., 2021). In practice, we observe that explicitly training the policy (i.e., PG and AC methods) generally outperforms the value-based methods in NCO, as shown in Kool et al. (2019) and our experimental results (See § 4).

## 3 RL4CO

RL4CO is a unified reinforcement learning (RL) for Combinatorial Optimization (CO) library that aims to provide a *modular*, *flexible*, and *unified* code base for training and evaluating AR methods and performs extensive benchmarking capabilities on various settings. As shown in Fig. 3.1, RL4CO decouples the major components of the AR-NCO solvers and their training routine while prioritizing reusability into the following components: 1) Policy, 2) Environment, 3) RL algorithm, 4) Trainer, and 5) Configuration management.

**Policy** This module is for autoregressively constructing solutions. Our investigation into various AR NCO solvers has revealed common structural patterns across various CO problems. The policy network $\pi_\theta$ (i.e., solver) follows an architecture that combines an encoder $f_\theta$ and a decoder $g_\theta$ as follows:

$$\pi_\theta(a|x) \triangleq \prod_{t=1}^{T-1} g_\theta(a_t|a_{t-1}, ..., a_0, f_\theta(x)) \tag{5}$$

Upon analyzing encoder-decoder architectures, we have identified components that hinder the encapsulation of the policy from the environment. To achieve greater modularity, RL4CO modularizes such components in the form of *embeddings*: `InitEmbedding`, `ContextEmbedding` and `DynamicEmbedding`.

The encoder's primary task is to encode input $x$ into a hidden embedding $h$. The structure of $f_\theta$ comprises two trainable modules: the `InitEmbedding` and encoder blocks. The `InitEmbedding` module typically transforms problem features into the latent space and problem-specific compared to the encoder blocks, which often involve plain multi-head attention (MHA):

$$h = f_\theta(x) \triangleq \text{EncoderBlocks}(\text{InitEmbedding}(x)) \tag{6}$$

The decoder $g_\theta$ autoregressively constructs the solution based on the encoder output $h$. Solution decoding involves iterative steps until a complete solution is constructed:

$$q_t = \text{ContextEmbedding}(h, a_{t-1:0}), \tag{7}$$

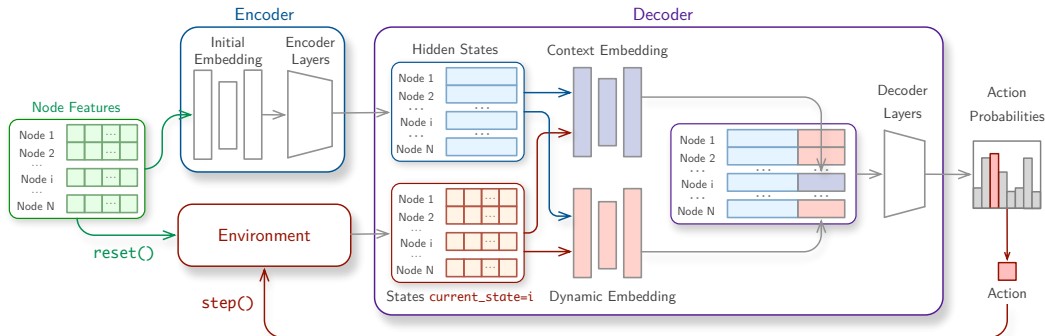

Figure 3.2: An overview of the modularized autoregressive policy in RL4CO.

$$\bar{q}_t = \text{MHA}(q_t, W_k^g \boldsymbol{h}, W_v^g \boldsymbol{h}), \tag{8}$$

$$\pi(a_t) = \text{MaskedSoftmax}(\bar{q}_t \cdot W_v \boldsymbol{h}, M_t), \tag{9}$$

where the `ContextEmbedding` is tailored to the specific problem environment, $q_t$ and $\bar{q}_t$ represent the query and attended query (also referred to as glimpse in Mnih et al. (2014)) at the $t$-th decoding step, $W_k^g$, $W_v^g$ and $W_v$ are trainable linear projections computing keys and values from $\boldsymbol{h}$, and $M_t$ denotes the action mask, which is provided by the environment to ensure solution feasibility. It is noteworthy that we also modularize the `DynamicEmbedding`, which dynamically updates the keys and values of MHA and Softmax during decoding. This approach is often used in dynamic routing settings, such as split delivery VRP. For the details, please refer to Appendix A.4.

From Eqs. (6) and (7), it is evident that creating embeddings demands problem-specific handling and often triggers coherence between the policy and CO problems. In RL4CO, we offer *plug-and-play* environment embeddings investigated from NCO literature (Kool et al., 2019; Li et al., 2021a; Kwon et al., 2021; Kim et al., 2023; Son et al., 2023) and, more importantly, allow a drop-in replacement of pre-coded embedding modules to user-defined embedding modules to attain higher modularity. Furthermore, we accommodate various decoding schemes (discussed in § 4) into a unified implementation so that those schemes can be applied to the different models, such as applying greedy multi-starts from POMO (Kwon et al., 2020) to the Attention Model (Kool et al., 2019).

**Environment**  This module fully specifies the problem, updates the problem construction steps based on the input action, and provides the result of updates (e.g., action masks) to the policy module. When implementing the `environment`, we focus on parallel execution of rollouts (i.e., problem-solving) while maintaining *statelessness* in updating every step of solution decoding. These features are essential for ensuring the reproducibility of NCO and supporting "look-back" decoding schemes such as Monte-Carlo Tree Search. Our environment designs and implementations are flexible to accommodate various types of NCO solvers that generate a single action $a_t$ at each decision-making step (Khalil et al., 2017; Zhang et al., 2020; Ahn et al., 2020b; Park et al., 2021; 2023). We provide further details of the benchmarked environments in Appendix C.

Our environment implementation is based on `TorchRL` (Bou et al., 2023), an open-source RL library for `PyTorch` (Paszke et al., 2017), which aims at high modularity and good runtime performance, especially on GPUs. This design choice makes the `Environment` implementation standalone, even outside of RL4CO, and consistently empowered by a community-supporting library – `TorchRL`. Moreover, we employ `TensorDicts` (Moens, 2023a) to move around data, allowing for further flexibility.

**RL Algorithm**  This module defines the routine that takes `Environment` and its problem instances and the `Policy` and computes the gradients of $\theta$ defined by each RL algorithm such as REINFORCE, A2C, and PPO. We decouple the routines for gradient computations and parameter updates to support modern training practices in `Trainer`. We found this decoupling especially helpful in attaining higher reusability of the RL algorithms (e.g., AM, POMO, and Sym-NCO are trained with the REINFORCE algorithm but with different baselines).

**Trainer**  Training a single NCO model is typically computationally demanding, especially since most CO problems are NP-hard. Therefore, implementing a modernized training routine becomes crucial. To this end, we implement the `Trainer` using `Lightning` (Falcon & The PyTorch Lightning team, 2019), which seamlessly supports features of modern training pipelines, including logging, checkpoint management, automatic mixed-precision training, various hardware acceleration supports (e.g., CPU, GPU, TPU, and Apple Silicon) and multi-GPU support in distributed settings (Li et al., 2020). We have found that using mixed-precision training significantly decreases training time without sacrificing NCO solver quality and enables us to leverage recent routines such as FlashAttention (Dao et al., 2022; Dao, 2023)[3].

The following code snippet shows minimalistic code that can train a model in few lines of code:

```python
from rl4co.envs import TSPEnv
from rl4co.models.zoo import AttentionModel, AutoregressivePolicy
from rl4co.utils import RL4COTrainer

# Instantiate TorchRL environment
env = TSPEnv(num_loc=50)

# Create policy and RL model
policy = AutoregressivePolicy(env)
model = AttentionModel(env, policy, baseline='rollout')

# Instantiate Trainer and fit
trainer = RL4COTrainer(max_epochs=100, accelerator="gpu")
trainer.fit(model)
```

**Configuration Management**  Optionally, but usefully, we adopt `Hydra` (Yadan, 2019), an open-source Python framework that enables hierarchical config management, making it easier to manage complex configurations and experiments with different settings.

We thoroughly test our library via continuous integration and create documentation to make it as easily accessible as possible for both newcomers and expert practitioners[4]. Installation instructions and open-source code are available at https://anonymous.4open.science/r/rl4co-iclr.

## 4  Experiments

### 4.1  Benchmark Setup

Our focus is to benchmark the NCO solvers under controlled settings, aiming to compare all benchmarked methods as closely as possible in terms of network architectures and the number of training samples consumed.

**Models**  We evaluate the following NCO solvers[5]:

- `AM` (Kool et al., 2019) employs the multi-head attention (MHA) encoder and single-head attention decoder trained using REINFORCE and the rollout baseline.
- `POMO` (Kwon et al., 2020) utilizes the shared baseline to train AM instead of the rollout baseline.
- `Sym-NCO` (Kim et al., 2022b) utilizes the symmetric baseline to train AM instead of the rollout baseline.

---

[3]We investigate mixed-precision training and FlashAttention in our pipeline in Appendix E.2.

[4]Documentation: https://anonymous.4open.science/w/rl4co-iclr/_build/html/index.html.

[5]We also re-implemented PointerNetworks (Vinyals et al., 2015a; Bello et al., 2016); however, we excluded them from the main table due to their poor performance, i.e. more than 4% optimality gap in TSP50.

Table 4.1: In-distribution benchmark results for routing problems with 50 nodes. We report the gaps to the best-known solutions of classical heuristics solvers.

| Method | TSP | | | CVRP | | | OP | | | PCTSP | | | PDP | | |
|---|---|---|---|---|---|---|---|---|---|---|---|---|---|---|---|
| | Cost ↓ | Gap | Time | Cost ↓ | Gap | Time | Prize ↑ | Gap | Time | Cost ↓ | Gap | Time | Cost ↓ | Gap | Time |
| *Classical Solvers* | | | | | | | | | | | | | | | |
| *Gurobi* | 5.70 | 0.00% | 2m | – | – | – | – | – | – | – | – | – | – | – | – |
| *Concorde* | 5.70 | 0.00% | 2m | – | – | – | – | – | – | – | – | – | – | – | – |
| *HGS* | – | – | – | 10.37 | 0.00% | 10h | – | – | – | – | – | – | – | – | – |
| *Compass* | – | – | – | – | – | – | 16.17 | 0.00% | 5m | – | – | – | – | – | – |
| *LKH3* | 5.70 | 0.00% | 5m | 10.38 | 0.10% | 12h | – | – | – | – | – | – | 6.86 | 0.00% | 1h30m |
| *OR Tools* | 5.80 | 1.83% | 5m | – | – | – | – | – | – | 4.48 | 0.00% | 5h | 7.36 | 7.29% | 2h |
| *Greedy One Shot Evaluation* | | | | | | | | | | | | | | | |
| A2C | 5.83 | 2.22% | (<1s) | 11.16 | 7.09% | (<1s) | 14.77 | 8.64% | (<1s) | 5.15 | 14.96% | (<1s) | 8.92 | 30.03% | (<1s) |
| AM | 5.78 | 1.41% | (<1s) | 10.95 | 5.30% | (<1s) | 15.46 | 4.40% | (<1s) | 4.59 | 2.46% | (<1s) | 7.51 | 9.88% | (<1s) |
| POMO | 5.75 | 0.89% | (<1s) | 10.80 | 3.99% | (<1s) | 13.86 | 14.26% | (<1s) | 5.00 | 11.61% | (<1s) | 7.59 | 10.64% | (<1s) |
| Sym-NCO | 5.72 | 0.47% | (<1s) | 10.87 | 4.61% | (<1s) | 15.67 | 3.09% | (<1s) | 4.52 | 2.12% | (<1s) | 7.39 | 7.73% | (<1s) |
| AM-XL | 5.73 | 0.54% | (<1s) | 10.84 | 4.31% | (<1s) | 15.69 | 2.98% | (<1s) | 4.53 | 2.44% | (<1s) | 7.31 | 6.56% | (<1s) |
| AM-PPO | 5.76 | 0.92% | (<1s) | 10.87 | 4.60% | (<1s) | 15.67 | 3.05% | (<1s) | 4.55 | 2.45% | (<1s) | 7.43 | 8.31% | (<1s) |
| *Sampling with width $M = 1280$* | | | | | | | | | | | | | | | |
| A2C | 5.74 | 0.72% | 40s | 10.70 | 3.07% | 1m24s | 15.14 | 6.37% | 48s | 4.96 | 10.71% | 57s | 8.48 | 23.62% | 1m15s |
| AM | 5.72 | 0.40% | 40s | 10.60 | 2.22% | 1m24s | 15.90 | 1.68% | 48s | 4.52 | 0.99% | 57s | 7.25 | 5.69% | 1m15s |
| POMO | 5.71 | 0.18% | 1m | 10.54 | 1.64% | 2m30s | 14.62 | 9.56% | 1m10s | 4.82 | 7.59% | 1m23s | 7.31 | 6.56% | 1m50s |
| Sym-NCO | 5.70 | 0.14% | 1m | 10.58 | 2.03% | 2m30s | 16.02 | 0.93% | 1m10s | 4.52 | 0.82% | 1m23s | 7.17 | 4.52% | 1m50s |
| AM-XL | 5.71 | 0.17% | 1m | 10.57 | 1.91% | 2m30s | 15.97 | 1.25% | 1m10s | 4.52 | 0.88% | 1m23s | 7.15 | 4.23% | 1m50s |
| AM-PPO | 5.70 | 0.15% | 1m | 10.52 | 1.52% | 1m30s | 16.04 | 0.78% | 50s | 4.48 | 0.18% | 1m | 7.17 | 4.52% | 1m20s |
| *Greedy Multistart (N)* | | | | | | | | | | | | | | | |
| A2C | 5.80 | 1.81% | 2s | 10.90 | 4.86% | 6s | 14.61 | 9.65% | 4s | 5.12 | 14.29% | 5s | 8.87 | 29.31% | 4s |
| AM | 5.77 | 1.21% | 2s | 10.73 | 3.39% | 6s | 15.71 | 2.84% | 4s | 4.56 | 1.89% | 5s | 7.46 | 8.75% | 4s |
| POMO | 5.71 | 0.29% | 3s | 10.58 | 2.04% | 8s | 13.95 | 13.71% | 7s | 4.98 | 11.16% | 7s | 7.46 | 8.75% | 6s |
| Sym-NCO | 5.72 | 0.36% | 3s | 10.71 | 3.17% | 8s | 15.88 | 1.79% | 7s | 4.55 | 1.59% | 7s | 7.38 | 7.58% | 6s |
| AM-XL | 5.72 | 0.42% | 3s | 10.68 | 2.88% | 8s | 15.85 | 1.95% | 7s | 4.56 | 1.79% | 7s | 7.25 | 5.69% | 6s |
| AM-PPO | 5.74 | 0.61% | 2s | 10.67 | 2.72% | 6s | 15.98 | 1.21% | 5s | 4.53 | 1.18% | 5s | 7.23 | 5.39% | 4s |
| *Greedy with Augmentation (1280)* | | | | | | | | | | | | | | | |
| A2C | 5.71 | 0.18% | 40s | 10.63 | 2.49% | 1m24s | 14.89 | 7.91% | 48s | 5.15 | 14.96% | 1m | 7.91 | 15.31% | 1m15s |
| AM | 5.70 | 0.07% | 40s | 10.53 | 1.56% | 1m24s | 15.88 | 1.79% | 48s | 4.59 | 2.46% | 1m | 7.14 | 4.08% | 1m15s |
| POMO | 5.70 | 0.06% | 1m | 10.55 | 1.72% | 2m30s | 14.23 | 11.97% | 1m15m | 5.09 | 13.61% | 1m42s | 7.15 | 4.23% | 1m45s |
| Sym-NCO | 5.70 | 0.01% | 1m | 10.53 | 1.54% | 2m30s | 15.94 | 1.41% | 1m15m | 4.58 | 2.17% | 1m42s | 7.03 | 2.48% | 1m45s |
| AM-XL | 5.70 | 0.01% | 1m | 10.52 | 1.47% | 2m30s | 15.90 | 1.66% | 1m15m | 4.59 | 2.54% | 1m42s | 6.98 | 1.75% | 1m45s |
| AM-PPO | 5.70 | 0.15% | 42m | 10.52 | 1.52% | 1m30s | 16.01 | 0.84% | 52s | 4.48 | 0.18% | 1m5s | 7.00 | 2.04% | 1m20s |
| *Greedy Multistart with Augmentation ($N \times 16$)* | | | | | | | | | | | | | | | |
| A2C | 5.72 | 0.41% | 32s | 10.67 | 2.81% | 1m | 15.22 | 5.88% | 30s | 5.06 | 12.94% | 35s | 7.88 | 14.87% | 50s |
| AM | 5.71 | 0.21% | 32s | 10.55 | 1.73% | 1m | 16.05 | 0.76% | 30s | 4.54 | 1.28% | 35s | 7.10 | 3.50% | 50s |
| POMO | 5.70 | 0.05% | 48s | 10.48 | 1.11% | 2m | 15.05 | 6.94% | 1m | 4.92 | 9.81% | 1m20s | 7.12 | 3.79% | 1m25 |
| Sym-NCO | 5.70 | 0.03% | 48s | 10.54 | 1.63% | 2m | 16.09 | 0.51% | 1m | 4.53 | 1.17% | 1m20s | 7.01 | 2.19% | 1m25 |
| AM-XL | 5.70 | 0.04% | 48s | 10.53 | 1.50% | 2m | 16.08 | 0.57% | 1m | 4.54 | 1.25% | 1m20s | 7.00 | 2.04% | 1m25 |
| AM-PPO | 5.70 | 0.03% | 35s | 10.51 | 1.45% | 1m | 16.09 | 0.49% | 35s | 4.49 | 0.89% | 38s | 6.98 | 1.75% | 55m |

- `AM-XL` is AM that adopts `POMO`-style MHA encoder, using six MHA layers and InstanceNorm instead of BatchNorm. We train `AM-XL` on the same number of samples as `POMO`.

- `A2C` is a variant of AM trained with Advantage Actor-Critic (A2C). It evaluates the rewards using the learned critic during training. We adopt Kool et al. (2019)'s implementation.

- `AM-PPO` is an A2C model trained with Proximal Policy Optimization (PPO, Schulman et al. (2017)) algorithm.

For fairness of comparison, we try to match the number of training steps to be the same and adjust the batch size accordingly. Specifically, we train models for 100 epochs as in Kool et al. (2019) using the Adam optimizer (Kingma & Ba, 2014) with an initial learning rate (LR) of 0.001 with a decay factor of 0.1 after the 80th and 95th epochs[6]. We provide additional details in Appendix D.

**Environments** We benchmark the NCO solvers on canonical routing problems: the Traveling Salesman Problem (TSP), the Capacitated Vehicle Routing Problem (CVRP), the Orienteering Problem (OP), and the Prize Collecting TSP (PCTSP), as evaluated in Kool et al. (2019). Additionally, we examine a relatively underexplored problem that often arises in real world applications, the Pickup and Delivery Problem (PDP), following the implementation of Li et al. (2021a). We also benchmark the Decap Placement Problems (DPP) from electronic design automation in Appendix B. Further details on environment implementations and data generation are provided in Appendix C.

---

[6]We find that simple learning rate scheduling with `MultiStepLinear` can improve performance i.e., compared to the original AM implementation.

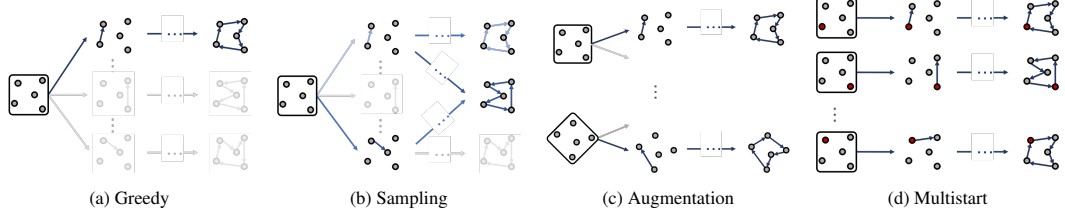

(a) Greedy          (b) Sampling          (c) Augmentation          (d) Multistart

Figure 4.1: Decoding schemes of the autoregressive NCO solvers evaluated in this paper.

**Decoding Schemes**   The solution quality of NCO solvers often shows large variations in performances to the different decoding schemes, even though using the same NCO solvers. Regarding that, we evaluate the trained solvers using five schemes shown in Fig. 4.1:

- `Greedy`: elects the highest probabilities at each decoding step.
- `Sampling`: concurrently samples $N$ solutions using a trained stochastic policy.
- `Multistart Greedy`: inspired by POMO, decodes from the first given nodes and considers the best results from $N$ cases starting at $N$ different cities. For example, in TSP with $N$ nodes, a single problem involves starting from $N$ different cities.
- `Augmentation`: selects the best greedy solutions from randomly augmented problems (e.g., random rotation and flipping) during evaluation.
- `Multistart Greedy + Augmentation`: combines Multistart Greedy and Augmentation.

## 4.2 BENCHMARK RESULTS

**In-distribution**   We first measure the performances of NCO solvers on the same dataset distribution on which they are trained. The results for training on 50 nodes are summarized in Table 4.1. We first observe that, counter to the commonly known trends that AM < POMO < Sym-NCO, the trends can change to decoding schemes and targeting CO problems. Especially when the solver decodes the solutions with `Augmentation` or `Greedy Multistart + Augmentation`, the performance differences among the benchmarked solvers on TSP and CVRP become less significant.

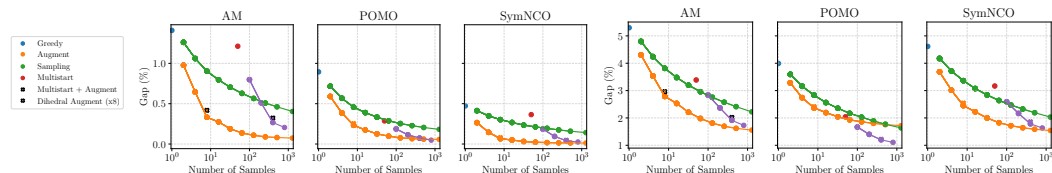

Figure 4.2: Pareto front of decoding schemes by number of samples. Left: TSP50; right: CVRP50.

We note that the original implementation of POMO [7] is not directly applicable to OP, PCTSP and PDP. Adapting it to solve new problems is not straightforward due to the coupling between environment and policy implementations. However, owing to the flexibility of RL4CO, we successfully implemented POMO for OP and PCTSP. Our results indicate that POMO underperforms in OP and PCTSP; unlike TSP, CVRP, and PDP, where all nodes need to be visited, OP and PCTSP are not constrained to visit all nodes. Due to such differences, POMO's visiting all nodes strategy may not work as an effective inductive bias. Further, we benchmark the NCO solvers for PDP, which is not originally supported natively by each of the benchmarked solvers. We apply the environment embeddings and the Heterogeneous Attention Encoder from HAM (Li et al., 2021a) to the NCO models for encoding pickup and delivery pairs, further emphasizing RL4CO's flexibility. We observe that AM-XL, which employs the same RL algorithm as AM but features the encoder architecture of POMO and is trained with an equivalent number of samples, yields performance comparable to NCO solvers using more sophisticated baselines. This suggests that careful controls on architecture and the number of training samples are required when evaluating NCO solvers.

---

[7]https://github.com/yd-kwon/POMO

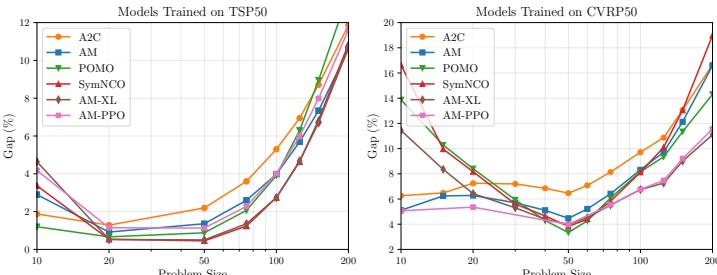

Figure 4.3: Validation cost curves and number of training samples consumed. Models with greater performance after full training may show worse convergence properties when number of training samples is limited.

Figure 4.4: Out-of-distribution generalization for models trained on 50 nodes. Stronger performance in distribution does not always translate to out-of-distribution.

During inference, investing more computational resources (i.e., increasing the number of samples), the trained NCO solver can discover improved solutions. We examine the performance gains achieved with varying numbers of samples. As shown in Fig. 4.2, the Augmentation decoding scheme achieves the Pareto front with limited samples and, notably, generally outperforms other decoding schemes.

We also evaluate the NCO solvers based on the number of training samples (i.e., the number of reward evaluations). As shown in Fig. 4.3, we found that actor-critic methods (e.g., A2C and PPO) can exhibit efficacy in scenarios with limited training samples, as demonstrated by the TSP50/100 results in Fig. 4.3. This observation suggests that NCO solvers with control over the number of samples may exhibit a different trend from the commonly recognized trends. In the extension of this viewpoint, we conducted additional benchmarking in a different problem domain: electronic design automation (EDA) - detailed in Appendix B - where reward evaluation is resource-intensive due to the necessity of electrical simulations, in which sample efficiency becomes even more crucial.

**Out-of-distribution**    In this section, we evaluate the out-of-distribution performance of the NCO solvers by measuring the optimality gap compared to the best-known tractable solver. The evaluation results are visualized in Fig. 4.4. Contrary to the in-distribution results, we find that NCO solvers with sophisticated baselines (i.e., POMO and Sym-NCO) tend to exhibit worse generalization when the problem size changes, either for solving smaller or larger instances. This can be seen as an indication of "overfitting" to the training sizes. On the other hand, variants of AM show relatively better generalization results overall. We also evaluate the solvers in two canonical public benchmark instances (TSPLib and CVRPLib) in Appendix F, which exhibit both variations in the number of nodes as well as their distributions and find a similar trend.

**Search methods**    One viable and prominent approach of NCO that mitigates distributional shift (e.g., the size of problems) is the (learnable) search methods, which involve training (a part of) a pre-trained NCO solver to adapt to CO instances of interest. We implement and evaluate the following search methods:

- AS: Active Search from Bello et al. (2017) finetunes a pre-trained model on the searched instances by adapting all the policy parameters.

- EAS: Efficient Active Search from Hottung et al. (2022) finetunes a subset of parameters (i.e., embeddings or new layers) and adds an imitation learning loss to improve convergence.

Table 4.2: Search Methods results of models pre-trained on 50 nodes. *Classic* refers to Concorde (David Applegate & Cook, 2023) for TSP and HGS (Vidal, 2022) for CVRP. OOM denotes "Out of Memory".

| Type | Metric | TSP | | | | | | CVRP | | | | | |
|---|---|---|---|---|---|---|---|---|---|---|---|---|---|
| | | POMO | | | Sym-NCO | | | POMO | | | Sym-NCO | | |
| | | 200 | 500 | 1000 | 200 | 500 | 1000 | 200 | 500 | 1000 | 200 | 500 | 1000 |
| *Classic* | Cost | 10.17 | 16.54 | 23.13 | 10.72 | 16.54 | 23.13 | 27.95 | 63.45 | 120.47 | 27.95 | 63.45 | 120.47 |
| *Zero-shot* | Cost | 13.15 | 29.96 | 58.01 | 13.30 | 29.42 | 56.47 | 29.16 | 92.30 | 141.76 | 32.75 | 86.82 | 190.69 |
| | Gap[%] | 29.30 | 81.14 | 150.80 | 24.07 | 77.87 | 144.14 | 4.33 | 45.47 | 17.67 | 17.17 | 36.83 | 58.29 |
| | Time[s] | 2.52 | 11.87 | 96.30 | 2.70 | 13.19 | 104.91 | 1.94 | 15.03 | 250.71 | 2.93 | 15.86 | 150.69 |
| *AS* | Cost | 11.16 | 20.03 | OOM | 11.92 | 22.41 | OOM | 28.12 | 63.98 | OOM | 28.51 | 66.49 | OOM |
| | Gap[%] | 4.13 | 21.12 | OOM | 11.21 | 35.48 | OOM | 0.60 | 0.83 | OOM | 2.00 | 4.79 | OOM |
| | Time[s] | 7504 | 10070 | OOM | 7917 | 10020 | OOM | 8860 | 21305 | OOM | 9679 | 24087 | OOM |
| *EAS* | Cost | 11.10 | 20.94 | 35.36 | 11.65 | 22.80 | 38.77 | 28.10 | 64.74 | 125.54 | 29.25 | 70.15 | 140.97 |
| | Gap[%] | 3.55 | 26.64 | 52.89 | 8.68 | 37.86 | 67.63 | 0.52 | 2.04 | 4.21 | 4.66 | 10.57 | 17.02 |
| | Time[s] | 348 | 1562 | 13661 | 376 | 1589 | 14532 | 432 | 1972 | 20650 | 460 | 2051 | 17640 |

We apply AS and EAS to POMO and Sym-NCO pre-trained on TSP and CVRP with 50 nodes to solve larger instances having $N \in [200, 500, 1000]$ nodes. As shown in Table 4.2, solvers with search methods improve the solution quality. However, POMO generally shows better improvements over Sym-NCO. This suggests once more that the "overfitting" of sophisticated baselines can perform better in-training distributions but eventually worse in different downstream tasks.

## 4.3 DISCUSSION

**Limitations** In this paragraph we identify some limitations and areas of improvement with our current library and benchmark experiments. Regarding benchmarking experiments, we mainly focus on training the solvers on relatively smaller sizes due to our limited computational budget. We also focus mainly on benchmarking the change of sizes rather than the change of data-generating distribution as in Bi et al. (2022), which could yield interesting results - although we do benchmark real-world instances of TSPLib and CVRPLib that include both new size and data distributions. In terms of the RL4CO library, we prioritized implementing RL AR approaches due to their flexibility, even though NAR approaches such as the ones trained with SL show good generalization performance in relatively less constrained routing problems as TSP (Joshi et al., 2019; Sun & Yang, 2023), although they require labeled solution datasets.

**Future Directions in RL4CO** As extensions of RL4CO library, we plan to support more methods aside from AR approaches. Given that recent NAR models often decode solutions in an AR manner to enforce the feasibility of the generated solutions, we are expecting supporting NAR methods (Kool et al., 2021; Xiao et al., 2023) would be possible under a unified manner in RL4CO. Improvement heuristics (Wu et al., 2021), which learn to improve suboptimal complete solutions (Hou et al., 2023; Xin et al., 2021; Kim et al., 2021; 2022a; Ye et al., 2023) are also an interesting future direction we would like to pursue. Such improvement heuristics would serve as useful toolkits for researchers and practitioners as they are applicable on top of existing NCO solvers with anytime-ness (i.e., can limit the runtime while having complete solutions). Furthermore, hybrid approaches that divide and conquer problems (Kim et al., 2021; Hou et al., 2023) via global and local policies are an exciting avenue for future works. Finally, we would like to support a wider variety of real-world problems, such as routing problems with soft/hard time windows.

## 5 CONCLUSION

This paper introduces RL4CO, a modular, flexible, and unified Reinforcement Learning (RL) for Combinatorial Optimization (CO) library. Our software library aims to fill the gap in unifying implementations for NCO by utilizing several best practices with the goal of providing researchers and practitioners with a flexible starting point for NCO research. With RL4CO, we benchmarked NCO solvers with experiments showing that a comparison of NCO solvers across different metrics and tasks is fundamental, as state-of-the-art approaches may perform worse than predecessors depending on specific settings. We hope that our benchmark library will provide a solid starting point for NCO researchers to explore new avenues and drive advancements.

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

## A    ADDITIONAL DISCUSSION ABOUT THE RL4CO LIBRARY

### A.1    WHY CHOOSING RL4CO?

In this paper, we introduce RL4CO, a *modular*, *flexible*, and *unified* implementation of NCO. In designing the library, we intended RL4CO to be used for various purposes ranging from research to production. RL4CO enables the users to have the following benefits.

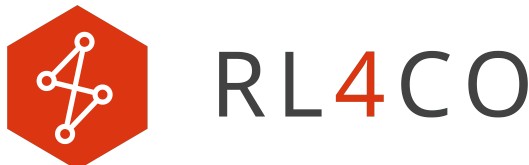

Figure A.1: RL4CO logo.

**Minimal Implementation of Boilerplate Codes for NCO**    As with the other RL projects, implementation of NCO with RL involves designing and coding the systems composed mainly of agents (i.e., policy) and environment (i.e., CO problem). However, this often involves a serious amount of engineering, especially to attain higher executions in training routines. Moreover, we found that a significant chunk of NCO solvers are based on AM or POMO implementation and the subroutines that have been done in the implementation. Regarding those practical aspects, RL4CO provides a modularized code base for each routine of NCO, including environment, policy network architecture, RL algorithm, and training. So that the users can easily mix and match SoTA NCO practices and user-defined modules while having full control over the entire RL pipeline.

**Easier Comparison Among NCO Algorithms**    Current NCO research shows a tendency to rely on two cornerstone implementations: AM and POMO. However, due to differences in implementation (e.g., network architecture, training scheme), direct head-to-head comparisons among the algorithms might not be straightforward. For example, applying POMO's state augmentation to AM's policy to reveal the effect of augmentation from the baseline selections can be challenging. RL4CO provides a unified implementation of NCO models (and their subroutines) to offer higher adaptability of routines from one algorithm to another. We believe this will promote easier comparisons among the models while developing novel NCO solvers to address various CO problems.

**Leveraging Standardized Open-Source Libraries**    During the development of RL4CO, we have decided to utilize standardized and reputable open-source libraries based on extensive research and expertise at the edge of software engineering and research, such as the recent TorchRL (Bou et al., 2023) and the `TensorDict` data structure from Moens (2023a). We believe these design choices will yield various practical benefits both in research and production. For instance, by disentangling the RL algorithm from the training subroutines of RL4CO, NCO solvers can undergo training using an array of state-of-the-art training methods supported by PyTorch Lightning (Falcon & The PyTorch Lightning team, 2019). Additionally, deploying the trained NCO solvers to production becomes seamless through the utilization of tools such as TorchServe, to name just a couple of examples.

### A.2    ON THE CHOICE OF THE SOFTWARE BACKBONE

During the development of RL4CO, we wanted to make it as simple as possible to integrate reproducible and standardized code adhering to the latest guidelines. As a main template for our codebase, we use Lightning-Hydra-Template [8] which we acknowledge being a solid starting point for reproducible deep learning. We further discuss framework choices below.

**TorchRL and TensorDict**    One of the software hindrances in RL is the bottleneck between CPU and GPU communication majorly due to CPU-based operating environments. For this reason, we did not opt for OpenAI Gym (Brockman et al., 2016) since, although it includes some level of

---

[8]https://github.com/ashleve/lightning-hydra-template

parallelization, this does not happen on GPU and would thus greatly hinder performance. Kool et al. (2019) creates *ad-hoc* environments in PyTorch to handle batched data efficiently. However, it could be cumbersome to integrate into standardized routines that include `step` and `reset` functions. As we searched for a better alternative, we found that TorchRL library (Moens, 2023b), an official PyTorch project that allows for efficient batched implementations on (multiple) GPUs as well as functions akin to OpenAI Gym. We also employ the TensorDict (Moens, 2023b) to handle tensors efficiently on multiple keys (i.e. in CVRP, we can directly operate transforms on multiple keys as locations, capacities, and more). This makes our environments compatible with the models in TorchRL, which we believe could further the interest in the CO area.

**PyTorch Lightning**   PyTorch Lightning (Falcon & The PyTorch Lightning team, 2019) is a useful tool for abstracting away the boilerplate code allowing researchers and practitioners to focus more on the core ideas and innovations. With its standardized training loop and extensive set of pre-built components, including automated checkpointing, distributed training, and logging, PyTorch Lightning accelerates development time and facilitates scalability. We employ PyTorch Lightning in RL4CO to integrate with the PyTorch ecosystem - which includes TorchRL- enabling us to leverage the rich set of tools and libraries available.

**Hydra**   Hydra (Yadan, 2019) is a powerful open-source framework for managing complex configurations in machine learning models and other software in the form of `yaml` files. Hydra facilitates creating hierarchical configurations, making it easy to manage even very large and intricate configurations. Moreover, it integrates with command-line interfaces, allowing the execution of different configurations directly from the command line, thereby enhancing reproducibility. We found Hydra to be effective when dealing with multiple experiments, since configurations are saved both locally as `yaml` files as well as uploaded on Wandb [9].

## A.3   OPEN SOURCE LICENSES

In this paragraph, we summarize the license of the software that we've employed in developing RL4CO. The license lists are as follows:

- PyTorch, Matplotlib: BSD license
- TorchRL, Einops, Hydra: MIT license
- Lightning: Apache-2.0 license
- Numpy, Scipy: BSD-3-Clause license

For reproducing the previous NCO research, we refer to the original implementation of AM [10], the original implementation of POMO [11], and the original implementation of SymNCO [12]. However, while faithful, our implementation considerably modifies the structure of the original implementation introducing several optimizations to single lines of codes to the overall structure. RL4CO is published under the liberal Apache-2.0 license.

## A.4   DECODER WITH DYNAMICEMBEDDING

RL4CO provides a modular, flexible, and unified policy network design that is used for various RL algorithms to solve multiple CO problems. In this section, we provide additional discussion about the decoder structure with the dynamic embedding module that is used to solve complex routing problems involving information updates during decoding. The `DynamicEmbedding` module dynamically updates the keys and values of multi-head-attention (MHA) and softmax to reflect the 'dynamic' change of information-related unselected nodes (e.g., unvisited cities in VRPs) during decoding. To be specific, the decoder with `DynamicEmbedding` is defined as follows:

$$q_t = \texttt{ContextEmbedding}(\boldsymbol{h}, a_{t-1:0}), \tag{10}$$

---

[9] https://wandb.ai/
[10] https://github.com/wouterkool/attention-learn-to-route
[11] https://github.com/yd-kwon/POMO
[12] https://github.com/alstn12088/Sym-NCO

$$\boldsymbol{K}_t^g, \boldsymbol{V}_t^g, \boldsymbol{V}_t = \texttt{DynamicEmbedding}(W_k^g \boldsymbol{h}, W_v^g \boldsymbol{h}, W_v \boldsymbol{h}, a_{t-1:0}, \boldsymbol{h}, \boldsymbol{x}), \tag{11}$$

$$\bar{q}_t = \text{MHA}(q_t, \boldsymbol{K}_t^g, \boldsymbol{V}_t^g), \tag{12}$$

$$\pi(a_t) = \text{MaskedSoftmax}(\bar{q}_t \cdot \boldsymbol{V}_t, M_t), \tag{13}$$

where the dynamic embedding dynamically modulates keys and values of MHA and softmax based on the decoded results (i.e., $a_{t-1:0}$) and inputs (i.e., $\boldsymbol{x}$ and $\boldsymbol{h}$).

## A.5 QUICKSTART NOTEBOOK

We showcase a quickstart Jupyter Notebook for training the Attention Model on TSP with 20 nodes that can be run in less than 2 minutes on Google Colab (Bisong & Bisong, 2019) on a free-tier GPU runtime[13].

```python
import torch
from rl4co.envs import TSPEnv
from rl4co.models.zoo.am import AttentionModel
from rl4co.utils.trainer import RL4COTrainer

# RL4CO env based on TorchRL
env = TSPEnv(num_loc=20)

# Model: default is AM with REINFORCE and greedy rollout baseline
model = AttentionModel(env,
                       baseline='rollout',
                       train_data_size=100_000,
                       val_data_size=10_000)
trainer = RL4COTrainer(max_epochs=3)

# Greedy rollouts over untrained model
device = torch.device("cuda" if torch.cuda.is_available() else "cpu")
td = env.reset(batch_size=[3]).to(device)
model = model.to(device)
out = model(td, phase="test", decode_type="greedy", return_actions=True)

# Plotting
print(f"Tour lengths: {[f'{-r.item():.2f}' for r in out['reward']]}")
for td_, actions in zip(td, out['actions'].cpu()):
    env.render(td_, actions)
```

Tour lengths: ['10.38', '7.69', '8.32']

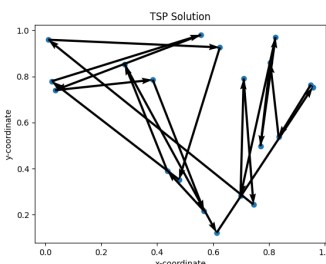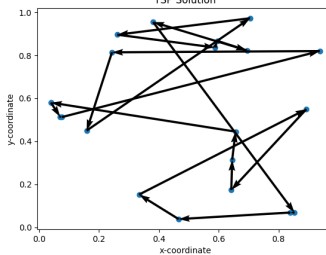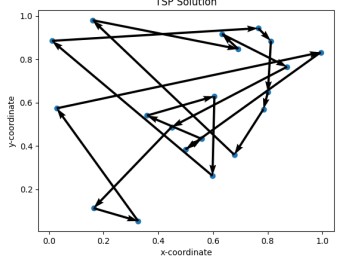

---

[13]

```
# RL4COTrainer with few epochs (wrapper around Lightning Trainer)
trainer = RL4COTrainer(
    max_epochs=3,
    accelerator="gpu",
    logger=None,
)

# Fit the model
trainer.fit(model)
```

```
INFO:
  | Name  | Type           | Params
----------------------------------------
0 | env   | TSPEnv         | 0
1 | model | AttentionModel | 1.4 M
----------------------------------------
1.4 M     Trainable params
0         Non-trainable params
1.4 M     Total params
5.669     Total estimated model params size (MB)

Epoch 2: [||||||||||||||||||||||||||||||||||||||||||||||||||||||||||] 100% 196/196
[00:44<00:00, 4.36it/s, train/reward=-4.09, train/loss=-.104, val/reward=-4.05]
```

```
# Greedy rollouts over trained model (same states as previous plot)
model = model.to(device)
out = model(td, phase="test", decode_type="greedy", return_actions=True)

# Plotting
print(f"Tour lengths: {[f'{-r.item():.2f}' for r in out['reward']]}")
for td_, actions in zip(td, out['actions'].cpu()):
    env.render(td_, actions)
```

```
Tour lengths: ['3.35', '3.51', '4.22']
```

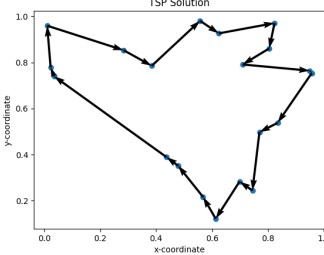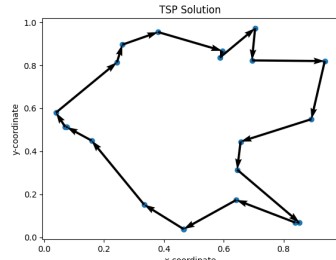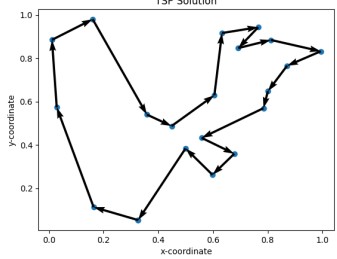

# B    ELECRONIC DESIGN AUTOMATION: THE DECAP PLACEMENT PROBLEM

## B.1    INTRODUCTION

The optimal placement of a given number of decoupling capacitors (decaps) can significantly impact electrical performance, specifically in terms of power integrity (PI) optimization. PI optimization is crucial in modern chip design, especially with the preference for 3D stacking memory systems like high bandwidth memory (HBM), shown in Fig. B.1. One of the challenges in power supplementation is the vertical transmission of power to 3D memory, which is located at the bottom of memory chips. Consequently, the optimal placement of decaps becomes increasingly important to support the current progress in 3D chip design and high bandwidth, which is critical for meeting the high memory requirements of deep learning technology (Hwang et al., 2021).

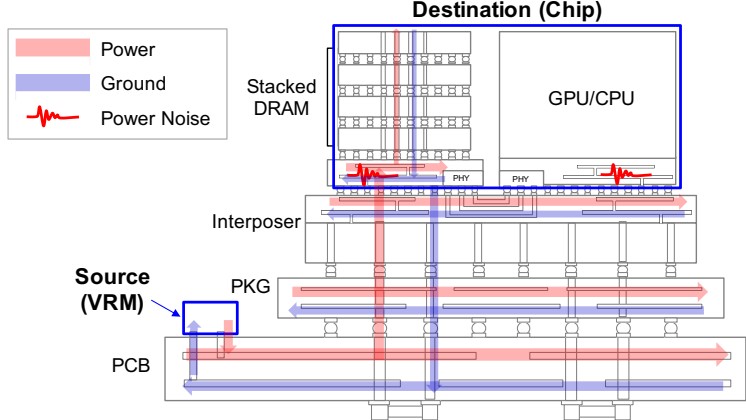

Figure B.1: An example of the power distribution network (PDN) of a high bandwidth memory (HBM) module for a high-performance AI computing system.

The optimal placement of decaps, also known as the decap placement problem (DPP), involves combinatorial optimization with a black box electrical simulator as the scoring function. Typically, this simulator is an expensive electromagnetic (EM) simulator that requires significant computational resources. To address this, we propose a fast approximated simulator that can be executed within a Python environment. While sacrificing some accuracy, our circuit-modeled simulation serves as a proxy score function. Additionally, we limit the number of simulation calls due to the high cost associated with running the real-world EM simulator.

The decap placement problem (DPP) is a highly complex task in hardware design due to two main reasons. Firstly, it involves exploring an extensive range of possibilities in a combinatorial space. Secondly, evaluating the objectives of DPP requires significant computational resources and time, making it necessary to achieve high sample efficiency. Genetic algorithm (GA) based methods have shown promise in addressing DPP because they can reduce the combinatorial space compared to exhaustive search methods (Juang et al., 2021). However, GAs still require a large number of iterations (M) for each problem as they are memoryless optimization methods. Reinforcement learning approaches have been attempted to solve DPP, and the most recently proposed DevFormer achieved the state of the art performance while achieving sample efficient in training and zero-shot inference (Kim et al., 2023).

## B.2    PROBLEM SETTING

The decap placement problem (DPP) represents a sample-efficient setting of NCO, in which calculating the reward is expensive due to complex simulation, and hence the number of samples should be limited (i.e. $10,000$ samples for pretraining each model in our experiments). The DPP aims to determine the optimal locations for decaps within a given instance. This instance consists of three key components: (1) the locations of probing ports, (2) the locations of keep-out regions (where decaps cannot be placed), and (3) the available regions where decaps can be positioned as shown in

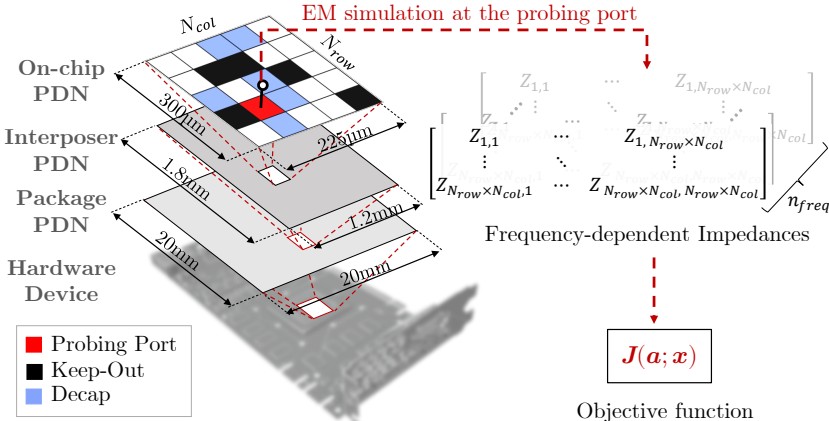

Figure B.2: Grid representation of the target on-chip PDN for the DPP problem with a single probing port.

Fig. B.2. For comprehensive details, we follow the configuration guidelines provided in (Kim et al., 2023).

**Environments: DPP and mDPP**    In order to compare our approach with the somewhat simplistic solution to DPP presented in (Kim et al., 2023), which emphasizes placing decaps near probing ports as a good solution, we introduce multi-port DPP (mDPP) and adopt different objectives:

- *Maxsum*: the objective is to maximize the average PI among multiple probing ports

- *Maxmin*: the objective is to maximize the minimum PI among multiple probing ports

These more intricate and practical approaches add complexity to the problem, enabling us to assess the effectiveness of the proposed method in a real-world scenario. Additional details, including data generation, can be found in Appendix C.6 and Appendix C.7.

**Baselines**    We employ two meta-heuristic baselines commonly used in hardware design as outlined in (Kim et al., 2023): random search (RS) and genetic algorithm (GA). GA has shown promise as a method for addressing the decap placement problem (DPP). In addition, we introduce DevFormer (Kim et al., 2023) (DF), an AM-variant model specifically designed for DPP. It is important to note that DevFormer is initially designed for offline training; however, in this study, we benchmark DevFormer as a sample-efficient online reinforcement learning approach.

We benchmark the DevFormer version for RL with the same embedding structure as the original in (Kim et al., 2023). We benchmark 3 variants, namely DF(PG,Critic): REINFORCE with Critic baseline, DF(PG,Rollout): REINFORCE with Rollout baseline as well as PPO. All experiments are run with the same hyperparameters as the other experiments except for the batch size set to $64$, maximum number of samples set to $10,000$, and a total of only $10$ epochs due to the nature of the benchmark sample efficiency.

### B.3    Benchmark Results

**Main benchmark**    Table E.2 shows the main numerical results for the task when RS, GA and DF models are trained for placing 20 decaps. While RS and GA need to take online shots to solve the problems (we restricted the number to $100$), DF models can successfully predict in a zero-shot manner and outperform the classical approaches. Interestingly, the vanilla critic-based method performed the worst, while our implementation of PPO almost matched the rollout policy gradients (PG) baseline; since extensive hyperparameter tuning was not performed, we expect PPO could outperform the rollout baseline given it requires fewer samples. Fig. B.3 shows example renderings of the solved environment.

Table B.1: Performance of different methods on the mDPP benchmark

| Method | # Shots | Score ↑ | |
|---|---|---|---|
| | | maxsum | maxmin |
| *Online Test Time Search* | | | |
| Random Search | 100 | 11.55 | 10.63 |
| Genetic Algorithm | 100 | 11.93 | 11.07 |
| *RL Pretraining & Zero Shot Inference* | | | |
| DF-(PG,Critic) | 0 | $10.89 \pm 0.63$ | $9.51 \pm 0.68$ |
| DF-(PPO) | 0 | $12.16 \pm 0.03$ | $11.17 \pm 0.11$ |
| DF-(PG,Rollout) | 0 | $12.21 \pm 0.01$ | $11.26 \pm 0.03$ |

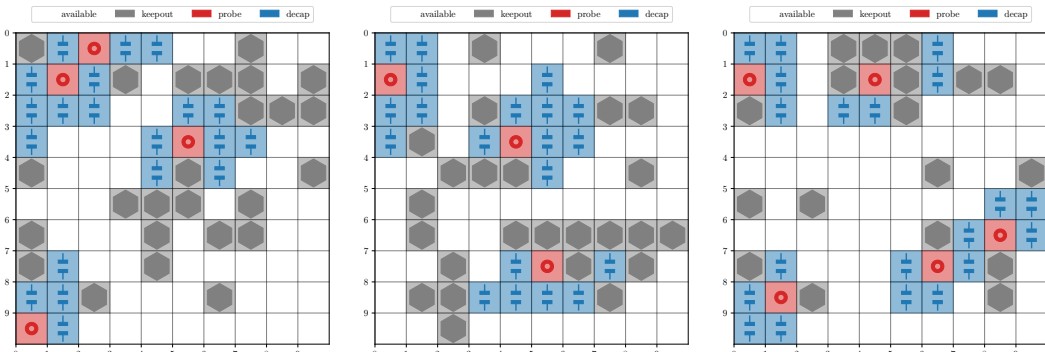

Figure B.3: Renders of the environment with *maxmin* objective solved by DF-(PG,Rollout). The model successfully learned one main heuristic for DPP problems, which is that optimal placement is generally close to probing ports.

**Generalization to different number of decaps**    In hardware design, the number of components is one major contribution to cost; ideally, one would want to use the least number of components possible with the best performance. In the DPP, increasing the number of decaps *generally* improves the performance at a greater cost, hence Pareto-efficient models are essential to identify. Fig. B.4 shows the performance of DF models trained on 20 decaps against the baselines. DF models PPO and PG-rollout can successfully generalize and are also Pareto-efficient with fewer decaps, important in practice for cost and material saving.

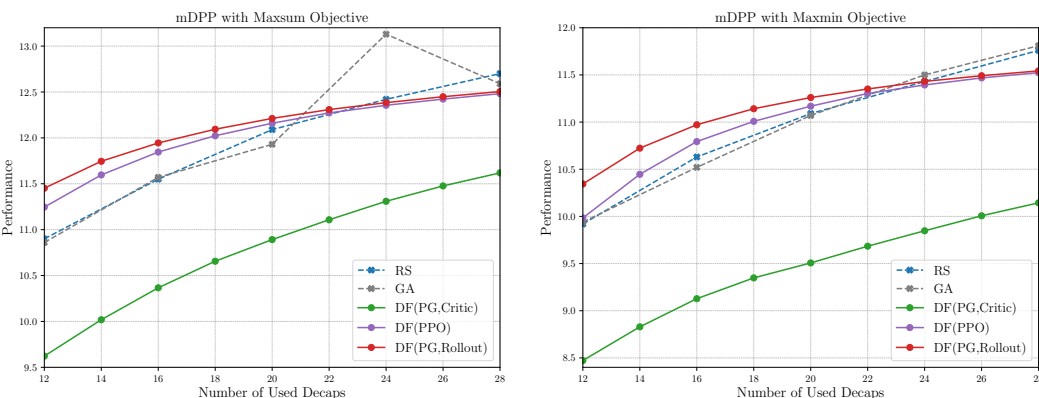

Figure B.4: Performance vs number of used decaps for mDPP with *maxsum* objective [Left] and *maxmin* objective [Right].

## C    ENVIRONMENTS

In this section, we discuss further details of environment implementation. As Kool et al. (2019) environment implementation de-facto standard of implementation, RL4CO also aims to reproduce the same environment settings along with generating the same data.

**Common instance generation details**    Following the standard protocol of NCO, we randomly sample node coordinates from the 2D unit square (i.e., $[0, 1]^2$). When generating the training data, we regulate the randomness by setting the random seed to 1234. Conversely, when generating 10,000 validation instances, we use random seed 4321. For the creation of the testing 10,000 instances, we use the random seed 1234. All protocols, including seed selection, align with the practices outlined by Kool et al. (2019).

### C.1    TRAVELING SALESMAN PROBLEM (TSP)

The Traveling Salesman Problem (TSP) is a fundamental routing problem that aims to find the Hamiltonian cycle of minimum length. While the original TSP formulation employs mixed-integer linear programming (MILP), to integrate TSP into autoregressive solution decoding (i.e., the construction process), we reinterpret the solution-finding process as sequential node selection in line with Kool et al. (2019). In each step of node selection, we preclude the selection of nodes already picked in previous rounds. This procedure ensures the feasibility of constructed solutions and also allows for the potential construction of an optimal solution for any TSP instance.

### C.2    CAPACITATED VEHICLE ROUTING PROBLEM (CVRP)

The Capacitated Vehicle Routing Problem (CVRP) is a popular extension of TSP, applicable to a variety of real-world logistics/routing problems (e.g., delivery services). In CVRP, each node has its own demand, and each vehicle has a specific capacity. A vehicle can "tour" until the total demand does not exceed its capacity. The vehicle returns to the depot - a unique type of node - and releases all demand, then embarks on another tour from the depot until all nodes have been visited. Similar to TSP, the original CVRP formulation utilizes MILP. However, by applying a similar logic to that of the TSP environment, we can reformulate CVRP as a sequential node selection problem, taking into account demands and capacity.

**Additional generation details**    To generate demand, we randomly sample integers between 1 and 10. Without loss of generality, we fix the capacity of the vehicle at 1.0. Instead, we normalize demand by multiplying it by a constant that varies according to the size of the CVRP. The specific constant can be found in our implementation.

### C.3    ORIENTEERING PROBLEM (OP)

The Orienteering Problem (OP) is a variant of TSP. In OP, each node (i.e., city) is assigned a prize. The objective of OP is to find a tour, starting and ending at the depot, that maximizes the total prize collected from visited cities, while abiding by a maximum tour length constraint. Like the previous problems, the original formulation utilizes MILP. However, OP can also be framed as a sequential decision-making problem by enforcing the "return to depot" action when no cities are visitable due to the maximal tour length constraint.

**Additional generation details**    To generate the prize, we use the prize distribution proposed in Fischetti et al. (1998), particularly the distribution that allocates larger prizes to nodes further from the depot.

### C.4    PRIZE COLLECTING TSP (PCTSP)

In the Prize Collecting TSP (PCTSP), each node is assigned both a prize and a penalty. The objective is to accumulate a minimum total prize while minimizing the combined length of the tour and the penalties for unvisited nodes. By making a minor adjustment to PCTSP, it becomes applicable

for solving different subproblems that arise in routing problems when using Branch-Price-and-Cut algorithms.

### C.5 PICKUP AND DELIVERY PROBLEM (PDP)

The Pickup and Delivery Problem (PDP) is an extension of TSP. In PDP, a pickup node has its own designated delivery node. The delivery node can be visited only when its paired pickup node is already visited, so-called, precedence constraints. The objective of PDP is to find a complete tour with a minimal tour length while starting from the depot node and satisfying the precedence constraints. We assume the 'stacking' is allowed, where the traveling agent can visit multiple pickups prior to visiting the paired deliveries.

**Additional generation details**   To generate the positions of the depot, pickups, and deliveries, we sample the node coordinates from the 2D unit square.

### C.6 DECAP PLACEMENT PROBLEM (DPP)

The decap placement problem (DPP) is an electronic design automation problem (EDA) in which the goal is to maximize the performance with a limited number of the decoupling capacitor (decap) placements on a hardware board characterized by asymmetric properties, measured via a probing port. The decaps cannot be placed on the location of the probing port or in keep-out regions (which represent other hardware components). The full problem description is provided in Appendix B.

**Instance generation details**   We use the same data for simulating the hardware board as Kim et al. (2023). We randomly select one probing port and a number between 1 and 50 keep-out regions sampled from a uniform distribution for generating instances. As in the routing benchmarks, we select seed 1234 for testing the 100 instances.

### C.7 MULTI-PORT DECAP PLACEMENT PROBLEM (mDPP)

The multi-port decap placement problem (mDPP) is a generalization of DPP from Appendix C.6 in which measurements from multiple probing ports are performed. The objective function can be either the mean of the reward from the probing ports (*maxsum*) or the minimum between them (*maxmin*).The full problem description is provided in Appendix B.

**Instance generation details**   The generation details are the same as DPP, except for the probing port. A number between 2 and 5 probing ports is sampled from a uniform distribution and probing ports are randomly placed on the board as the other components.

### C.8 ADDITIONAL ENVIRONMENTS

We also include in the RL4CO library additional environments on which we did not benchmark models for the time being due to time and resource constraints: the Pickup and Delivery Problem and its multi-agent version (PDP and mPDP), the multiple Traveling Salesman Problem (mTSP), as well as asymmetric CO environments Asymmetric Traveling Salesman Problem (ATSP) and Flexible Flow Shop Problem (FFSP). We include also the Stochastic variant of PCTSP (SPCTSP) and a variation of the CVRP that allows for split deliveries to be considered, namely the Split Delivery Vehicle Routing Problem (SDVRP) - we show an example notebook on the latter under the `notebook/` folder of the library. Given both near and longer-term plans for the library and the RL4CO community, we expect to add several more variations as well as new environments in the future.

# D  EXPERIMENTAL DETAILS

## D.1  HARDWARE

Experiments were carried out on a machine equipped with two AMD EPYC 7542 32-CORE PRO-
CESSOR CPU with 64 threads and four NVIDIA RTX A6000 graphic cards with 48 GB of VRAM.

## D.2  SOFTWARE

Software-wise, we used Python 3.10 and the latest PyTorch 2.0 (Paszke et al., 2019) (dur-
ing development, we used beta wheels as well as manually installed version of FlashAt-
tention (Dao et al., 2022; Dao, 2023)), most notably due to the native implementation of
scaled_dot_product_attention. Given that most models in RL constructive methods for
CO generally use attention for encoding states, FlashAttention has some boost on the performance
(between 5% and 20% saved time depending on the problem size) when training is subject to mixed-
precision training, which we do for all experiments. During decoding, the FlashAttention routine is
not called since, at the time of writing, it does not support maskings other than causal; this could fur-
ther boost performance compared to older implementations. Refer to Appendix A.2 for additional
details regarding notable software choices of our library, namely TorchRL, PyTorch Lightning and
Hydra.

## D.3  COMMON HYPERPARAMETERS

Common hyperparameters can be found in the config/ folder from the RL4CO library, which
can be conveniently loaded by Hydra. We provide yaml-like configuration files below, divided by
experiments in Listing 1.

## D.4  CLASSICAL SOLVERS

We report performance in terms of optimality gaps compared with the best-known solutions we
obtain across methods. For the TSP, we use the Gurobi results (Gurobi Optimization, 2021) results
from (Kool et al., 2019) and Concorde (Mulder & Wunsch II, 2003; David Applegate & Cook, 2023).
For CVRP, we used HGS (Vidal, 2022) as a baseline, which is shown to be superior to alternatives.
For the OP, we report the results from Compass (Golden et al., 1987). As more general-purpose
solvers for routing problems, we also report LKH3 Helsgaun (2017) and Google OR Tools (Perron
& Furnon, 2023).

## D.5  MAIN TABLES

We run all models to try and match as much as possible the original implementation details. In
particular, we run all models for $250,000$ gradient steps with the same Adam (Kingma & Ba, 2014)
optimizer with a learning rate of $10^{-4}$ and 0 weight decay. For POMO, we match the original
implementation details of weight decay as $10^{-6}$. For POMO, the number of multistarts is the same
as the number of possible initial locations in the environment (for instance, for TSP50, 50 starts are
considered). In the case of Sym-NCO, we use 10 as augmentation for the shared baseline; we match
the number of effective samples of AM-XL to the ones of Sym-NCO to demonstrate the differences
between models.

```
   # RL Model configuration (policy+baseline)
 1
 2 model:
 3     policy:
 4         encoder:
 5             type: GraphAttentionEncoder
 6             num_heads: 8
 7             num_layers: 3 # POMO uses 6
 8             normalization: "batch" # POMO uses "instance"
 9             hidden_dim: 512
10             embedding_dim: 128
11         decoder:

12             num_heads: 8
13             embedding_dim: 128
14             use_graph_context: True # POMO does not use it
15             tanh_clipping: 10.0
16             mask_inner: True
17             mask_logits: True
18             normalize: True
19             softmax_temp: 1.0
20     baseline:
21         "rollout" # default baseline

23 # Training configuration

24 train:
25     optimizer:
26         type: Adam
27         learning_rate: 1e-4
28         weight_decay: 0 # POMO uses 1e-6
29     scheduler:
30         type: MultiStepLR # original AM implementation does not use
            ↪ this
31         step_size: [80, 95]
32         gamma: 0.1
33         scheduler_interval: epoch
34     gradient_clip_val: 1.0

35     max_epochs: 100 # we set AM-XL to 500
36     precision: "16-mixed" # allows for FlashAttention
37     strategy: DDPStrategy # efficient for multiple GPUs
38
```

Listing 1: Common configuration for RL model (policy + baseline) and training. Some differences are highlighted, such as the POMO implementation-level ones, and are provided as comments.

The number of epochs for all models is 100, except for AM-XL (500). We also employ learning rate scheduling, in particular, `MultiStepLR` [14] with $\gamma = 0.1$ on epoch 80 and 95; for AM-XL, this applies on epoch 480 and 495.

## D.6   SAMPLE EFFICIENCY EXPERIMENTS

We keep the same hyperparameters as Appendix D.5, except for the number of epochs and scheduling. We consider 5 independent runs that match the number of samples *per step* (i.e., the batch size is exactly the same for all models after considering techniques such as the multistart and symmetric baselines). For AM Rollout, we employ half the batch size of other models since it requires double the number of evaluations due to its baseline.

---
[14]https://pytorch.org/docs/stable/generated/torch.optim.lr_scheduler.MultiStepLR

### D.7 SEARCH METHODS EXPERIMENTS

For these experiments, we employ the same models trained in the in-distribution benchmark on 50 nodes. For Active Search (AS), we run 200 iterations for each instance and an augmentation size of 8. The Adam optimizer is used with learning rate of $2.6 \times 10^{-4}$ and weight decay of $10^{-6}$. For Efficient Active Search, we benchmark EAS-Lay (with an added layer during the single-head computation, `LogitAttention` in our code) with the original hyperparameters proposed by Hottung et al. (2022). The learning rate is set to $0.0041$ and weight decay to $10^{-6}$. The search is restricted to 200 iterations with dihedral augmentation of 8 as well as imitation learning weight $\lambda = 0.013$.

Testing is performed on 100 instances on both TSP and CVRP for $N \in [200, 500, 1000]$, generated with the usual random seed for testing 1234.

### D.8 PPO IMPLEMENTATION DETAILS

We also implemented PPO, the Attention Model trained with Proximal Policy Optimization (Schulman et al., 2017), which was not considered in previous works. We use the same critic network as the critic network for the critic REINFORCE baseline and perform training with the same settings as the AM-Critic based on REINFORCE.

Contrary to the common interpretation that views the solution generation process of CO problems (i.e., Eq. (1)) as a Markov Decision Process (MDP), where each decoding step corresponds to a state transition, we view it as a single-stage problem with an auto-regressive policy construction structure. Following this interpretation, the iterations in the decoding steps do not coincide with MDP's time step updates.

This interpretation requires some modifications to PPO, especially in computing the training labels of the value network (i.e., the value of the value function) to train NCO solvers. Since we have only one stage, our value function predicts the expected cost given the policy from the given problem instance. It is noteworthy that the problem is single-staged; hence, the Generalized Advantage Estimator (GAE) is not applicable, as GAE computes the values in the multi-stage setting. We found that recent fine-tuning of a large language model (Ouyang et al., 2022) carried out by PPO also interprets the decoding scheme as a single-state problem and hence does not apply GAE.

As for other hyperparameters, we set the number of epochs to 2, mini-batch size to $512$, clip range to $0.2$, and entropy coefficient $c_2 = 0.01$. Interestingly, we found that normalizing the advantage as done in the Stable Baselines PPO2 implementation[15] slightly hurt performance, so we set the normalize advantage parameter to `False`. We suspect this is because the NCO solvers are trained on *multiple* problem instances, unlike the other RL applications that aim to learn a policy for a single MDP.

---

[15]https://stable-baselines.readthedocs.io/en/master/modules/ppo2.html

# E ADDITIONAL EXPERIMENTS

## E.1 TRAINING ON SMALLER SIZES

In this section, we provide additional benchmark results on smaller instances, where the problems have 20 nodes. These include Table E.1[16], Fig. E.1 and Fig. E.2. The general trends of the results are similar to the ones observed from the problems with 50 nodes.

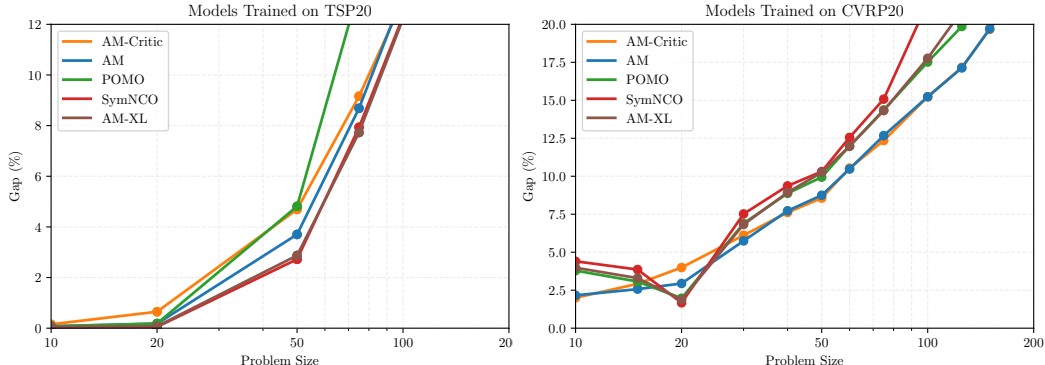

Figure E.1: Out-of-distribution generalization results. Models tend to perform better on problem of the same size of those seen during training.

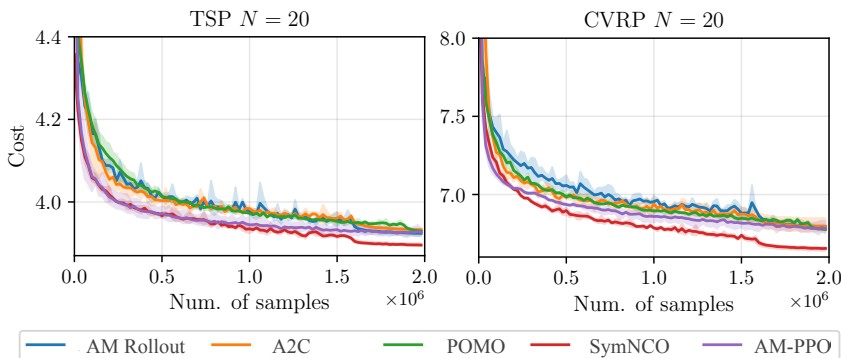

Figure E.2: Additional sample efficiency results for smaller problems with 20 nodes. Interestingly, SymNCO can outperform baselines even in TSP in terms of sample efficiency.

## E.2 FLOATING POINT PRECISION AND FLASH ATTENTION

RL4CO supports multiple device types as well as floating point precisions by leveraging PyTorch Lightning (Falcon & The PyTorch Lightning team, 2019).

As Appendix E.2 shows mixed-precision training can successfully reduce computational costs both in terms of runtime and especially with memory usage. Moreover, we designed our model to natively support FlashAttention (Dao et al., 2022; Dao, 2023) from both PyTorch 2.0 and the original FlashAttention repository [17].

As shown in Fig. E.3, different implementations can make a difference, especially with large problem sizes. It should be noted that while more scalable, FlashAttention at the moment is restricted to no or causal masks only. Therefore, usage in the masked attention decoding scheme is not possible

---

[16]We take the result for CPLEX from HAM (Li et al., 2021a).

[17]Available at https://github.com/Dao-AILab/flash-attention.

Table E.1: In-distribution results for model trained on 20 nodes.

| Method | TSP | | | CVRP | | | OP | | | PCTSP | | | PDP | | |
|---|---|---|---|---|---|---|---|---|---|---|---|---|---|---|---|
| | Cost ↓ | Gap | Time | Cost ↓ | Gap | Time | Prize ↑ | Gap | Time | Cost ↓ | Gap | Time | Cost ↓ | Gap | Time |
| | | | | | | | *Classical Solvers* | | | | | | | | |
| *Gurobi*[†] | 3.84 | 0.00% | 7s | – | – | – | – | – | – | – | – | – | – | – | – |
| *Concorde* | 3.84 | 0.00% | 1m | – | – | – | 5.39 | 0.00% | 16m | 3.13 | 0.00% | 2m | – | – | – |
| *HGS* | – | – | – | 6.13 | 0.00% | 4h | – | – | – | – | – | – | – | – | – |
| *Compass* | – | – | – | – | – | – | – | – | – | – | – | – | – | – | – |
| *LKH3* | 3.84 | 0.00% | 15s | 6.14 | 0.16% | 5h | – | – | – | – | – | – | – | – | – |
| *OR Tools* | 3.85 | 0.37% | 1m | – | – | – | – | – | – | 3.13 | 0.00% | 5h | 4.70 | 3.16% | (<1s) |
| *CPLEX* | – | – | – | – | – | – | – | – | – | – | – | – | 4.56 | 0.00% | 7m23s |
| | | | | | | | *Greedy One Shot Evaluation* | | | | | | | | |
| A2C | 3.86 | 0.64% | (<1s) | 6.46 | 5.00% | (<1s) | 5.01 | 6.70% | (<1s) | 3.36 | 7.35% | (<1s) | 5.50 | 20.61% | (<1s) |
| AM | 3.84 | 0.19% | (<1s) | 6.39 | 3.92% | (<1s) | 5.20 | 3.17% | (<1s) | 3.17 | 1.28% | (<1s) | 4.82 | 5.70% | (<1s) |
| POMO | 3.84 | 0.18% | (<1s) | 6.33 | 3.00% | (<1s) | 4.69 | 12.69% | (<1s) | 3.41 | 8.95% | (<1s) | 4.85 | 6.36% | (<1s) |
| Sym-NCO | 3.84 | 0.05% | (<1s) | 6.30 | 2.58% | (<1s) | 5.30 | 1.37% | (<1s) | 3.15 | 0.64% | (<1s) | 4.70 | 3.07% | (<1s) |
| AM-XL | 3.84 | 0.07% | (<1s) | 6.31 | 2.81% | (<1s) | 5.25 | 2.23% | (<1s) | 3.17 | 1.26% | (<1s) | 4.71 | 3.29% | (<1s) |
| | | | | | | | *Sampling with width $M = 1280$* | | | | | | | | |
| A2C | 3.84 | 0.15% | 20s | 6.26 | 2.08% | 24s | 5.12 | 4.66% | 22s | 3.28 | 4.79% | 23s | 5.06 | 10.96% | 23s |
| AM | 3.84 | 0.04% | 20s | 6.24 | 1.78% | 24s | 5.30 | 1.30% | 22s | 3.15 | 0.78% | 23s | 4.66 | 2.19% | 23s |
| POMO | 3.84 | 0.02% | 36s | 6.20 | 1.06% | 40s | 4.90 | 8.83% | 37s | 3.33 | 6.39% | 39s | 4.68 | 2.63% | 39s |
| Sym-NCO | 3.84 | 0.01% | 36s | 6.22 | 1.44% | 40s | 5.34 | 0.59% | 37s | 3.14 | 0.35% | 39s | 4.64 | 1.75% | 39s |
| AM-XL | 3.84 | 0.02% | 36s | 6.22 | 1.46% | 40s | 5.32 | 0.93% | 37s | 3.15 | 0.56% | 39s | 4.64 | 1.75% | 39s |
| | | | | | | | *Greedy Multistart ($N$)* | | | | | | | | |
| A2C | 3.85 | 0.36% | (<1s) | 6.33 | 3.04% | 3s | 5.06 | 5.77% | 2s | 3.30 | 5.18% | 2s | 5.18 | 13.60% | 23s |
| AM | 3.84 | 0.12% | (<1s) | 6.28 | 2.27% | 3s | 5.24 | 2.42% | 2s | 3.16 | 4.67% | 2s | 4.67 | 2.41% | 23s |
| POMO | 3.84 | 0.05% | (<1s) | 6.21 | 1.27% | 4s | 4.76 | 11.32% | 3s | 3.35 | 4.65% | 4s | 4.66 | 2.19% | 42s |
| Sym-NCO | 3.84 | 0.03% | (<1s) | 6.22 | 1.48% | 4s | 5.32 | 0.87% | 3s | 3.15 | 4.69% | 4s | 4.69 | 2.85% | 42s |
| AM-XL | 3.84 | 0.05% | (<1s) | 6.22 | 1.38% | 4s | 5.29 | 1.49% | 3s | 3.15 | 4.6% | 4s | 4.65 | 1.97% | 42s |
| | | | | | | | *Greedy with Augmentation (1280)* | | | | | | | | |
| A2C | 3.84 | 0.01% | 20s | 6.22 | 1.35% | 24s | 5.04 | 6.10% | 22s | 3.33 | 6.39% | 23s | 4.88 | 7.02% | 23s |
| AM | 3.84 | 0.00% | 20s | 6.20 | 1.07% | 24s | 5.25 | 2.25% | 22s | 3.16 | 0.96% | 23s | 4.63 | 1.54% | 23s |
| POMO | 3.84 | 0.00% | 36s | 6.18 | 0.84% | 45s | 4.85 | 9.76% | 38s | 3.37 | 7.55% | 42s | 4.62 | 1.32% | 42s |
| Sym-NCO | 3.84 | 0.00% | 36s | 6.17 | 0.71% | 45s | 5.33 | 0.77% | 38s | 3.15 | 0.63% | 42s | 4.61 | 0.95% | 42s |
| AM-XL | 3.84 | 0.00% | 36s | 6.17 | 0.68% | 45s | 5.30 | 1.30% | 38s | 3.15 | 0.68% | 42s | 4.61 | 0.96% | 42s |
| | | | | | | | *Greedy Multistart with Augmentation ($N \times 16$)* | | | | | | | | |
| A2C | 3.84 | 0.01% | 9s | 6.20 | 1.12% | 48s | 5.20 | 3.17% | 32s | 3.28 | 4.95% | 25s | 4.87 | 6.80% | 23s |
| AM | 3.84 | 0.00% | 9s | 6.18 | 0.78% | 48s | 5.34 | 0.56% | 32s | 3.14 | 0.32% | 25s | 4.63 | 1.52% | 23s |
| POMO | 3.84 | 0.00% | 13s | 6.16 | 0.50% | 1m | 5.09 | 5.29% | 45s | 3.35 | 6.95% | 38s | 4.61 | 1.10% | 42s |
| Sym-NCO | 3.84 | 0.00% | 13s | 6.17 | 0.61% | 1m | 5.35 | 0.39% | 45s | 3.14 | 0.24% | 38s | 4.60 | 0.89% | 42s |
| AM-XL | 3.84 | 0.00% | 13s | 6.16 | 0.44% | 1m | 5.35 | 0.46% | 45s | 3.14 | 0.28% | 38s | 4.60 | 0.87% | 42s |

Table E.2: Running time and memory usage of the AM model trained using FP32 and FP16 mixed precision (FP16-mix), evaluated over 5 epochs with a training size of 10,000 in the CVPR20, CVPR50, and CVPR100.

| Problem | Precision | Running time [s] | Memory usage [GiB] |
|---|---|---|---|
| CVRP20 | FP32 | $6.33 \pm 0.26$ | $1.41 \pm 0.04$ |
| | FP16-mix | $5.89 \pm 0.07$ | $0.84 \pm 0.01$ |
| CVRP50 | FP32 | $13.58 \pm 0.12$ | $4.79 \pm 0.40$ |
| | FP16-mix | $11.68 \pm 0.30$ | $2.30 \pm 0.25$ |
| CVRP100 | FP32 | $35.09 \pm 0.71$ | $13.47 \pm 0.63$ |
| | FP16-mix | $25.11 \pm 0.66$ | $8.14 \pm 0.82$ |

for the time being, although it could be even more impactful due to the auto-regressive nature of our encoder-decoder scheme [18].

---

[18]A recent work, Pagliardini et al. (2023), may be useful in extending FlashAttention to other masking patterns.

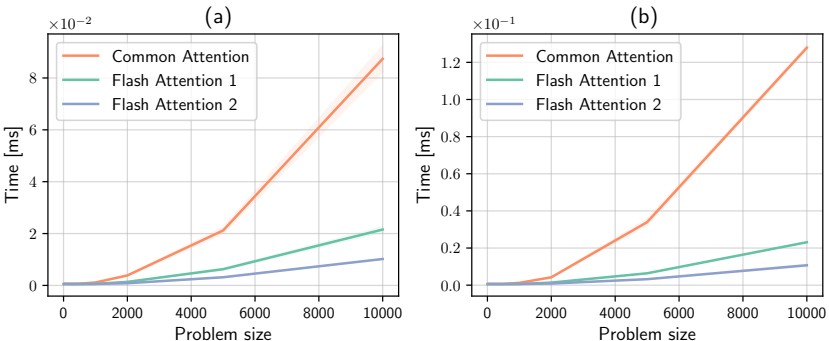

Figure E.3: Running time of the graph attention encoder from the Attention Model, equipped with a standard attention layer, FlashAttention1, and FlashAttention2, across different problem sizes for both (a) the TSP and (b) the CVRP environments.

# F  TSP AND CVRP PUBLIC BENCHMARK

In this section, we evaluate the NCO models trained on randomly generated uniform datasets with 20 and 50 nodes against public benchmark datasets. For the Traveling Salesman Problem (TSP), we evaluate the models using instances from TSPLib (Reinelt, 1991) with fewer than 250 nodes. For the Capacitated Vehicle Routing Problem (CVRP), we evaluate the models using instances from Set A, B, E, F, and M from CVRPLib (Ivan). The optimal or best-known solutions (BKS) of evaluated TSP and CVRP instances are taken from (Reinelt, 1991) and (Ivan). Note that we observed that NCO models with `Augmentation` discovered the solution with a lower cost than the reported BKS for B-n51-k7 of CVRPLib.

**Evaluation results**  Similar to the random instance evaluations, we tested the models with different decoding schemes, including `Greedy`, `Sampling`, `Multistart`, and `Augmentation`. We provide a shortcut link to each combination of results in Table F.1. From the results, we once again confirmed that `Augmentation` generally outperforms the other sampling techniques, with a similar (or smaller) number of sample evaluations and the network forward, similar to the random instance benchmarks.

Table F.1: Table of TSPLib and CVRPLib results

| Dataset | Trained on | Decoding scheme | | | |
|---------|-----------|---------|----------|------------|--------------|
|         |           | Greedy | Sampling | Multistart | Augmentation |
| TSPLib  | 20        | Table F.2 | Table F.3 | Table F.4 | Table F.5 |
|         | 50        | Table F.6 | Table F.7 | Table F.8 | Table F.9 |
| CVRPLib | 20        | Table F.10 | Table F.11 | Table F.12 | Table F.13 |
|         | 50        | Table F.14 | Table F.15 | Table F.16 | Table F.17 |

**Visualized results**  Here we share the tours (i.e., solutions) of the CVRP instances where NCO models tend to have worse and better performances compared to the optimal (or BKS) solutions. We found that NCO solvers have a tendency that NCO solvers generalize quite well to the variation of size with proper scaling on the coordinates of city positions by inspecting the A sets in CVRP. (See Fig. F.3.) However, they often show drastic performance degradation in the instances that are not generated from uniform (or close to) distribution by inspecting the non-A datasets (e.g., the sets B and F) as shown in Figs. F.4 and F.5.

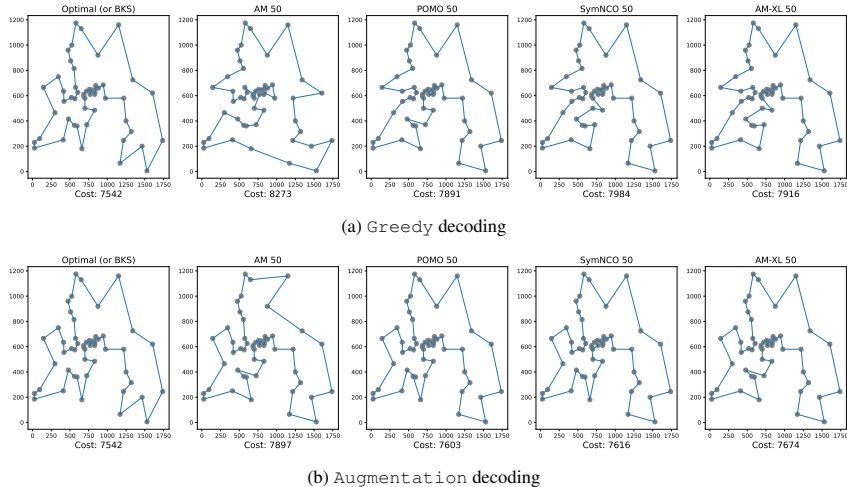

(a) `Greedy` decoding

(b) `Augmentation` decoding

Figure F.1: (TSPLib) Solutions of Berlin52 instance

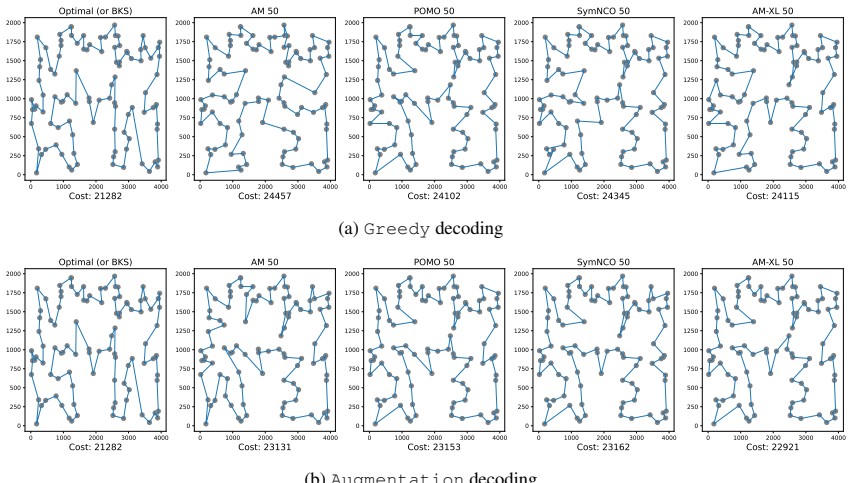

(a) `Greedy` decoding

(b) `Augmentation` decoding

Figure F.2: (TSPLib) Solutions of KroA100 instance

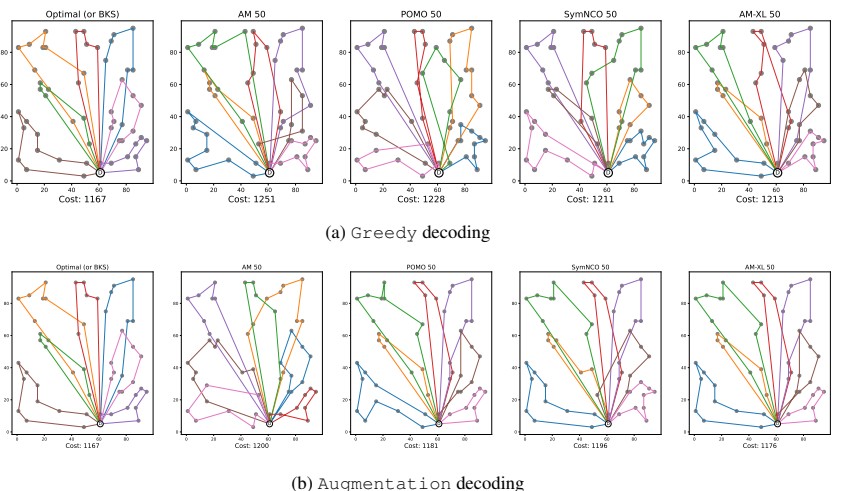

(a) `Greedy` decoding

(b) `Augmentation` decoding

Figure F.3: (CVRPLib) Solutions of A-n54-k7 instance

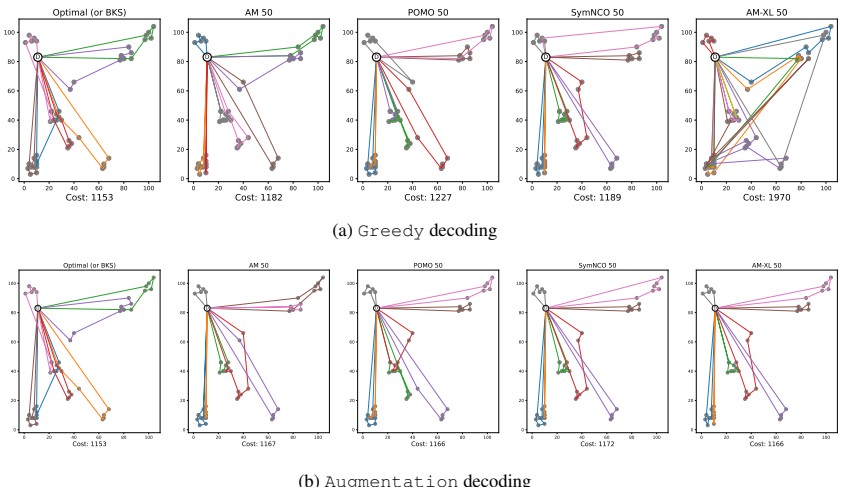

(a) `Greedy` decoding

(b) `Augmentation` decoding

Figure F.4: (CVRPLib) Solutions of B-n57-k7 instance

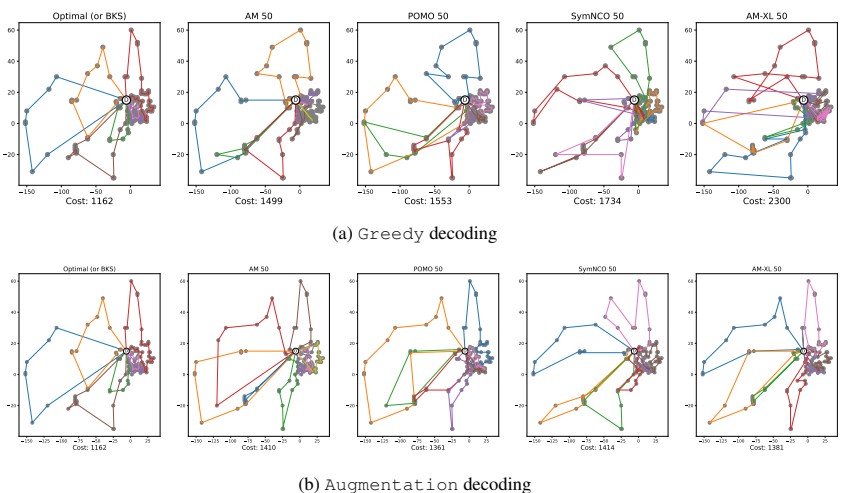

(a) `Greedy` decoding

(b) `Augmentation` decoding

Figure F.5: (CVRPLib) Solutions of F-n135-k7 instance

Table F.2: TSPLib results. The models are trained on TSP20. Greedy decoding is used.

| Instance | Opt. (BKS) | AM | | POMO | | SymNCO | | AM-XL | |
|---|---|---|---|---|---|---|---|---|---|
| | | Cost | Gap ↓ | Cost | Gap ↓ | Cost | Gap ↓ | Cost | Gap ↓ |
| eil51 | 426 | 458 | 6.99 % | 488 | 12.70 % | 462 | 7.79 % | 455 | 6.37 % |
| berlin52 | 7542 | 8623 | 12.54 % | 8757 | 13.87 % | 8518 | 11.46 % | 8621 | 12.52 % |
| st70 | 675 | 718 | 5.99 % | 736 | 8.29 % | 744 | 9.27 % | 733 | 7.91 % |
| eil76 | 538 | 602 | 10.63 % | 606 | 11.22 % | 623 | 13.64 % | 618 | 12.94 % |
| pr76 | 108159 | 116536 | 7.19 % | 123427 | 12.37 % | 119133 | 9.21 % | 115082 | 6.02 % |
| rat99 | 1211 | 1764 | 31.35 % | 1648 | 26.52 % | 1593 | 23.98 % | 1728 | 29.92 % |
| kroA100 | 21282 | 24999 | 14.87 % | 28827 | 26.17 % | 26330 | 19.17 % | 26826 | 20.67 % |
| kroB100 | 22141 | 27325 | 18.97 % | 32980 | 32.87 % | 28256 | 21.64 % | 26209 | 15.52 % |
| kroC100 | 20749 | 24908 | 16.70 % | 25392 | 18.29 % | 26762 | 22.47 % | 26980 | 23.09 % |
| kroD100 | 21294 | 25742 | 17.28 % | 27993 | 23.93 % | 25152 | 15.34 % | 25180 | 15.43 % |
| kroE100 | 22068 | 25985 | 15.07 % | 26754 | 17.52 % | 24751 | 10.84 % | 25494 | 13.44 % |
| rd100 | 7910 | 9324 | 15.17 % | 9011 | 12.22 % | 9121 | 13.28 % | 9096 | 13.04 % |
| eil101 | 629 | 752 | 16.36 % | 769 | 18.21 % | 733 | 14.19 % | 732 | 14.07 % |
| lin105 | 14379 | 17540 | 18.02 % | 17681 | 18.68 % | 17448 | 17.59 % | 17209 | 16.44 % |
| pr124 | 59030 | 63859 | 7.56 % | 68615 | 13.97 % | 65788 | 10.27 % | 68607 | 13.96 % |
| bier127 | 118282 | 141651 | 16.50 % | 167812 | 29.52 % | 145686 | 18.81 % | 139922 | 15.47 % |
| ch130 | 6110 | 6993 | 12.63 % | 8114 | 24.70 % | 7122 | 14.21 % | 7367 | 17.06 % |
| pr136 | 96772 | 116722 | 17.09 % | 117786 | 17.84 % | 114860 | 15.75 % | 112008 | 13.60 % |
| pr144 | 58537 | 64888 | 9.79 % | 65724 | 10.94 % | 66244 | 11.63 % | 66012 | 11.32 % |
| kroA150 | 26524 | 34041 | 22.08 % | 39355 | 32.60 % | 34499 | 23.12 % | 35034 | 24.29 % |
| kroB150 | 26130 | 34394 | 24.03 % | 40379 | 35.29 % | 35975 | 27.37 % | 34700 | 24.70 % |
| pr152 | 73682 | 85034 | 13.35 % | 103086 | 28.52 % | 85237 | 13.56 % | 87594 | 15.88 % |
| u159 | 42080 | 52067 | 19.18 % | 67675 | 37.82 % | 53603 | 21.50 % | 52161 | 19.33 % |
| rat195 | 2323 | 3761 | 38.23 % | 3492 | 33.48 % | 3431 | 32.29 % | 3533 | 34.25 % |
| kroA200 | 29368 | 38338 | 23.40 % | 45189 | 35.01 % | 41007 | 28.38 % | 38696 | 24.11 % |
| ts225 | 126643 | 170904 | 25.90 % | 175644 | 27.90 % | 165864 | 23.65 % | 165968 | 23.69 % |
| tsp225 | 3919 | 5514 | 28.93 % | 5490 | 28.62 % | 5251 | 25.37 % | 5279 | 25.76 % |
| pr226 | 80369 | 96413 | 16.64 % | 117181 | 31.41 % | 94562 | 15.01 % | 88617 | 9.31 % |
| Avg. Gap | 0.00 % | 17.23% | | 22.87% | | 17.53% | | 17.15% | |

Table F.3: TSPLib results. The models are trained on TSP20. Sampling decoding (100 with temperature $\tau = 0.05$) is used.

| Instance | Opt. (BKS) | AM | | POMO | | SymNCO | | AM-XL | |
|---|---|---|---|---|---|---|---|---|---|
| | | Cost | Gap ↓ | Cost | Gap ↓ | Cost | Gap ↓ | Cost | Gap ↓ |
| eil51 | 426 | 452 | 5.75 % | 455 | 6.37 % | 453 | 5.96 % | 452 | 5.75 % |
| berlin52 | 7542 | 8623 | 12.54 % | 8659 | 12.90 % | 8503 | 11.30 % | 8326 | 9.42 % |
| st70 | 675 | 714 | 5.46 % | 715 | 5.59 % | 722 | 6.51 % | 733 | 7.91 % |
| eil76 | 538 | 589 | 8.66 % | 593 | 9.27 % | 594 | 9.43 % | 580 | 7.24 % |
| pr76 | 108159 | 113356 | 4.58 % | 121733 | 11.15 % | 113680 | 4.86 % | 114846 | 5.82 % |
| rat99 | 1211 | 1683 | 28.05 % | 1556 | 22.17 % | 1545 | 21.62 % | 1592 | 23.93 % |
| kroA100 | 21282 | 24614 | 13.54 % | 27074 | 21.39 % | 26549 | 19.84 % | 25947 | 17.98 % |
| kroB100 | 22141 | 25083 | 11.73 % | 28736 | 22.95 % | 26982 | 17.94 % | 24498 | 9.62 % |
| kroC100 | 20749 | 24330 | 14.72 % | 24666 | 15.88 % | 25006 | 17.02 % | 24432 | 15.07 % |
| kroD100 | 21294 | 24307 | 12.40 % | 25464 | 16.38 % | 24590 | 13.40 % | 24602 | 13.45 % |
| kroE100 | 22068 | 25292 | 12.75 % | 26232 | 15.87 % | 24879 | 11.30 % | 24610 | 10.33 % |
| rd100 | 7910 | 9028 | 12.38 % | 8604 | 8.07 % | 8990 | 12.01 % | 8754 | 9.64 % |
| eil101 | 629 | 706 | 10.91 % | 732 | 14.07 % | 715 | 12.03 % | 710 | 11.41 % |
| lin105 | 14379 | 17194 | 16.37 % | 16953 | 15.18 % | 16629 | 13.53 % | 17081 | 15.82 % |
| pr124 | 59030 | 66264 | 10.92 % | 71454 | 17.39 % | 68208 | 13.46 % | 69766 | 15.39 % |
| bier127 | 118282 | 135364 | 12.62 % | 146716 | 19.38 % | 144157 | 17.95 % | 140514 | 15.82 % |
| ch130 | 6110 | 6897 | 11.41 % | 7225 | 15.43 % | 6940 | 11.96 % | 6945 | 12.02 % |
| pr136 | 96772 | 111847 | 13.48 % | 114434 | 15.43 % | 112828 | 14.23 % | 111362 | 13.10 % |
| pr144 | 58537 | 67643 | 13.46 % | 68830 | 14.95 % | 72190 | 18.91 % | 70068 | 16.46 % |
| kroA150 | 26524 | 33358 | 20.49 % | 34358 | 22.80 % | 34281 | 22.63 % | 33881 | 21.71 % |
| kroB150 | 26130 | 31668 | 17.49 % | 35325 | 26.03 % | 34494 | 24.25 % | 33365 | 21.68 % |
| pr152 | 73682 | 86191 | 14.51 % | 89573 | 17.74 % | 89469 | 17.65 % | 88763 | 16.99 % |
| u159 | 42080 | 52987 | 20.58 % | 58999 | 28.68 % | 54263 | 22.45 % | 52393 | 19.68 % |
| rat195 | 2323 | 3558 | 34.71 % | 3472 | 33.09 % | 3329 | 30.22 % | 3557 | 34.69 % |
| kroA200 | 29368 | 37013 | 20.65 % | 43916 | 33.13 % | 41634 | 29.46 % | 39579 | 25.80 % |
| ts225 | 126643 | 175030 | 27.64 % | 179713 | 29.53 % | 182349 | 30.55 % | 176018 | 28.05 % |
| tsp225 | 3919 | 5390 | 27.29 % | 5686 | 31.08 % | 5510 | 28.87 % | 5389 | 27.28 % |
| pr226 | 80369 | 100274 | 19.85 % | 108547 | 25.96 % | 106235 | 24.35 % | 105694 | 23.96 % |
| Avg. Gap | 0.00 % | 15.53% | | 18.85% | | 17.27% | | 16.29% | |

Table F.4: TSPLib results. The models are trained on TSP20. Greedy multi-start decoding is used.

| Instance | Opt. (BKS) | AM | | POMO | | SymNCO | | AM-XL | |
|---|---|---|---|---|---|---|---|---|---|
| | | Cost | Gap ↓ | Cost | Gap ↓ | Cost | Gap ↓ | Cost | Gap ↓ |
| eil51 | 426 | 458 | 6.99 % | 449 | 5.12 % | 453 | 5.96 % | 454 | 6.17 % |
| berlin52 | 7542 | 8623 | 12.54 % | 8075 | 6.60 % | 8251 | 8.59 % | 8419 | 10.42 % |
| st70 | 675 | 713 | 5.33 % | 711 | 5.06 % | 735 | 8.16 % | 716 | 5.73 % |
| eil76 | 538 | 602 | 10.63 % | 606 | 11.22 % | 601 | 10.48 % | 600 | 10.33 % |
| pr76 | 108159 | 113356 | 4.58 % | 113227 | 4.48 % | 118432 | 8.67 % | 113647 | 4.83 % |
| rat99 | 1211 | 1680 | 27.92 % | 1556 | 22.17 % | 1559 | 22.32 % | 1647 | 26.47 % |
| kroA100 | 21282 | 23771 | 10.47 % | 26118 | 18.52 % | 25840 | 17.64 % | 26826 | 20.67 % |
| kroB100 | 22141 | 25721 | 13.92 % | 27440 | 19.31 % | 26081 | 15.11 % | 25275 | 12.40 % |
| kroC100 | 20749 | 24377 | 14.88 % | 23883 | 13.12 % | 25633 | 19.05 % | 24224 | 14.35 % |
| kroD100 | 21294 | 25429 | 16.26 % | 24914 | 14.53 % | 23878 | 10.82 % | 25047 | 14.98 % |
| kroE100 | 22068 | 25354 | 12.96 % | 25843 | 14.61 % | 24751 | 10.84 % | 24857 | 11.22 % |
| rd100 | 7910 | 9035 | 12.45 % | 8708 | 9.16 % | 9088 | 12.96 % | 9062 | 12.71 % |
| eil101 | 629 | 740 | 15.00 % | 731 | 13.95 % | 720 | 12.64 % | 726 | 13.36 % |
| lin105 | 14379 | 17466 | 17.67 % | 17149 | 16.15 % | 16968 | 15.26 % | 16616 | 13.46 % |
| pr124 | 59030 | 63859 | 7.56 % | 64806 | 8.91 % | 64998 | 9.18 % | 67139 | 12.08 % |
| bier127 | 118282 | 140462 | 15.79 % | 141116 | 16.18 % | 142547 | 17.02 % | 139922 | 15.47 % |
| ch130 | 6110 | 6993 | 12.63 % | 6936 | 11.91 % | 6960 | 12.21 % | 6857 | 10.89 % |
| pr136 | 96772 | 115122 | 15.94 % | 112504 | 13.98 % | 113532 | 14.76 % | 110596 | 12.50 % |
| pr144 | 58537 | 63789 | 8.23 % | 65724 | 10.94 % | 66029 | 11.35 % | 66012 | 11.32 % |
| kroA150 | 26524 | 33285 | 20.31 % | 33897 | 21.75 % | 33870 | 21.69 % | 34631 | 23.41 % |
| kroB150 | 26130 | 32483 | 19.56 % | 33308 | 21.55 % | 32578 | 19.79 % | 33184 | 21.26 % |
| pr152 | 73682 | 84881 | 13.19 % | 84819 | 13.13 % | 85237 | 13.56 % | 86964 | 15.27 % |
| u159 | 42080 | 50776 | 17.13 % | 54142 | 22.28 % | 52220 | 19.42 % | 51686 | 18.59 % |
| rat195 | 2323 | 3568 | 34.89 % | 3258 | 28.70 % | 3341 | 30.47 % | 3533 | 34.25 % |
| kroA200 | 29368 | 37345 | 21.36 % | 38861 | 24.43 % | 38629 | 23.97 % | 37628 | 21.95 % |
| ts225 | 126643 | 167811 | 24.53 % | 159519 | 20.61 % | 155767 | 18.70 % | 156938 | 19.30 % |
| tsp225 | 3919 | 5389 | 27.28 % | 5392 | 27.32 % | 5134 | 23.67 % | 5184 | 24.40 % |
| pr226 | 80369 | 91949 | 12.59 % | 92146 | 12.78 % | 92011 | 12.65 % | 88476 | 9.16 % |
| Avg. Gap | 0.00 % | 15.45% | | 15.30% | | 15.25% | | 15.25% | |

Table F.5: TSPLib results. The models are trained on TSP20. Augmentation decoding (100) is used.

| Instance | Opt. (BKS) | AM | | POMO | | SymNCO | | AM-XL | |
|---|---|---|---|---|---|---|---|---|---|
| | | Cost | Gap ↓ | Cost | Gap ↓ | Cost | Gap ↓ | Cost | Gap ↓ |
| eil51 | 426 | 445 | 4.27 % | 443 | 3.84 % | 438 | 2.74 % | 441 | 3.40 % |
| berlin52 | 7542 | 7874 | 4.22 % | 8253 | 8.62 % | 7805 | 3.37 % | 7914 | 4.70 % |
| st70 | 675 | 696 | 3.02 % | 701 | 3.71 % | 704 | 4.12 % | 696 | 3.02 % |
| eil76 | 538 | 582 | 7.56 % | 598 | 10.03 % | 574 | 6.27 % | 571 | 5.78 % |
| pr76 | 108159 | 110726 | 2.32 % | 111534 | 3.03 % | 111035 | 2.59 % | 111416 | 2.92 % |
| rat99 | 1211 | 1426 | 15.08 % | 1526 | 20.64 % | 1533 | 21.00 % | 1493 | 18.89 % |
| kroA100 | 21282 | 24327 | 12.52 % | 26154 | 18.63 % | 23784 | 10.52 % | 24798 | 14.18 % |
| kroB100 | 22141 | 25582 | 13.45 % | 26913 | 17.73 % | 25068 | 11.68 % | 24378 | 9.18 % |
| kroC100 | 20749 | 23991 | 13.51 % | 22781 | 8.92 % | 23780 | 12.75 % | 23712 | 12.50 % |
| kroD100 | 21294 | 24289 | 12.33 % | 25099 | 15.16 % | 23815 | 10.59 % | 24052 | 11.47 % |
| kroE100 | 22068 | 24889 | 11.33 % | 25946 | 14.95 % | 24373 | 9.46 % | 24728 | 10.76 % |
| rd100 | 7910 | 8738 | 9.48 % | 8840 | 10.52 % | 8674 | 8.81 % | 8419 | 6.05 % |
| eil101 | 629 | 701 | 10.27 % | 711 | 11.53 % | 706 | 10.91 % | 696 | 9.63 % |
| lin105 | 14379 | 16670 | 13.74 % | 16956 | 15.20 % | 16297 | 11.77 % | 16656 | 13.67 % |
| pr124 | 59030 | 63859 | 7.56 % | 66313 | 10.98 % | 62469 | 5.51 % | 62260 | 5.19 % |
| bier127 | 118282 | 134016 | 11.74 % | 149337 | 20.80 % | 135108 | 12.45 % | 136150 | 13.12 % |
| ch130 | 6110 | 6816 | 10.36 % | 6995 | 12.65 % | 6732 | 9.24 % | 6791 | 10.03 % |
| pr136 | 96772 | 113441 | 14.69 % | 112384 | 13.89 % | 110893 | 12.73 % | 109425 | 11.56 % |
| pr144 | 58537 | 63032 | 7.13 % | 63877 | 8.36 % | 62506 | 6.35 % | 63297 | 7.52 % |
| kroA150 | 26524 | 32533 | 18.47 % | 33432 | 20.66 % | 32734 | 18.97 % | 31569 | 15.98 % |
| kroB150 | 26130 | 31116 | 16.02 % | 32583 | 19.80 % | 31105 | 15.99 % | 31246 | 16.37 % |
| pr152 | 73682 | 81811 | 9.94 % | 81998 | 10.14 % | 81797 | 9.92 % | 81166 | 9.22 % |
| u159 | 42080 | 51050 | 17.57 % | 52328 | 19.58 % | 50831 | 17.22 % | 50282 | 16.31 % |
| rat195 | 2323 | 3231 | 28.10 % | 3273 | 29.03 % | 3238 | 28.26 % | 3250 | 28.52 % |
| kroA200 | 29368 | 36065 | 18.57 % | 38996 | 24.69 % | 36201 | 18.88 % | 37249 | 21.16 % |
| ts225 | 126643 | 160088 | 20.89 % | 161870 | 21.76 % | 152170 | 16.78 % | 153857 | 17.69 % |
| tsp225 | 3919 | 5273 | 25.68 % | 5247 | 25.31 % | 5169 | 24.18 % | 5150 | 23.90 % |
| pr226 | 80369 | 87684 | 8.34 % | 90981 | 11.66 % | 87330 | 7.97 % | 85462 | 5.96 % |
| Avg. Gap | 0.00 % | 12.43% | | 14.71% | | 11.82% | | 11.74% | |

Table F.6: TSPLib results. The models are trained on TSP50. Greedy decoding is used.

| Instance | Opt. (BKS) | AM | | POMO | | SymNCO | | AM-XL | |
|---|---|---|---|---|---|---|---|---|---|
| | | Cost | Gap ↓ | Cost | Gap ↓ | Cost | Gap ↓ | Cost | Gap ↓ |
| eil51 | 426 | 440 | 3.18 % | 436 | 2.29 % | 434 | 1.84 % | 436 | 2.29 % |
| berlin52 | 7542 | 8273 | 8.84 % | 7891 | 4.42 % | 7984 | 5.54 % | 7916 | 4.72 % |
| st70 | 675 | 694 | 2.74 % | 700 | 3.57 % | 696 | 3.02 % | 686 | 1.60 % |
| eil76 | 538 | 580 | 7.24 % | 562 | 4.27 % | 572 | 5.94 % | 553 | 2.71 % |
| pr76 | 108159 | 110798 | 2.38 % | 113731 | 4.90 % | 111953 | 3.39 % | 111360 | 2.87 % |
| rat99 | 1211 | 1482 | 18.29 % | 1503 | 19.43 % | 1442 | 16.02 % | 1456 | 16.83 % |
| kroA100 | 21282 | 24457 | 12.98 % | 24102 | 11.70 % | 24345 | 12.58 % | 24115 | 11.75 % |
| kroB100 | 22141 | 26447 | 16.28 % | 24086 | 8.08 % | 25146 | 11.95 % | 24607 | 10.02 % |
| kroC100 | 20749 | 24211 | 14.30 % | 23334 | 11.08 % | 22725 | 8.70 % | 23362 | 11.18 % |
| kroD100 | 21294 | 23117 | 7.89 % | 24180 | 11.94 % | 23326 | 8.71 % | 23751 | 10.34 % |
| kroE100 | 22068 | 24476 | 9.84 % | 28393 | 22.28 % | 23933 | 7.79 % | 24865 | 11.25 % |
| rd100 | 7910 | 8163 | 3.10 % | 8139 | 2.81 % | 8072 | 2.01 % | 8082 | 2.13 % |
| eil101 | 629 | 678 | 7.23 % | 681 | 7.64 % | 691 | 8.97 % | 679 | 7.36 % |
| lin105 | 14379 | 15590 | 7.77 % | 16418 | 12.42 % | 16545 | 13.09 % | 16029 | 10.29 % |
| pr124 | 59030 | 61155 | 3.47 % | 61062 | 3.33 % | 59809 | 1.30 % | 59332 | 0.51 % |
| bier127 | 118282 | 130236 | 9.18 % | 154102 | 23.24 % | 137533 | 14.00 % | 133332 | 11.29 % |
| ch130 | 6110 | 6440 | 5.12 % | 6441 | 5.14 % | 6308 | 3.14 % | 6320 | 3.32 % |
| pr136 | 96772 | 104110 | 7.05 % | 104135 | 7.07 % | 103969 | 6.92 % | 102428 | 5.52 % |
| pr144 | 58537 | 63959 | 8.48 % | 63372 | 7.63 % | 60421 | 3.12 % | 61613 | 4.99 % |
| kroA150 | 26524 | 30287 | 12.42 % | 30933 | 14.25 % | 30614 | 13.36 % | 30685 | 13.56 % |
| kroB150 | 26130 | 30565 | 14.51 % | 30997 | 15.70 % | 29083 | 10.15 % | 29528 | 11.51 % |
| pr152 | 73682 | 84632 | 12.94 % | 79409 | 7.21 % | 78854 | 6.56 % | 78362 | 5.97 % |
| u159 | 42080 | 46792 | 10.07 % | 46249 | 9.01 % | 46818 | 10.12 % | 44975 | 6.44 % |
| rat195 | 2323 | 3182 | 27.00 % | 3357 | 30.80 % | 3149 | 26.23 % | 3403 | 31.74 % |
| kroA200 | 29368 | 35068 | 16.25 % | 37667 | 22.03 % | 35349 | 16.92 % | 34728 | 15.43 % |
| ts225 | 126643 | 147273 | 14.01 % | 145364 | 12.88 % | 139645 | 9.31 % | 138527 | 8.58 % |
| tsp225 | 3919 | 4836 | 18.96 % | 5196 | 24.58 % | 4813 | 18.57 % | 5103 | 23.20 % |
| pr226 | 80369 | 86665 | 7.26 % | 87200 | 7.83 % | 85375 | 5.86 % | 86397 | 6.98 % |
| Avg. Gap | 0.00 % | | 10.31% | | 11.34% | | 9.11% | | 9.09% |

Table F.7: TSPLib results. The models are trained on TSP50. Sampling decoding (100 with temperature $\tau = 0.05$) is used.

| Instance | Opt. (BKS) | AM | | POMO | | SymNCO | | AM-XL | |
|---|---|---|---|---|---|---|---|---|---|
| | | Cost | Gap $\downarrow$ | Cost | Gap $\downarrow$ | Cost | Gap $\downarrow$ | Cost | Gap $\downarrow$ |
| eil51 | 426 | 440 | 3.18 % | 436 | 2.29 % | 433 | 1.62 % | 436 | 2.29 % |
| berlin52 | 7542 | 8258 | 8.67 % | 7891 | 4.42 % | 7984 | 5.54 % | 7916 | 4.72 % |
| st70 | 675 | 694 | 2.74 % | 699 | 3.43 % | 695 | 2.88 % | 686 | 1.60 % |
| eil76 | 538 | 572 | 5.94 % | 562 | 4.27 % | 570 | 5.61 % | 553 | 2.71 % |
| pr76 | 108159 | 110798 | 2.38 % | 113034 | 4.31 % | 111953 | 3.39 % | 111360 | 2.87 % |
| rat99 | 1211 | 1476 | 17.95 % | 1461 | 17.11 % | 1442 | 16.02 % | 1423 | 14.90 % |
| kroA100 | 21282 | 23863 | 10.82 % | 23674 | 10.10 % | 24230 | 12.17 % | 24115 | 11.75 % |
| kroB100 | 22141 | 24852 | 10.91 % | 24086 | 8.08 % | 24816 | 10.78 % | 24472 | 9.53 % |
| kroC100 | 20749 | 23350 | 11.14 % | 23250 | 10.76 % | 22725 | 8.70 % | 23358 | 11.17 % |
| kroD100 | 21294 | 23117 | 7.89 % | 24103 | 11.65 % | 23326 | 8.71 % | 23720 | 10.23 % |
| kroE100 | 22068 | 24464 | 9.79 % | 24321 | 9.26 % | 23718 | 6.96 % | 24561 | 10.15 % |
| rd100 | 7910 | 8092 | 2.25 % | 8070 | 1.98 % | 8068 | 1.96 % | 8014 | 1.30 % |
| eil101 | 629 | 677 | 7.09 % | 677 | 7.09 % | 684 | 8.04 % | 679 | 7.36 % |
| lin105 | 14379 | 15559 | 7.58 % | 16369 | 12.16 % | 16073 | 10.54 % | 15520 | 7.35 % |
| pr124 | 59030 | 61017 | 3.26 % | 60711 | 2.77 % | 59809 | 1.30 % | 59332 | 0.51 % |
| bier127 | 118282 | 127519 | 7.24 % | 145576 | 18.75 % | 136891 | 13.59 % | 127554 | 7.27 % |
| ch130 | 6110 | 6354 | 3.84 % | 6399 | 4.52 % | 6291 | 2.88 % | 6308 | 3.14 % |
| pr136 | 96772 | 103066 | 6.11 % | 103024 | 6.07 % | 103054 | 6.10 % | 101760 | 4.90 % |
| pr144 | 58537 | 60827 | 3.76 % | 61126 | 4.24 % | 60040 | 2.50 % | 60694 | 3.55 % |
| kroA150 | 26524 | 30015 | 11.63 % | 30664 | 13.50 % | 30510 | 13.06 % | 30355 | 12.62 % |
| kroB150 | 26130 | 29521 | 11.49 % | 29313 | 10.86 % | 28883 | 9.53 % | 29029 | 9.99 % |
| pr152 | 73682 | 80769 | 8.77 % | 78405 | 6.02 % | 77298 | 4.68 % | 77052 | 4.37 % |
| u159 | 42080 | 45508 | 7.53 % | 45598 | 7.72 % | 46084 | 8.69 % | 44547 | 5.54 % |
| rat195 | 2323 | 3051 | 23.86 % | 3120 | 25.54 % | 3060 | 24.08 % | 3098 | 25.02 % |
| kroA200 | 29368 | 34515 | 14.91 % | 35815 | 18.00 % | 33866 | 13.28 % | 34432 | 14.71 % |
| ts225 | 126643 | 141706 | 10.63 % | 142392 | 11.06 % | 139255 | 9.06 % | 138130 | 8.32 % |
| tsp225 | 3919 | 4726 | 17.08 % | 4935 | 20.59 % | 4560 | 14.06 % | 4644 | 15.61 % |
| pr226 | 80369 | 85410 | 5.90 % | 85033 | 5.48 % | 85232 | 5.71 % | 85741 | 6.27 % |
| Avg. Gap | 0.00 % | | 8.73% | | 9.36% | | 8.27% | | 7.85% |

Table F.8: TSPLib results. The models are trained on TSP50. Greedy multi-start decoding is used.

| Instance | Opt. (BKS) | AM Cost | AM Gap ↓ | POMO Cost | POMO Gap ↓ | SymNCO Cost | SymNCO Gap ↓ | AM-XL Cost | AM-XL Gap ↓ |
|---|---|---|---|---|---|---|---|---|---|
| eil51 | 426 | 440 | 3.18 % | 431 | 1.16 % | 430 | 0.93 % | 436 | 2.29 % |
| berlin52 | 7542 | 8270 | 8.80 % | 7679 | 1.78 % | 7984 | 5.54 % | 7731 | 2.44 % |
| st70 | 675 | 693 | 2.60 % | 693 | 2.60 % | 696 | 3.02 % | 686 | 1.60 % |
| eil76 | 538 | 580 | 7.24 % | 559 | 3.76 % | 563 | 4.44 % | 553 | 2.71 % |
| pr76 | 108159 | 110601 | 2.21 % | 111732 | 3.20 % | 111953 | 3.39 % | 111360 | 2.87 % |
| rat99 | 1211 | 1475 | 17.90 % | 1422 | 14.84 % | 1442 | 16.02 % | 1417 | 14.54 % |
| kroA100 | 21282 | 23993 | 11.30 % | 23241 | 8.43 % | 23817 | 10.64 % | 24102 | 11.70 % |
| kroB100 | 22141 | 25097 | 11.78 % | 24071 | 8.02 % | 24026 | 7.85 % | 24607 | 10.02 % |
| kroC100 | 20749 | 23354 | 11.15 % | 22539 | 7.94 % | 22725 | 8.70 % | 22943 | 9.56 % |
| kroD100 | 21294 | 23117 | 7.89 % | 23025 | 7.52 % | 22731 | 6.32 % | 23136 | 7.96 % |
| kroE100 | 22068 | 24476 | 9.84 % | 23746 | 7.07 % | 23908 | 7.70 % | 23834 | 7.41 % |
| rd100 | 7910 | 8159 | 3.05 % | 8047 | 1.70 % | 8072 | 2.01 % | 8021 | 1.38 % |
| eil101 | 629 | 666 | 5.56 % | 661 | 4.84 % | 671 | 6.26 % | 661 | 4.84 % |
| lin105 | 14379 | 15509 | 7.29 % | 15619 | 7.94 % | 16404 | 12.34 % | 15570 | 7.65 % |
| pr124 | 59030 | 61134 | 3.44 % | 59365 | 0.56 % | 59809 | 1.30 % | 59332 | 0.51 % |
| bier127 | 118282 | 129821 | 8.89 % | 129934 | 8.97 % | 137100 | 13.73 % | 128116 | 7.68 % |
| ch130 | 6110 | 6368 | 4.05 % | 6315 | 3.25 % | 6308 | 3.14 % | 6314 | 3.23 % |
| pr136 | 96772 | 102727 | 5.80 % | 100055 | 3.28 % | 102949 | 6.00 % | 100513 | 3.72 % |
| pr144 | 58537 | 61943 | 5.50 % | 60386 | 3.06 % | 60421 | 3.12 % | 61131 | 4.24 % |
| kroA150 | 26524 | 30112 | 11.92 % | 29083 | 8.80 % | 30478 | 12.97 % | 30438 | 12.86 % |
| kroB150 | 26130 | 29673 | 11.94 % | 29123 | 10.28 % | 28947 | 9.73 % | 28912 | 9.62 % |
| pr152 | 73682 | 81953 | 10.09 % | 76996 | 4.30 % | 78300 | 5.90 % | 78214 | 5.79 % |
| u159 | 42080 | 46594 | 9.69 % | 44452 | 5.34 % | 46503 | 9.51 % | 44917 | 6.32 % |
| rat195 | 2323 | 3095 | 24.94 % | 3075 | 24.46 % | 3088 | 24.77 % | 3100 | 25.06 % |
| kroA200 | 29368 | 34825 | 15.67 % | 34971 | 16.02 % | 34050 | 13.75 % | 34422 | 14.68 % |
| ts225 | 126643 | 144315 | 12.25 % | 137942 | 8.19 % | 139548 | 9.25 % | 138438 | 8.52 % |
| tsp225 | 3919 | 4749 | 17.48 % | 4580 | 14.43 % | 4709 | 16.78 % | 4908 | 20.15 % |
| pr226 | 80369 | 86582 | 7.18 % | 83980 | 4.30 % | 85375 | 5.86 % | 86237 | 6.80 % |
| Avg. Gap | 0.00 % | | 9.24% | | 7.00% | | 8.25% | | 7.72% |

Table F.9: TSPLib results. The models are trained on TSP50. Augmentation decoding (100) is used.

| Instance | Opt. (BKS) | AM Cost | AM Gap ↓ | POMO Cost | POMO Gap ↓ | SymNCO Cost | SymNCO Gap ↓ | AM-XL Cost | AM-XL Gap ↓ |
|---|---|---|---|---|---|---|---|---|---|
| eil51 | 426 | 431 | 1.16 % | 432 | 1.39 % | 429 | 0.70 % | 429 | 0.70 % |
| berlin52 | 7542 | 7897 | 4.50 % | 7722 | 2.33 % | 7576 | 0.45 % | 7674 | 1.72 % |
| st70 | 675 | 678 | 0.44 % | 680 | 0.74 % | 678 | 0.44 % | 678 | 0.44 % |
| eil76 | 538 | 557 | 3.41 % | 559 | 3.76 % | 552 | 2.54 % | 551 | 2.36 % |
| pr76 | 108159 | 110215 | 1.87 % | 109523 | 1.25 % | 109920 | 1.60 % | 109775 | 1.47 % |
| rat99 | 1211 | 1435 | 15.61 % | 1436 | 15.67 % | 1412 | 14.24 % | 1424 | 14.96 % |
| kroA100 | 21282 | 23253 | 8.48 % | 23125 | 7.97 % | 23056 | 7.69 % | 22915 | 7.13 % |
| kroB100 | 22141 | 23987 | 7.70 % | 23840 | 7.13 % | 23583 | 6.11 % | 23381 | 5.30 % |
| kroC100 | 20749 | 22041 | 5.86 % | 22458 | 7.61 % | 22504 | 7.80 % | 21706 | 4.41 % |
| kroD100 | 21294 | 22826 | 6.71 % | 22540 | 5.53 % | 22865 | 6.87 % | 22880 | 6.93 % |
| kroE100 | 22068 | 23436 | 5.84 % | 23623 | 6.58 % | 23396 | 5.68 % | 23257 | 5.11 % |
| rd100 | 7910 | 7935 | 0.32 % | 8006 | 1.20 % | 7945 | 0.44 % | 7972 | 0.78 % |
| eil101 | 629 | 662 | 4.98 % | 665 | 5.41 % | 655 | 3.97 % | 661 | 4.84 % |
| lin105 | 14379 | 15386 | 6.54 % | 15638 | 8.05 % | 15270 | 5.83 % | 15228 | 5.58 % |
| pr124 | 59030 | 60586 | 2.57 % | 60150 | 1.86 % | 59565 | 0.90 % | 59332 | 0.51 % |
| bier127 | 118282 | 124017 | 4.62 % | 129382 | 8.58 % | 127516 | 7.24 % | 125509 | 5.76 % |
| ch130 | 6110 | 6248 | 2.21 % | 6351 | 3.79 % | 6217 | 1.72 % | 6239 | 2.07 % |
| pr136 | 96772 | 100325 | 3.54 % | 99948 | 3.18 % | 99458 | 2.70 % | 98595 | 1.85 % |
| pr144 | 58537 | 60478 | 3.21 % | 59754 | 2.04 % | 59202 | 1.12 % | 59255 | 1.21 % |
| kroA150 | 26524 | 29298 | 9.47 % | 29127 | 8.94 % | 29587 | 10.35 % | 29278 | 9.41 % |
| kroB150 | 26130 | 29116 | 10.26 % | 28840 | 9.40 % | 28469 | 8.22 % | 28498 | 8.31 % |
| pr152 | 73682 | 76378 | 3.53 % | 76074 | 3.14 % | 75825 | 2.83 % | 75495 | 2.40 % |
| u159 | 42080 | 44124 | 4.63 % | 43795 | 3.92 % | 43765 | 3.85 % | 43846 | 4.03 % |
| rat195 | 2323 | 3040 | 23.59 % | 3060 | 24.08 % | 3043 | 23.66 % | 2976 | 21.94 % |
| kroA200 | 29368 | 33751 | 12.99 % | 33743 | 12.97 % | 32690 | 10.16 % | 33660 | 12.75 % |
| ts225 | 126643 | 140185 | 9.66 % | 139927 | 9.49 % | 139024 | 8.91 % | 138401 | 8.50 % |
| tsp225 | 3919 | 4633 | 15.41 % | 4624 | 15.25 % | 4528 | 13.45 % | 4622 | 15.21 % |
| pr226 | 80369 | 83220 | 3.43 % | 83512 | 3.76 % | 83164 | 3.36 % | 83479 | 3.73 % |
| Avg. Gap | 0.00 % | | 6.52% | | 6.61% | | 5.82% | | 5.69% |

Table F.10: CVRPLib results. The models are trained on CVRP20. Greedy decoding is used.

| Instance | Opt. (BKS) | AM | | POMO | | SymNCO | | AM-XL | |
|---|---|---|---|---|---|---|---|---|---|
| | | Cost | Gap ↓ | Cost | Gap ↓ | Cost | Gap ↓ | Cost | Gap ↓ |
| A-n53-k7 | 1010 | 1180 | 14.41 % | 1138 | 11.25 % | 1138 | 11.25 % | 1115 | 9.42 % |
| A-n54-k7 | 1167 | 1272 | 8.25 % | 1351 | 13.62 % | 1300 | 10.23 % | 1303 | 10.44 % |
| A-n55-k9 | 1073 | 1283 | 16.37 % | 1252 | 14.30 % | 1196 | 10.28 % | 1226 | 12.48 % |
| A-n60-k9 | 1354 | 1430 | 5.31 % | 1511 | 10.39 % | 1522 | 11.04 % | 1583 | 14.47 % |
| A-n61-k9 | 1034 | 1186 | 12.82 % | 1222 | 15.38 % | 1201 | 13.91 % | 1244 | 16.88 % |
| A-n62-k8 | 1288 | 1464 | 12.02 % | 1422 | 9.42 % | 1393 | 7.54 % | 1386 | 7.07 % |
| A-n63-k9 | 1616 | 1697 | 4.77 % | 1824 | 11.40 % | 1864 | 13.30 % | 1815 | 10.96 % |
| A-n63-k10 | 1314 | 1371 | 4.16 % | 1577 | 16.68 % | 1536 | 14.45 % | 1549 | 15.17 % |
| A-n64-k9 | 1401 | 1545 | 9.32 % | 1642 | 14.68 % | 1559 | 10.13 % | 1490 | 5.97 % |
| A-n65-k9 | 1174 | 1358 | 13.55 % | 1287 | 8.78 % | 1408 | 16.62 % | 1322 | 11.20 % |
| A-n69-k9 | 1159 | 1345 | 13.83 % | 1360 | 14.78 % | 1319 | 12.13 % | 1334 | 13.12 % |
| A-n80-k10 | 1763 | 2017 | 12.59 % | 2114 | 16.60 % | 2051 | 14.04 % | 2004 | 12.03 % |
| B-n51-k7 | 1032 | 1049 | 1.62 % | 1065 | 3.10 % | 1129 | 8.59 % | 1054 | 2.09 % |
| B-n52-k7 | 747 | 793 | 5.80 % | 893 | 16.35 % | 929 | 19.59 % | 820 | 8.90 % |
| B-n56-k7 | 707 | 789 | 10.39 % | 815 | 13.25 % | 845 | 16.33 % | 905 | 21.88 % |
| B-n57-k7 | 1153 | 1375 | 16.15 % | 1401 | 17.70 % | 1435 | 19.65 % | 1340 | 13.96 % |
| B-n57-k9 | 1598 | 1768 | 9.62 % | 1746 | 8.48 % | 1752 | 8.79 % | 1719 | 7.04 % |
| B-n63-k10 | 1496 | 1629 | 8.16 % | 1627 | 8.05 % | 1712 | 12.62 % | 1696 | 11.79 % |
| B-n64-k9 | 861 | 950 | 9.37 % | 997 | 13.64 % | 1065 | 19.15 % | 1098 | 21.58 % |
| B-n66-k9 | 1316 | 1429 | 7.91 % | 1525 | 13.70 % | 1452 | 9.37 % | 1399 | 5.93 % |
| B-n67-k10 | 1032 | 1163 | 11.26 % | 1151 | 10.34 % | 1237 | 16.57 % | 1183 | 12.76 % |
| B-n68-k9 | 1272 | 1463 | 13.06 % | 1498 | 15.09 % | 1444 | 11.91 % | 1476 | 13.82 % |
| B-n78-k10 | 1221 | 1376 | 11.26 % | 1473 | 17.11 % | 1455 | 16.08 % | 1450 | 15.79 % |
| E-n51-k5 | 521 | 566 | 7.95 % | 629 | 17.17 % | 621 | 16.10 % | 582 | 10.48 % |
| E-n76-k7 | 682 | 840 | 18.81 % | 808 | 15.59 % | 847 | 19.48 % | 860 | 20.70 % |
| E-n76-k8 | 735 | 861 | 14.63 % | 840 | 12.50 % | 884 | 16.86 % | 904 | 18.69 % |
| E-n76-k10 | 830 | 998 | 16.83 % | 957 | 13.27 % | 986 | 15.82 % | 1047 | 20.73 % |
| E-n76-k14 | 1021 | 1151 | 11.29 % | 1232 | 17.13 % | 1216 | 16.04 % | 1184 | 13.77 % |
| E-n101-k8 | 815 | 1113 | 26.77 % | 1051 | 22.45 % | 1183 | 31.11 % | 1101 | 25.98 % |
| E-n101-k14 | 1067 | 1222 | 12.68 % | 1306 | 18.30 % | 1413 | 24.49 % | 1317 | 18.98 % |
| F-n72-k4 | 237 | 290 | 18.28 % | 291 | 18.56 % | 404 | 41.34 % | 344 | 31.10 % |
| F-n135-k7 | 1162 | 1998 | 41.84 % | 2148 | 45.90 % | 2425 | 52.08 % | 2037 | 42.96 % |
| M-n101-k10 | 820 | 1302 | 37.02 % | 1142 | 28.20 % | 1214 | 32.45 % | 1137 | 27.88 % |
| M-n121-k7 | 1034 | 1417 | 27.03 % | 1423 | 27.34 % | 2379 | 56.54 % | 1617 | 36.05 % |
| M-n151-k12 | 1015 | 1284 | 20.95 % | 1374 | 26.13 % | 2013 | 49.58 % | 1582 | 35.84 % |
| M-n200-k16 | 1274 | 1845 | 30.95 % | 1790 | 28.83 % | 2955 | 56.89 % | 1988 | 35.92 % |
| M-n200-k17 | 1275 | 1845 | 30.89 % | 1790 | 28.77 % | 2955 | 56.85 % | 1988 | 35.87 % |
| Avg. Gap | 0.00 % | | 14.81% | | 16.60% | | 21.33% | | 17.56% |

Table F.11: CVRPLib results. The models are trained on CVRP20. Sampling decoding (100 with temperature $\tau = 0.05$) is used.

| Instance | Opt. (BKS) | AM Cost | AM Gap ↓ | POMO Cost | POMO Gap ↓ | SymNCO Cost | SymNCO Gap ↓ | AM-XL Cost | AM-XL Gap ↓ |
|---|---|---|---|---|---|---|---|---|---|
| A-n53-k7 | 1010 | 1171 | 13.75 % | 1073 | 5.87 % | 1138 | 11.25 % | 1096 | 7.85 % |
| A-n54-k7 | 1167 | 1257 | 7.16 % | 1257 | 7.16 % | 1299 | 10.16 % | 1271 | 8.18 % |
| A-n55-k9 | 1073 | 1268 | 15.38 % | 1191 | 9.91 % | 1174 | 8.60 % | 1224 | 12.34 % |
| A-n60-k9 | 1354 | 1430 | 5.31 % | 1484 | 8.76 % | 1522 | 11.04 % | 1498 | 9.61 % |
| A-n61-k9 | 1034 | 1112 | 7.01 % | 1183 | 12.60 % | 1136 | 8.98 % | 1202 | 13.98 % |
| A-n62-k8 | 1288 | 1418 | 9.17 % | 1417 | 9.10 % | 1369 | 5.92 % | 1381 | 6.73 % |
| A-n63-k9 | 1616 | 1697 | 4.77 % | 1807 | 10.57 % | 1844 | 12.36 % | 1741 | 7.18 % |
| A-n63-k10 | 1314 | 1366 | 3.81 % | 1443 | 8.94 % | 1441 | 8.81 % | 1488 | 11.69 % |
| A-n64-k9 | 1401 | 1541 | 9.09 % | 1592 | 12.00 % | 1518 | 7.71 % | 1483 | 5.53 % |
| A-n65-k9 | 1174 | 1304 | 9.97 % | 1282 | 8.42 % | 1355 | 13.36 % | 1294 | 9.27 % |
| A-n69-k9 | 1159 | 1303 | 11.05 % | 1288 | 10.02 % | 1272 | 8.88 % | 1317 | 12.00 % |
| A-n80-k10 | 1763 | 2004 | 12.03 % | 2037 | 13.45 % | 2042 | 13.66 % | 1986 | 11.23 % |
| B-n51-k7 | 1032 | 1049 | 1.62 % | 1054 | 2.09 % | 1121 | 7.94 % | 1051 | 1.81 % |
| B-n52-k7 | 747 | 787 | 5.08 % | 805 | 7.20 % | 927 | 19.42 % | 803 | 6.97 % |
| B-n56-k7 | 707 | 779 | 9.24 % | 789 | 10.39 % | 827 | 14.51 % | 786 | 10.05 % |
| B-n57-k7 | 1153 | 1366 | 15.59 % | 1253 | 7.98 % | 1238 | 6.87 % | 1326 | 13.05 % |
| B-n57-k9 | 1598 | 1687 | 5.28 % | 1728 | 7.52 % | 1747 | 8.53 % | 1711 | 6.60 % |
| B-n63-k10 | 1496 | 1628 | 8.11 % | 1576 | 5.08 % | 1654 | 9.55 % | 1673 | 10.58 % |
| B-n64-k9 | 861 | 944 | 8.79 % | 974 | 11.60 % | 996 | 13.55 % | 1011 | 14.84 % |
| B-n66-k9 | 1316 | 1419 | 7.26 % | 1449 | 9.18 % | 1389 | 5.26 % | 1368 | 3.80 % |
| B-n67-k10 | 1032 | 1161 | 11.11 % | 1095 | 5.75 % | 1132 | 8.83 % | 1169 | 11.72 % |
| B-n68-k9 | 1272 | 1444 | 11.91 % | 1454 | 12.52 % | 1415 | 10.11 % | 1371 | 7.22 % |
| B-n78-k10 | 1221 | 1371 | 10.94 % | 1375 | 11.20 % | 1364 | 10.48 % | 1366 | 10.61 % |
| E-n51-k5 | 521 | 563 | 7.46 % | 562 | 7.30 % | 593 | 12.14 % | 580 | 10.17 % |
| E-n76-k7 | 682 | 818 | 16.63 % | 795 | 14.21 % | 806 | 15.38 % | 808 | 15.59 % |
| E-n76-k8 | 735 | 860 | 14.53 % | 824 | 10.80 % | 842 | 12.71 % | 852 | 13.73 % |
| E-n76-k10 | 830 | 959 | 13.45 % | 939 | 11.61 % | 940 | 11.70 % | 919 | 9.68 % |
| E-n76-k14 | 1021 | 1148 | 11.06 % | 1147 | 10.99 % | 1141 | 10.52 % | 1139 | 10.36 % |
| E-n101-k8 | 815 | 1003 | 18.74 % | 1012 | 19.47 % | 1155 | 29.44 % | 1046 | 22.08 % |
| E-n101-k14 | 1067 | 1214 | 12.11 % | 1302 | 18.05 % | 1286 | 17.03 % | 1297 | 17.73 % |
| F-n72-k4 | 237 | 290 | 18.28 % | 288 | 17.71 % | 356 | 33.43 % | 311 | 23.79 % |
| F-n135-k7 | 1162 | 1785 | 34.90 % | 2380 | 51.18 % | 2312 | 49.74 % | 1772 | 34.42 % |
| M-n101-k10 | 820 | 1127 | 27.24 % | 1159 | 29.25 % | 1137 | 27.88 % | 1104 | 25.72 % |
| M-n121-k7 | 1034 | 1386 | 25.40 % | 1575 | 34.35 % | 1972 | 47.57 % | 1407 | 26.51 % |
| M-n151-k12 | 1015 | 1249 | 18.73 % | 1607 | 36.84 % | 2114 | 51.99 % | 1515 | 33.00 % |
| M-n200-k16 | 1274 | 1652 | 22.88 % | 2424 | 47.44 % | 2802 | 54.53 % | 2013 | 36.71 % |
| M-n200-k17 | 1275 | 1640 | 22.26 % | 2414 | 47.18 % | 2807 | 54.58 % | 2003 | 36.35 % |
| Avg. Gap | 0.00 % | | 12.62% | | 15.23% | | 17.96% | | 14.29% |

Table F.12: CVRPLib results. The models are trained on CVRP20. Greedy multi-start decoding is used.

| Instance | Opt. (BKS) | AM | | POMO | | SymNCO | | AM-XL | |
|---|---|---|---|---|---|---|---|---|---|
| | | Cost | Gap ↓ | Cost | Gap ↓ | Cost | Gap ↓ | Cost | Gap ↓ |
| A-n53-k7 | 1010 | 1078 | 6.31 % | 1038 | 2.70 % | 1059 | 4.63 % | 1069 | 5.52 % |
| A-n54-k7 | 1167 | 1262 | 7.53 % | 1273 | 8.33 % | 1258 | 7.23 % | 1250 | 6.64 % |
| A-n55-k9 | 1073 | 1139 | 5.79 % | 1202 | 10.73 % | 1154 | 7.02 % | 1218 | 11.90 % |
| A-n60-k9 | 1354 | 1430 | 5.31 % | 1442 | 6.10 % | 1482 | 8.64 % | 1508 | 10.21 % |
| A-n61-k9 | 1034 | 1125 | 8.09 % | 1170 | 11.62 % | 1129 | 8.41 % | 1169 | 11.55 % |
| A-n62-k8 | 1288 | 1389 | 7.27 % | 1380 | 6.67 % | 1385 | 7.00 % | 1375 | 6.33 % |
| A-n63-k9 | 1616 | 1689 | 4.32 % | 1762 | 8.29 % | 1789 | 9.67 % | 1760 | 8.18 % |
| A-n63-k10 | 1314 | 1371 | 4.16 % | 1480 | 11.22 % | 1396 | 5.87 % | 1422 | 7.59 % |
| A-n64-k9 | 1401 | 1545 | 9.32 % | 1512 | 7.34 % | 1510 | 7.22 % | 1486 | 5.72 % |
| A-n65-k9 | 1174 | 1243 | 5.55 % | 1286 | 8.71 % | 1344 | 12.65 % | 1285 | 8.64 % |
| A-n69-k9 | 1159 | 1256 | 7.72 % | 1279 | 9.38 % | 1248 | 7.13 % | 1299 | 10.78 % |
| A-n80-k10 | 1763 | 1981 | 11.00 % | 2032 | 13.24 % | 1980 | 10.96 % | 1931 | 8.70 % |
| B-n51-k7 | 1032 | 1049 | 1.62 % | 1052 | 1.90 % | 1081 | 4.53 % | 1044 | 1.15 % |
| B-n52-k7 | 747 | 791 | 5.56 % | 822 | 9.12 % | 857 | 12.84 % | 800 | 6.62 % |
| B-n56-k7 | 707 | 757 | 6.61 % | 761 | 7.10 % | 805 | 12.17 % | 774 | 8.66 % |
| B-n57-k7 | 1153 | 1298 | 11.17 % | 1212 | 4.87 % | 1231 | 6.34 % | 1258 | 8.35 % |
| B-n57-k9 | 1598 | 1753 | 8.84 % | 1689 | 5.39 % | 1711 | 6.60 % | 1665 | 4.02 % |
| B-n63-k10 | 1496 | 1629 | 8.16 % | 1560 | 4.10 % | 1624 | 7.88 % | 1635 | 8.50 % |
| B-n64-k9 | 861 | 947 | 9.08 % | 953 | 9.65 % | 993 | 13.29 % | 998 | 13.73 % |
| B-n66-k9 | 1316 | 1423 | 7.52 % | 1423 | 7.52 % | 1388 | 5.19 % | 1396 | 5.73 % |
| B-n67-k10 | 1032 | 1152 | 10.42 % | 1120 | 7.86 % | 1130 | 8.67 % | 1145 | 9.87 % |
| B-n68-k9 | 1272 | 1410 | 9.79 % | 1443 | 11.85 % | 1334 | 4.65 % | 1352 | 5.92 % |
| B-n78-k10 | 1221 | 1368 | 10.75 % | 1367 | 10.68 % | 1340 | 8.88 % | 1357 | 10.02 % |
| E-n51-k5 | 521 | 561 | 7.13 % | 579 | 10.02 % | 590 | 11.69 % | 564 | 7.62 % |
| E-n76-k7 | 682 | 771 | 11.54 % | 777 | 12.23 % | 822 | 17.03 % | 791 | 13.78 % |
| E-n76-k8 | 735 | 825 | 10.91 % | 796 | 7.66 % | 861 | 14.63 % | 827 | 11.12 % |
| E-n76-k10 | 830 | 942 | 11.89 % | 928 | 10.56 % | 950 | 12.63 % | 917 | 9.49 % |
| E-n76-k14 | 1021 | 1099 | 7.10 % | 1150 | 11.22 % | 1135 | 10.04 % | 1134 | 9.96 % |
| E-n101-k8 | 815 | 979 | 16.75 % | 979 | 16.75 % | 1117 | 27.04 % | 1042 | 21.79 % |
| E-n101-k14 | 1067 | 1215 | 12.18 % | 1254 | 14.91 % | 1239 | 13.88 % | 1262 | 15.45 % |
| F-n72-k4 | 237 | 284 | 16.55 % | 291 | 18.56 % | 359 | 33.98 % | 317 | 25.24 % |
| F-n135-k7 | 1162 | 1820 | 36.15 % | 1883 | 38.29 % | 2158 | 46.15 % | 1676 | 30.67 % |
| M-n101-k10 | 820 | 1058 | 22.50 % | 1056 | 22.35 % | 1121 | 26.85 % | 1032 | 20.54 % |
| M-n121-k7 | 1034 | 1355 | 23.69 % | 1302 | 20.58 % | 1936 | 46.59 % | 1415 | 26.93 % |
| M-n151-k12 | 1015 | 1274 | 20.33 % | 1309 | 22.46 % | 1856 | 45.31 % | 1435 | 29.27 % |
| M-n200-k16 | 1274 | 1606 | 20.67 % | 1654 | 22.97 % | 2519 | 49.42 % | 1883 | 32.34 % |
| M-n200-k17 | 1275 | 1606 | 20.61 % | 1654 | 22.91 % | 2519 | 49.38 % | 1883 | 32.29 % |
| Avg. Gap | 0.00 % | | 11.08% | | 11.78% | | 16.00% | | 12.72% |

Table F.13: CVRPLib results. The models are trained on CVRP20. Augmentation decoding (100) is used.

| Instance | Opt. (BKS) | AM | | POMO | | SymNCO | | AM-XL | |
|---|---|---|---|---|---|---|---|---|---|
| | | Cost | Gap ↓ | Cost | Gap ↓ | Cost | Gap ↓ | Cost | Gap ↓ |
| A-n53-k7 | 1010 | 1051 | 3.90 % | 1028 | 1.75 % | 1080 | 6.48 % | 1094 | 7.68 % |
| A-n54-k7 | 1167 | 1204 | 3.07 % | 1255 | 7.01 % | 1224 | 4.66 % | 1256 | 7.09 % |
| A-n55-k9 | 1073 | 1139 | 5.79 % | 1153 | 6.94 % | 1155 | 7.10 % | 1174 | 8.60 % |
| A-n60-k9 | 1354 | 1425 | 4.98 % | 1438 | 5.84 % | 1481 | 8.58 % | 1459 | 7.20 % |
| A-n61-k9 | 1034 | 1119 | 7.60 % | 1144 | 9.62 % | 1127 | 8.25 % | 1149 | 10.01 % |
| A-n62-k8 | 1288 | 1349 | 4.52 % | 1410 | 8.65 % | 1374 | 6.26 % | 1376 | 6.40 % |
| A-n63-k9 | 1616 | 1692 | 4.49 % | 1715 | 5.77 % | 1767 | 8.55 % | 1741 | 7.18 % |
| A-n63-k10 | 1314 | 1364 | 3.67 % | 1380 | 4.78 % | 1368 | 3.95 % | 1395 | 5.81 % |
| A-n64-k9 | 1401 | 1490 | 5.97 % | 1501 | 6.66 % | 1513 | 7.40 % | 1490 | 5.97 % |
| A-n65-k9 | 1174 | 1251 | 6.16 % | 1258 | 6.68 % | 1255 | 6.45 % | 1263 | 7.05 % |
| A-n69-k9 | 1159 | 1250 | 7.28 % | 1263 | 8.23 % | 1238 | 6.38 % | 1254 | 7.58 % |
| A-n80-k10 | 1763 | 1907 | 7.55 % | 1937 | 8.98 % | 1946 | 9.40 % | 1928 | 8.56 % |
| B-n51-k7 | 1032 | 1040 | 0.77 % | 1040 | 0.77 % | 1053 | 1.99 % | 1041 | 0.86 % |
| B-n52-k7 | 747 | 770 | 2.99 % | 775 | 3.61 % | 755 | 1.06 % | 765 | 2.35 % |
| B-n56-k7 | 707 | 765 | 7.58 % | 761 | 7.10 % | 769 | 8.06 % | 766 | 7.70 % |
| B-n57-k7 | 1153 | 1215 | 5.10 % | 1183 | 2.54 % | 1231 | 6.34 % | 1183 | 2.54 % |
| B-n57-k9 | 1598 | 1655 | 3.44 % | 1656 | 3.50 % | 1681 | 4.94 % | 1635 | 2.26 % |
| B-n63-k10 | 1496 | 1603 | 6.67 % | 1553 | 3.67 % | 1604 | 6.73 % | 1603 | 6.67 % |
| B-n64-k9 | 861 | 940 | 8.40 % | 932 | 7.62 % | 922 | 6.62 % | 950 | 9.37 % |
| B-n66-k9 | 1316 | 1407 | 6.47 % | 1406 | 6.40 % | 1375 | 4.29 % | 1357 | 3.02 % |
| B-n67-k10 | 1032 | 1113 | 7.28 % | 1095 | 5.75 % | 1096 | 5.84 % | 1097 | 5.93 % |
| B-n68-k9 | 1272 | 1339 | 5.00 % | 1345 | 5.43 % | 1337 | 4.86 % | 1371 | 7.22 % |
| B-n78-k10 | 1221 | 1349 | 9.49 % | 1348 | 9.42 % | 1315 | 7.15 % | 1335 | 8.54 % |
| E-n51-k5 | 521 | 536 | 2.80 % | 565 | 7.79 % | 573 | 9.08 % | 563 | 7.46 % |
| E-n76-k7 | 682 | 771 | 11.54 % | 747 | 8.70 % | 781 | 12.68 % | 762 | 10.50 % |
| E-n76-k8 | 735 | 819 | 10.26 % | 805 | 8.70 % | 826 | 11.02 % | 815 | 9.82 % |
| E-n76-k10 | 830 | 911 | 8.89 % | 919 | 9.68 % | 915 | 9.29 % | 919 | 9.68 % |
| E-n76-k14 | 1021 | 1097 | 6.93 % | 1109 | 7.94 % | 1131 | 9.73 % | 1114 | 8.35 % |
| E-n101-k8 | 815 | 948 | 14.03 % | 922 | 11.61 % | 1093 | 25.43 % | 946 | 13.85 % |
| E-n101-k14 | 1067 | 1184 | 9.88 % | 1212 | 11.96 % | 1219 | 12.47 % | 1225 | 12.90 % |
| F-n72-k4 | 237 | 274 | 13.50 % | 288 | 17.71 % | 305 | 22.30 % | 291 | 18.56 % |
| F-n135-k7 | 1162 | 1596 | 27.19 % | 1505 | 22.79 % | 1735 | 33.03 % | 1597 | 27.24 % |
| M-n101-k10 | 820 | 1017 | 19.37 % | 1015 | 19.21 % | 1089 | 24.70 % | 1061 | 22.71 % |
| M-n121-k7 | 1034 | 1233 | 16.14 % | 1251 | 17.35 % | 1662 | 37.79 % | 1312 | 21.19 % |
| M-n151-k12 | 1015 | 1208 | 15.98 % | 1236 | 17.88 % | 1655 | 38.67 % | 1334 | 23.91 % |
| M-n200-k16 | 1274 | 1559 | 18.28 % | 1668 | 23.62 % | 2360 | 46.02 % | 1815 | 29.81 % |
| M-n200-k17 | 1275 | 1572 | 18.89 % | 1704 | 25.18 % | 2423 | 47.38 % | 1791 | 28.81 % |
| Avg. Gap | 0.00 % | | 8.70% | | 9.37% | | 13.00% | | 10.55% |

Table F.14: CVRPLib results. The models are trained on CVRP50. Greedy decoding is used.

| Instance | Opt. (BKS) | AM | | POMO | | SymNCO | | AM-XL | |
|---|---|---|---|---|---|---|---|---|---|
| | | Cost | Gap ↓ | Cost | Gap ↓ | Cost | Gap ↓ | Cost | Gap ↓ |
| A-n53-k7 | 1010 | 1077 | 6.22 % | 1071 | 5.70 % | 1086 | 7.00 % | 1079 | 6.39 % |
| A-n54-k7 | 1167 | 1251 | 6.71 % | 1228 | 4.97 % | 1211 | 3.63 % | 1213 | 3.79 % |
| A-n55-k9 | 1073 | 1166 | 7.98 % | 1107 | 3.07 % | 1158 | 7.34 % | 1203 | 10.81 % |
| A-n60-k9 | 1354 | 1460 | 7.26 % | 1419 | 4.58 % | 1452 | 6.75 % | 1417 | 4.45 % |
| A-n61-k9 | 1034 | 1079 | 4.17 % | 1089 | 5.05 % | 1050 | 1.52 % | 1126 | 8.17 % |
| A-n62-k8 | 1288 | 1367 | 5.78 % | 1385 | 7.00 % | 1369 | 5.92 % | 1331 | 3.23 % |
| A-n63-k9 | 1616 | 1682 | 3.92 % | 1679 | 3.75 % | 1671 | 3.29 % | 1659 | 2.59 % |
| A-n63-k10 | 1314 | 1347 | 2.45 % | 1426 | 7.85 % | 1398 | 6.01 % | 1402 | 6.28 % |
| A-n64-k9 | 1401 | 1493 | 6.16 % | 1436 | 2.44 % | 1469 | 4.63 % | 1469 | 4.63 % |
| A-n65-k9 | 1174 | 1247 | 5.85 % | 1247 | 5.85 % | 1216 | 3.45 % | 1255 | 6.45 % |
| A-n69-k9 | 1159 | 1264 | 8.31 % | 1210 | 4.21 % | 1232 | 5.93 % | 1224 | 5.31 % |
| A-n80-k10 | 1763 | 1864 | 5.42 % | 1923 | 8.32 % | 1921 | 8.22 % | 1881 | 6.27 % |
| B-n51-k7 | 1032 | 1134 | 8.99 % | 1153 | 10.49 % | 1166 | 11.49 % | 1116 | 7.53 % |
| B-n52-k7 | 747 | 770 | 2.99 % | 770 | 2.99 % | 809 | 7.66 % | 784 | 4.72 % |
| B-n56-k7 | 707 | 751 | 5.86 % | 748 | 5.48 % | 813 | 13.04 % | 779 | 9.24 % |
| B-n57-k7 | 1153 | 1182 | 2.45 % | 1227 | 6.03 % | 1189 | 3.03 % | 1970 | 41.47 % |
| B-n57-k9 | 1598 | 1670 | 4.31 % | 1686 | 5.22 % | 1660 | 3.73 % | 1660 | 3.73 % |
| B-n63-k10 | 1496 | 1634 | 8.45 % | 1644 | 9.00 % | 1621 | 7.71 % | 1598 | 6.38 % |
| B-n64-k9 | 861 | 964 | 10.68 % | 970 | 11.24 % | 972 | 11.42 % | 922 | 6.62 % |
| B-n66-k9 | 1316 | 1374 | 4.22 % | 1363 | 3.45 % | 1362 | 3.38 % | 1388 | 5.19 % |
| B-n67-k10 | 1032 | 1137 | 9.23 % | 1148 | 10.10 % | 1135 | 9.07 % | 1189 | 13.20 % |
| B-n68-k9 | 1272 | 1397 | 8.95 % | 1329 | 4.29 % | 1317 | 3.42 % | 1374 | 7.42 % |
| B-n78-k10 | 1221 | 1325 | 7.85 % | 1316 | 7.22 % | 1320 | 7.50 % | 1294 | 5.64 % |
| E-n51-k5 | 521 | 554 | 5.96 % | 582 | 10.48 % | 567 | 8.11 % | 569 | 8.44 % |
| E-n76-k7 | 682 | 712 | 4.21 % | 744 | 8.33 % | 750 | 9.07 % | 740 | 7.84 % |
| E-n76-k8 | 735 | 760 | 3.29 % | 816 | 9.93 % | 795 | 7.55 % | 776 | 5.28 % |
| E-n76-k10 | 830 | 867 | 4.27 % | 901 | 7.88 % | 893 | 7.05 % | 880 | 5.68 % |
| E-n76-k14 | 1021 | 1075 | 5.02 % | 1076 | 5.11 % | 1133 | 9.89 % | 1115 | 8.43 % |
| E-n101-k8 | 815 | 920 | 11.41 % | 900 | 9.44 % | 896 | 9.04 % | 892 | 8.63 % |
| E-n101-k14 | 1067 | 1171 | 8.88 % | 1171 | 8.88 % | 1193 | 10.56 % | 1139 | 6.32 % |
| F-n72-k4 | 237 | 305 | 22.30 % | 295 | 19.66 % | 320 | 25.94 % | 293 | 19.11 % |
| F-n135-k7 | 1162 | 1499 | 22.48 % | 1553 | 25.18 % | 1734 | 32.99 % | 2300 | 49.48 % |
| M-n101-k10 | 820 | 929 | 11.73 % | 917 | 10.58 % | 982 | 16.50 % | 888 | 7.66 % |
| M-n121-k7 | 1034 | 1209 | 14.47 % | 1300 | 20.46 % | 1373 | 24.69 % | 1177 | 12.15 % |
| M-n151-k12 | 1015 | 1163 | 12.73 % | 1161 | 12.58 % | 1261 | 19.51 % | 1157 | 12.27 % |
| M-n200-k16 | 1274 | 1597 | 20.23 % | 1505 | 15.35 % | 1742 | 26.87 % | 1479 | 13.86 % |
| M-n200-k17 | 1275 | 1597 | 20.16 % | 1505 | 15.28 % | 1742 | 26.81 % | 1479 | 13.79 % |
| Avg. Gap | 0.00 % | | 8.42% | | 8.58% | | 10.26% | | 9.69% |

Table F.15: CVRPLib results. The models are trained on CVRP50. Sampling decoding (100 with temperature $\tau = 0.05$) is used.

| Instance | Opt. (BKS) | AM | | POMO | | SymNCO | | AM-XL | |
|---|---|---|---|---|---|---|---|---|---|
| | | Cost | Gap ↓ | Cost | Gap ↓ | Cost | Gap ↓ | Cost | Gap ↓ |
| A-n53-k7 | 1010 | 1072 | 5.78 % | 1071 | 5.70 % | 1086 | 7.00 % | 1079 | 6.39 % |
| A-n54-k7 | 1167 | 1251 | 6.71 % | 1204 | 3.07 % | 1211 | 3.63 % | 1213 | 3.79 % |
| A-n55-k9 | 1073 | 1166 | 7.98 % | 1107 | 3.07 % | 1158 | 7.34 % | 1203 | 10.81 % |
| A-n60-k9 | 1354 | 1457 | 7.07 % | 1408 | 3.84 % | 1448 | 6.49 % | 1417 | 4.45 % |
| A-n61-k9 | 1034 | 1076 | 3.90 % | 1089 | 5.05 % | 1050 | 1.52 % | 1123 | 7.93 % |
| A-n62-k8 | 1288 | 1361 | 5.36 % | 1382 | 6.80 % | 1368 | 5.85 % | 1328 | 3.01 % |
| A-n63-k9 | 1616 | 1670 | 3.23 % | 1679 | 3.75 % | 1667 | 3.06 % | 1659 | 2.59 % |
| A-n63-k10 | 1314 | 1346 | 2.38 % | 1366 | 3.81 % | 1398 | 6.01 % | 1402 | 6.28 % |
| A-n64-k9 | 1401 | 1493 | 6.16 % | 1430 | 2.03 % | 1460 | 4.04 % | 1465 | 4.37 % |
| A-n65-k9 | 1174 | 1241 | 5.40 % | 1229 | 4.48 % | 1216 | 3.45 % | 1255 | 6.45 % |
| A-n69-k9 | 1159 | 1234 | 6.08 % | 1199 | 3.34 % | 1230 | 5.77 % | 1219 | 4.92 % |
| A-n80-k10 | 1763 | 1862 | 5.32 % | 1882 | 6.32 % | 1904 | 7.41 % | 1881 | 6.27 % |
| B-n51-k7 | 1032 | 1134 | 8.99 % | 1153 | 10.49 % | 1166 | 11.49 % | 1116 | 7.53 % |
| B-n52-k7 | 747 | 769 | 2.86 % | 768 | 2.73 % | 809 | 7.66 % | 784 | 4.72 % |
| B-n56-k7 | 707 | 751 | 5.86 % | 747 | 5.35 % | 811 | 12.82 % | 762 | 7.22 % |
| B-n57-k7 | 1153 | 1181 | 2.37 % | 1225 | 5.88 % | 1182 | 2.45 % | 1703 | 32.30 % |
| B-n57-k9 | 1598 | 1669 | 4.25 % | 1680 | 4.88 % | 1660 | 3.73 % | 1654 | 3.39 % |
| B-n63-k10 | 1496 | 1634 | 8.45 % | 1592 | 6.03 % | 1620 | 7.65 % | 1598 | 6.38 % |
| B-n64-k9 | 861 | 955 | 9.84 % | 967 | 10.96 % | 972 | 11.42 % | 922 | 6.62 % |
| B-n66-k9 | 1316 | 1371 | 4.01 % | 1356 | 2.95 % | 1362 | 3.38 % | 1383 | 4.84 % |
| B-n67-k10 | 1032 | 1136 | 9.15 % | 1142 | 9.63 % | 1133 | 8.91 % | 1180 | 12.54 % |
| B-n68-k9 | 1272 | 1383 | 8.03 % | 1324 | 3.93 % | 1317 | 3.42 % | 1374 | 7.42 % |
| B-n78-k10 | 1221 | 1325 | 7.85 % | 1285 | 4.98 % | 1316 | 7.22 % | 1294 | 5.64 % |
| E-n51-k5 | 521 | 551 | 5.44 % | 574 | 9.23 % | 565 | 7.79 % | 569 | 8.44 % |
| E-n76-k7 | 682 | 707 | 3.54 % | 738 | 7.59 % | 744 | 8.33 % | 740 | 7.84 % |
| E-n76-k8 | 735 | 760 | 3.29 % | 816 | 9.93 % | 786 | 6.49 % | 774 | 5.04 % |
| E-n76-k10 | 830 | 862 | 3.71 % | 895 | 7.26 % | 893 | 7.05 % | 876 | 5.25 % |
| E-n76-k14 | 1021 | 1075 | 5.02 % | 1076 | 5.11 % | 1128 | 9.49 % | 1105 | 7.60 % |
| E-n101-k8 | 815 | 894 | 8.84 % | 891 | 8.53 % | 890 | 8.43 % | 888 | 8.22 % |
| E-n101-k14 | 1067 | 1171 | 8.88 % | 1161 | 8.10 % | 1166 | 8.49 % | 1138 | 6.24 % |
| F-n72-k4 | 237 | 305 | 22.30 % | 293 | 19.11 % | 307 | 22.80 % | 289 | 17.99 % |
| F-n135-k7 | 1162 | 1453 | 20.03 % | 1521 | 23.60 % | 1557 | 25.37 % | 2141 | 45.73 % |
| M-n101-k10 | 820 | 927 | 11.54 % | 915 | 10.38 % | 969 | 15.38 % | 888 | 7.66 % |
| M-n121-k7 | 1034 | 1145 | 9.69 % | 1276 | 18.97 % | 1347 | 23.24 % | 1121 | 7.76 % |
| M-n151-k12 | 1015 | 1148 | 11.59 % | 1128 | 10.02 % | 1228 | 17.35 % | 1120 | 9.38 % |
| M-n200-k16 | 1274 | 1520 | 16.18 % | 1486 | 14.27 % | 1642 | 22.41 % | 1462 | 12.86 % |
| M-n200-k17 | 1275 | 1558 | 18.16 % | 1475 | 13.56 % | 1664 | 23.38 % | 1460 | 12.67 % |
| Avg. Gap | 0.00 % | 7.71% | | 7.70% | | 9.40% | | 8.88% | |

Table F.16: CVRPLib results. The models are trained on CVRP50. Greedy multi-start decoding is used.

| Instance | Opt. (BKS) | AM | | POMO | | SymNCO | | AM-XL | |
|---|---|---|---|---|---|---|---|---|---|
| | | Cost | Gap ↓ | Cost | Gap ↓ | Cost | Gap ↓ | Cost | Gap ↓ |
| A-n53-k7 | 1010 | 1063 | 4.99 % | 1061 | 4.81 % | 1071 | 5.70 % | 1059 | 4.63 % |
| A-n54-k7 | 1167 | 1217 | 4.11 % | 1188 | 1.77 % | 1201 | 2.83 % | 1211 | 3.63 % |
| A-n55-k9 | 1073 | 1116 | 3.85 % | 1096 | 2.10 % | 1117 | 3.94 % | 1144 | 6.21 % |
| A-n60-k9 | 1354 | 1410 | 3.97 % | 1393 | 2.80 % | 1390 | 2.59 % | 1379 | 1.81 % |
| A-n61-k9 | 1034 | 1067 | 3.09 % | 1077 | 3.99 % | 1050 | 1.52 % | 1082 | 4.44 % |
| A-n62-k8 | 1288 | 1346 | 4.31 % | 1348 | 4.45 % | 1354 | 4.87 % | 1331 | 3.23 % |
| A-n63-k9 | 1616 | 1682 | 3.92 % | 1659 | 2.59 % | 1668 | 3.12 % | 1657 | 2.47 % |
| A-n63-k10 | 1314 | 1347 | 2.45 % | 1336 | 1.65 % | 1362 | 3.52 % | 1348 | 2.52 % |
| A-n64-k9 | 1401 | 1486 | 5.72 % | 1426 | 1.75 % | 1438 | 2.57 % | 1461 | 4.11 % |
| A-n65-k9 | 1174 | 1231 | 4.63 % | 1220 | 3.77 % | 1216 | 3.45 % | 1226 | 4.24 % |
| A-n69-k9 | 1159 | 1189 | 2.52 % | 1189 | 2.52 % | 1200 | 3.42 % | 1209 | 4.14 % |
| A-n80-k10 | 1763 | 1845 | 4.44 % | 1834 | 3.87 % | 1840 | 4.18 % | 1811 | 2.65 % |
| B-n51-k7 | 1032 | 1038 | 0.58 % | 1037 | 0.48 % | 1078 | 4.27 % | 1078 | 4.27 % |
| B-n52-k7 | 747 | 769 | 2.86 % | 764 | 2.23 % | 797 | 6.27 % | 768 | 2.73 % |
| B-n56-k7 | 707 | 748 | 5.48 % | 743 | 4.85 % | 786 | 10.05 % | 774 | 8.66 % |
| B-n57-k7 | 1153 | 1177 | 2.04 % | 1171 | 1.54 % | 1169 | 1.37 % | 1680 | 31.37 % |
| B-n57-k9 | 1598 | 1662 | 3.85 % | 1651 | 3.21 % | 1655 | 3.44 % | 1642 | 2.68 % |
| B-n63-k10 | 1496 | 1612 | 7.20 % | 1590 | 5.91 % | 1606 | 6.85 % | 1598 | 6.38 % |
| B-n64-k9 | 861 | 936 | 8.01 % | 943 | 8.70 % | 939 | 8.31 % | 922 | 6.62 % |
| B-n66-k9 | 1316 | 1365 | 3.59 % | 1348 | 2.37 % | 1351 | 2.59 % | 1376 | 4.36 % |
| B-n67-k10 | 1032 | 1123 | 8.10 % | 1111 | 7.11 % | 1128 | 8.51 % | 1137 | 9.23 % |
| B-n68-k9 | 1272 | 1345 | 5.43 % | 1315 | 3.27 % | 1312 | 3.05 % | 1351 | 5.85 % |
| B-n78-k10 | 1221 | 1289 | 5.28 % | 1278 | 4.46 % | 1298 | 5.93 % | 1288 | 5.20 % |
| E-n51-k5 | 521 | 554 | 5.96 % | 533 | 2.25 % | 553 | 5.79 % | 567 | 8.11 % |
| E-n76-k7 | 682 | 711 | 4.08 % | 719 | 5.15 % | 742 | 8.09 % | 707 | 3.54 % |
| E-n76-k8 | 735 | 760 | 3.29 % | 770 | 4.55 % | 765 | 3.92 % | 776 | 5.28 % |
| E-n76-k10 | 830 | 854 | 2.81 % | 870 | 4.60 % | 890 | 6.74 % | 878 | 5.47 % |
| E-n76-k14 | 1021 | 1062 | 3.86 % | 1059 | 3.59 % | 1078 | 5.29 % | 1073 | 4.85 % |
| E-n101-k8 | 815 | 893 | 8.73 % | 890 | 8.43 % | 885 | 7.91 % | 890 | 8.43 % |
| E-n101-k14 | 1067 | 1154 | 7.54 % | 1155 | 7.62 % | 1182 | 9.73 % | 1139 | 6.32 % |
| F-n72-k4 | 237 | 295 | 19.66 % | 268 | 11.57 % | 294 | 19.39 % | 290 | 18.28 % |
| F-n135-k7 | 1162 | 1414 | 17.82 % | 1409 | 17.53 % | 1554 | 25.23 % | 1984 | 41.43 % |
| M-n101-k10 | 820 | 877 | 6.50 % | 892 | 8.07 % | 922 | 11.06 % | 886 | 7.45 % |
| M-n121-k7 | 1034 | 1151 | 10.17 % | 1234 | 16.21 % | 1352 | 23.52 % | 1113 | 7.10 % |
| M-n151-k12 | 1015 | 1146 | 11.43 % | 1141 | 11.04 % | 1256 | 19.19 % | 1118 | 9.21 % |
| M-n200-k16 | 1274 | 1537 | 17.11 % | 1477 | 13.74 % | 1629 | 21.79 % | 1453 | 12.32 % |
| M-n200-k17 | 1275 | 1537 | 17.05 % | 1477 | 13.68 % | 1629 | 21.73 % | 1453 | 12.25 % |
| Avg. Gap | 0.00 % | | 6.39% | | 5.63% | | 7.88% | | 7.61% |

Table F.17: CVRPLib results. The models are trained on CVRP50. Augmentation decoding (100) is used.

| Instance | Opt. (BKS) | AM | | POMO | | SymNCO | | AM-XL | |
|---|---|---|---|---|---|---|---|---|---|
| | | Cost | Gap ↓ | Cost | Gap ↓ | Cost | Gap ↓ | Cost | Gap ↓ |
| A-n53-k7 | 1010 | 1045 | 3.35 % | 1046 | 3.44 % | 1062 | 4.90 % | 1064 | 5.08 % |
| A-n54-k7 | 1167 | 1200 | 2.75 % | 1181 | 1.19 % | 1197 | 2.51 % | 1176 | 0.77 % |
| A-n55-k9 | 1073 | 1090 | 1.56 % | 1107 | 3.07 % | 1104 | 2.81 % | 1117 | 3.94 % |
| A-n60-k9 | 1354 | 1378 | 1.74 % | 1392 | 2.73 % | 1389 | 2.52 % | 1380 | 1.88 % |
| A-n61-k9 | 1034 | 1058 | 2.27 % | 1050 | 1.52 % | 1050 | 1.52 % | 1065 | 2.91 % |
| A-n62-k8 | 1288 | 1334 | 3.45 % | 1329 | 3.09 % | 1330 | 3.16 % | 1321 | 2.50 % |
| A-n63-k9 | 1616 | 1655 | 2.36 % | 1651 | 2.12 % | 1659 | 2.59 % | 1659 | 2.59 % |
| A-n63-k10 | 1314 | 1339 | 1.87 % | 1358 | 3.24 % | 1344 | 2.23 % | 1351 | 2.74 % |
| A-n64-k9 | 1401 | 1442 | 2.84 % | 1434 | 2.30 % | 1435 | 2.37 % | 1443 | 2.91 % |
| A-n65-k9 | 1174 | 1205 | 2.57 % | 1206 | 2.65 % | 1199 | 2.09 % | 1217 | 3.53 % |
| A-n69-k9 | 1159 | 1201 | 3.50 % | 1199 | 3.34 % | 1186 | 2.28 % | 1196 | 3.09 % |
| A-n80-k10 | 1763 | 1811 | 2.65 % | 1843 | 4.34 % | 1816 | 2.92 % | 1822 | 3.24 % |
| B-n51-k7 | 1032 | 1026 | -0.58 % | 1027 | -0.49 % | 1031 | -0.10 % | 1025 | -0.68 % |
| B-n52-k7 | 747 | 763 | 2.10 % | 759 | 1.58 % | 763 | 2.10 % | 758 | 1.45 % |
| B-n56-k7 | 707 | 736 | 3.94 % | 723 | 2.21 % | 741 | 4.59 % | 732 | 3.42 % |
| B-n57-k7 | 1153 | 1167 | 1.20 % | 1159 | 0.52 % | 1175 | 1.87 % | 1166 | 1.11 % |
| B-n57-k9 | 1598 | 1651 | 3.21 % | 1634 | 2.20 % | 1625 | 1.66 % | 1623 | 1.54 % |
| B-n63-k10 | 1496 | 1581 | 5.38 % | 1580 | 5.32 % | 1576 | 5.08 % | 1587 | 5.73 % |
| B-n64-k9 | 861 | 930 | 7.42 % | 920 | 6.41 % | 920 | 6.41 % | 922 | 6.62 % |
| B-n66-k9 | 1316 | 1349 | 2.45 % | 1338 | 1.64 % | 1345 | 2.16 % | 1359 | 3.16 % |
| B-n67-k10 | 1032 | 1073 | 3.82 % | 1061 | 2.73 % | 1069 | 3.46 % | 1088 | 5.15 % |
| B-n68-k9 | 1272 | 1312 | 3.05 % | 1323 | 3.85 % | 1317 | 3.42 % | 1346 | 5.50 % |
| B-n78-k10 | 1221 | 1265 | 3.48 % | 1286 | 5.05 % | 1277 | 4.39 % | 1269 | 3.78 % |
| E-n51-k5 | 521 | 535 | 2.62 % | 547 | 4.75 % | 545 | 4.40 % | 551 | 5.44 % |
| E-n76-k7 | 682 | 707 | 3.54 % | 718 | 5.01 % | 712 | 4.21 % | 717 | 4.88 % |
| E-n76-k8 | 735 | 753 | 2.39 % | 761 | 3.42 % | 759 | 3.16 % | 763 | 3.67 % |
| E-n76-k10 | 830 | 847 | 2.01 % | 863 | 3.82 % | 861 | 3.60 % | 859 | 3.38 % |
| E-n76-k14 | 1021 | 1052 | 2.95 % | 1049 | 2.67 % | 1068 | 4.40 % | 1073 | 4.85 % |
| E-n101-k8 | 815 | 866 | 5.89 % | 874 | 6.75 % | 887 | 8.12 % | 869 | 6.21 % |
| E-n101-k14 | 1067 | 1136 | 6.07 % | 1143 | 6.65 % | 1157 | 7.78 % | 1131 | 5.66 % |
| F-n72-k4 | 237 | 271 | 12.55 % | 269 | 11.90 % | 266 | 10.90 % | 281 | 15.66 % |
| F-n135-k7 | 1162 | 1307 | 11.09 % | 1358 | 14.43 % | 1390 | 16.40 % | 1364 | 14.81 % |
| M-n101-k10 | 820 | 873 | 6.07 % | 905 | 9.39 % | 891 | 7.97 % | 851 | 3.64 % |
| M-n121-k7 | 1034 | 1129 | 8.41 % | 1225 | 15.59 % | 1149 | 10.01 % | 1116 | 7.35 % |
| M-n151-k12 | 1015 | 1112 | 8.72 % | 1126 | 9.86 % | 1190 | 14.71 % | 1103 | 7.98 % |
| M-n200-k16 | 1274 | 1473 | 13.51 % | 1430 | 10.91 % | 1580 | 19.37 % | 1430 | 10.91 % |
| M-n200-k17 | 1275 | 1482 | 13.97 % | 1466 | 13.03 % | 1580 | 19.30 % | 1416 | 9.96 % |
| Avg. Gap | 0.00 % | | 4.49% | | 4.93% | | 5.44% | | 4.77% |

