# OpenReview forum: "RL4CO: a Unified Reinforcement Learning for Combinatorial Optimization Library"
_ICLR.cc/2024/Conference — Submitted to ICLR 2024_

### Official Review · Reviewer_iFP3 · 2023-10-22

**Soundness:** 3 good
**Presentation:** 3 good
**Contribution:** 2 fair
**Rating:** 5
**Confidence:** 3

**Summary:**

The RL4CO paper introduces a unified Reinforcement Learning for Combinatorial Optimization library that provides a standardized code base for practitioners to address combinatorial problems using deep reinforcement learning. The library is designed to be flexible, easily modifiable, and extensible by researchers, and it benchmarks baseline RL solvers with different evaluation schemes on zero-shot performance, generalization, and adaptability on diverse tasks. The paper also discusses the limitations and areas of improvement with the current library and benchmark experiments. The paper aims to provide a comprehensive resource for researchers and practitioners to develop and evaluate RL-based approaches for combinatorial optimization problems.

**Strengths:**

1. This work introduces a benchmark that applies the most common RL methods to solve routing-related problems such as TSP and CVRP, and it is implemented in a modular way. This is beneficial for the community and practitioners to quickly compare results.
2. The paper presents some interesting results that might be helpful for the community, like some recent approaches might not match the performance of earlier methods under certain evaluation conditions.

**Weaknesses:**

1. The title of this paper is RL4CO, but it only addresses routing-related problems. Would it be more appropriate to name it RL4Routing?
2. While this paper is confined to the scope of RL4Routing, it implements various RL methods for these problems. However, at their core, these methods belong to the same category (and share very similar algorithm procedures), and there is a lack of insights to organize this category effectively. If this article could provide some useful conclusions or derive new methods based on these benchmarks, it would become novel.
3. Additionally, one cannot ignore the issue that, for example, in the case of TSP problems, many non-autoregressive (or non-RL) methods [1,2,3] have recently achieved better performance than RL-based, like faster inference time, lower optimal gap, and runnable on larger instances (TSP1000 and TSP10000). Therefore, as a benchmark for routing problems(TSP, CVRP, etc.), the comparison of methods is not comprehensive enough.


As a reviewer, I find myself in a dilemma regarding this paper. First and foremost, I acknowledge the substantial effort and contribution to the community. However, I have reservations about the completeness and rationality of the organization of RL4CO Lib, particularly with respect to the concerns raised above. Therefore,  I am still unsure whether it meets the standards of ICLR.

[1] Sun Z, Yang Y. Difusco: Graph-based diffusion solvers for combinatorial optimization[J]. arXiv preprint arXiv:2302.08224, 2023.
[2] Qiu R, Sun Z, Yang Y. Dimes: A differentiable meta solver for combinatorial optimization problems[J]. Advances in Neural Information Processing Systems, 2022, 35: 25531-25546.
[3] Sun H, Goshvadi K, Nova A, et al. Revisiting Sampling for Combinatorial Optimization[J]. ICML, 2023

**Questions:**

See the weakness part of the review.

---

> ### Author Response · Authors · 2023-11-21
> **Reply (1)**
>
> Thanks for your thorough review and for acknowledging our contributions! We will address your concerns in the following answers.
>
> > The title of this paper is RL4CO, but it only addresses routing-related problems. Would it be more appropriate to name it RL4Routing?
>
> RL4CO is not confined to routing problems only. In fact, Appendix B is about the decap placement problems from electronic design automation (EDA), which we did not include in the main text mainly because of space - given the practicality of the problem needed more background, we felt it would be better for EDA to be in a separate section of the paper.
>
> Our goal is to include a wide variety of problems with the help of our community. For instance, we recently introduced the library scheduling problems including the Single Machine Total Weighted Tardiness Problem (SMTWTP) and the Flexible Flow Shop Problem (FFSP). Moreover, RL4CO is easily extensible to other problems; for instance, a recent paper [1r] has extended our implementation to the max cover problem and facility location, which we plan to integrate in RL4CO.
>
> > This paper […]  implements various RL methods for these problems. However, at their core, these methods belong to the same category (and share very similar algorithm procedures), and there is a lack of insights to organize this category effectively. If this article could provide some useful conclusions or derive new methods based on these benchmarks, it would become novel.
>
> Thanks for your insight! We suppose by the same category you mean autoregressive constructive methods for combinatorial optimization, i.e., that directly construct a feasible solution given a problem instance node-by-node. In terms of software organization, our fundamental insight is to divide and conquer the problems (in terms of data, environment, model, training) into more manageable subproblems. For instance, while several practical implementations in the NCO community separate different problems into separate schemes - one can think about different folders each with its own problem to solve but with mostly redundant code, that make management and progressive development hard, especially for new researchers - we decided to rely on unified structures. In the case of data, this was e.g. thanks to TensorDict, that allows for batched operations on named tensors, greatly simplifying passing around instances. In the case of environment, thanks to TorchRL we created a blueprint for vectorized CO environments that can run on GPU thus greatly alleviating the typical CPU simulation bottleneck typical of RL. As for the models, given common structures in the encoding-decoding architecture, we managed to modularize the policies as described in Figure 3.2, with the main idea being to abstract away problem-specific modules from common structures as encoder and decoder layers. For the training, we use PyTorch Lightning to reorganize the flow from input to output (in training, from data to loss function to optimize the parameter). Once again, many of these procedures are common among different RL algorithms (e.g. in A2C and REINFORCE with different baselines, this means modularizing the baselines). In terms of research, we have identified several pitfalls that can be encountered when training CO solvers, such as out-of-distribution generalization, that could be alleviated by introducing several inductive biases or exploring different directions as highlited in the paragraph “Future Directions in RL4CO” in Section 4.3. Finally, we believe the first steps in deriving a new methods is to recognize the problems as we highlighted in our benchmarking sections, and importantly, having a solid software base from which to start experimenting.

---

> > ### Author Response · Authors · 2023-11-21
> > **Reply (2)**
> >
> > > Additionally, one cannot ignore the issue that, for example, in the case of TSP problems, many non-autoregressive (or non-RL) methods [1,2,3] have recently achieved better performance than RL-based, like faster inference time, lower optimal gap, and runnable on larger instances (TSP1000 and TSP10000). Therefore, as a benchmark for routing problems(TSP, CVRP, etc.), the comparison of methods is not comprehensive enough.
> >
> > We agree with you. Non-autoregressive methods as the supervised DIFUSCO [1], the RL-based DIMES [2], and new sampling techniques as iSCO [3] can scale well to thousands of nodes.
> >
> > However, we should note that non-autoregressive methods often suffer to poor generalizability to more constrained and practical problems - in many cases, these methods can only extend to NP-complete problems, which are relatively easier than NP-hard problems such as the vehicle routing problems. DIFUSCO [1] specifically restricts to NP-complete problems as TSP. In DIMES [2], it is also unclear whether it can be extended to problems with more complex constraints; in fact, the authors only mention possible extensions to other NP-complete problems. As for the sampling-based iSCO [3], quoting the authors, “The current EBM formulation relies on the penalty form of the original problem. When it becomes nontrivial to find even just a feasible solution, the generic formulation of iSCO would probably fail.”.
> >
> > For the above reasons, we believe autoregressive approaches, such as the ones in the nowadays ubiquitous large language models, are more general and can deal with a wider variety of problems, that include, in combinatorial optimization, tasks such as routing, electronic design automation, scheduling, and ultimately more complex practical problems.
> >
> > ---
> >
> > [1r] Anonymous. “Unsupervised combinatorial optimization under complex conditions: Principled objectives and incremental greedy derandomization”. Under submission at ICLR (2024).

---

> > > ### Comment · Reviewer_iFP3 · 2023-11-22
> > >
> > > Thanks for your response.
> > >
> > >  I appreciate the efforts of the authors that more CO problems and applications are contained beside the routing-based problems, which makes the benchmark more complete.
> > >
> > > Regarding the design of the benchmark, I acknowledge that auto-regressive algorithms are more general in dealing with different problems. However, currently, different problems still rely on different problem formulations, so practitioners still need to design specific procedures to satisfy different constraints.  Moreover, though non-auto-regressive methods are constrained to some NP-complete problems, the performance gain is significant, especially on large TSP/VRP problems (like TSP-1000/10000, where small-scale TSP-50 is solved well for current approaches).
> > > Since the main focus of the paper is on routing-related problems in benchmarks, I suggest the author consider both generality and benchmark completeness, and consider how to organize the benchmark to better help the community. For now, I would keep my score of 5.

---

### Official Review · Reviewer_CdMT · 2023-10-23

**Soundness:** 2 fair
**Presentation:** 3 good
**Contribution:** 3 good
**Rating:** 5
**Confidence:** 4

**Summary:**

In this paper, the authors propose a general reinforcement learning framework for solving combinatorial optimization problems. They aim to unify several existing methods into one general code library called RL4CO. The code repository is employed with SOTA software, which makes it flexible, modifiable, and extensible to the users. Based on their proposed RL4CO, they are able to conduct multiple evaluations for existing methods.

**Strengths:**

1. This paper is well-written and easy to follow.
2. The proposed unified framework is necessary for the community, and the code is open-source.

**Weaknesses:**

1. Though the authors claim that they aim to propose a unified framework, the methods considered in their paper are mainly based on AM and POMO, in other words, the auto-regressive methods. As far as I know, there are also other methods other than auto-regressive, such as Local-rewrite[1]. Instead of constructing the solution from scratch, methods like Local-rewrite aim to improve an existing solution. Are these kinds of methods able to integrate into the proposed RL4CO framework?
2. The proposed unified framework mainly focuses on TSP/VRP and the graph-based CO problems. However, there are other kinds of CO such as bin-packing, job scheduling, and mixed integer programming. I wonder if the proposed RL4CO covers these CO problems.
3. In the experiments, only three methods are considered as baselines. I think the authors should test more RL4CO methods as they claimed they propose a unified library.
4. Still in the experiments, the size of TSP/VRP is only 50. I consider this size to be relatively small.


[1] Chen, Xinyun, and Yuandong Tian. "Learning to perform local rewriting for combinatorial optimization." Advances in Neural Information Processing Systems 32 (2019).

**Questions:**

As I mentioned before, I consider the proposed unified framework to be meaningful to the community, but I think the authors may over-claim their work. They call their framework "RL4CO", but they only cover 3 RL methods and limited CO problems.

---

> ### Author Response · Authors · 2023-11-21
> **Reply (1)**
>
> Thank you for your insightful feedback. We will answer to your concerns point by point.
>
> > Though the authors claim that they aim to propose a unified framework, the methods considered in their paper are mainly based on AM and POMO, in other words, the auto-regressive methods. As far as I know, there are also other methods other than auto-regressive, such as Local-rewrite[1]. Instead of constructing the solution from scratch, methods like Local-rewrite aim to improve an existing solution. Are these kinds of methods able to integrate into the proposed RL4CO framework?
>
> You are correct. Indeed, RL algorithms for CO can be generally divided into 1) constructive and 2) improvement.
>
> Constructive methods usually generate the solution from scratch, which we focus on in the paper. They are generally faster, but the quality of solutions can be lower than that of improvement methods. In construction methods, inference techniques (such as greedy multistart, sampling and augmentation) focus on getting the best out of several generated solutions, while search methods such as EAS focus on improving the quality of generated solutions efficiently by updating the model at test time.
>
> On the other hand, improvement methods (such as the papers [1,2]) learn to improve existing solutions. Usually obtain a better solution, but are slower and may be less general (i.e. specialized to a class of problems such as routing). The reason why in RL4CO we focus first on construction methods is that they are more generally applicable to a wide variety of problems, such as routing and electronic design automation, while most learned improvement approaches are ad-hoc for a single class of problems, such as integer programming [2r] or routing problems [1r].
>
> > The proposed unified framework mainly focuses on TSP/VRP and the graph-based CO problems. However, there are other kinds of CO such as bin-packing, job scheduling, and mixed integer programming. I wonder if the proposed RL4CO covers these CO problems.
>
> We agree with your opinion. We created RL4CO with not just routing problems in mind - for instance, as shown in Appendix B, we implemented electronic design automation. Our goal is to include a wide variety of problems with the help of our community. For instance, we recently introduced in the library scheduling problems including the Single Machine Total Weighted Tardiness Problem (SMTWTP) and the Flexible Flow Shop Problem (FFSP). Moreover, RL4CO is easily extensible to other problems; for instance, a recent paper [4r] has extended our implementation to the max cover problem and facility location, which we plan to integrate in RL4CO.

---

> > ### Author Response · Authors · 2023-11-21
> > **Reply (2)**
> >
> > > Still in the experiments, the size of TSP/VRP is only 50. I consider this size to be relatively small.
> >
> > We partially agree, in the sense that most papers in the NCO area pre-train models on 100 nodes. While pre-training on several hundred nodes and even more is possible, especially thanks to advancements we have included as FlashAttention, this would result in burdensome computational costs that would be prohibitive with our limited resources - for instance, Sym-NCO took the authors around 2 weeks to train on a single problem of 100 nodes.  We believe that pre-training on small instances and generalizing to larger ones is an important topic in NCO (for instance, some recent supervised approaches can effectively scale to several thousands of nodes such as [3r] - as such, it would be an interesting avenue of research to make RL scale better as well). Another approach is adaptation, as we did with AS and EAS (up to 1000 nodes in our experiments from the pre-trained models on 50 nodes).
> >
> > > In the experiments, only three methods are considered as baselines. I think the authors should test more RL4CO methods as they claimed they propose a unified library. […] As I mentioned before, I consider the proposed unified framework to be meaningful to the community, but I think the authors may over-claim their work. They call their framework "RL4CO", but they only cover 3 RL methods and limited CO problems.
> >
> > Thanks for your comment. Ideally, we would like to include as many baselines and problems as possible, and given resource constraints, we found ourselves with an optimization problem for which we decided to prioritize user experience, flexibility and easyness of future expansion. As we are constantly developing the library in parallel with other research projects, we expect it to grow to include much more in the near future - also thanks to our active community.
> >
> > ---
> >
> > [1r] Hottung, André, and Kevin Tierney. "Neural large neighborhood search for the capacitated vehicle routing problem." ECAI 2020 (2020).
> >
> > [2r] Learning Large Neighborhood Search Policy for Integer Programming. Yaoxin Wu, Wen Song, Zhiguang Cao, Jie Zhang. NeurIPS 2021.
> >
> > [3r] Luo, Fu, et al. "Neural Combinatorial Optimization with Heavy Decoder: Toward Large Scale Generalization." NeurIPS2023 (2023).
> >
> > [4r] Anonymous. “Unsupervised combinatorial optimization under complex conditions: Principled objectives and incremental greedy derandomization”. Under submission at ICLR2024 (2023).

---

> > > ### Comment · Reviewer_CdMT · 2023-11-21
> > >
> > > Thank you for the clarifications.
> > > I agree that the RL4CO library is meaningful to the community and I appreciate the authors for developing and maintaining it.
> > > I have increased my score to 5, but I think its current version is not good enough for ICLR. I still recommend the authors add more baselines and the size of experiments, which is important for a benchmark paper in my view.
> > > Actually, benchmark paper is rare in ICLR and I have not read one before. It is hard to make a fair evaluation without guidelines. Therefore, I suggest the authors to submit their paper to the conference suitable for benchmarks

---

> > > > ### Author Response · Authors · 2023-11-21
> > > >
> > > > Thank you for acknowledging our rebuttal, insights, and the score increase!
> > > >
> > > > > Actually, benchmark paper is rare in ICLR and I have not read one before. It is hard to make a fair evaluation without guidelines. Therefore, I suggest the authors to submit their paper to the conference suitable for benchmarks.
> > > >
> > > > We completely agree with you about the lack of guidelines for libraries and benchmark papers in ICLR. Citing our comment to Reviewer bdvZ, we found several libraries and benchmark papers accepted in ICLR in the past, whose contribution relies mostly on the software aspect (for example, [1r, 2r, 3r]), even though they were not optimally suited for the research track as pointed out by their reviews.
> > > >
> > > > We hope that in the future ICLR will offer a benchmark track so that such works may be evaluated accordingly based on their contributions.
> > > >
> > > > ---
> > > >
> > > > [1r] Rogozhnikov, Alex. "Einops: Clear and reliable tensor manipulations with einstein-like notation." ICLR2022 (2022)
> > > >
> > > > [2r] Morad, Steven, et al. "POPGym: Benchmarking Partially Observable Reinforcement Learning." ICLR2023 (2023)
> > > >
> > > > [3r] Choe, Sang Keun, et al. "Betty: An automatic differentiation library for multilevel optimization." ICLR2023 (2023)

---

### Official Review · Reviewer_bdvZ · 2023-10-28

**Soundness:** 3 good
**Presentation:** 4 excellent
**Contribution:** 2 fair
**Rating:** 5
**Confidence:** 4

**Summary:**

This paper introduces RL4CO, a modular, flexible, and unified Reinforcement Learning library for solving combinatorial optimization problems. It is built upon the best software practices and libraries and implements several classical RL for CO algorithms. The introduction of this library could enable a fair and unified comparison among different methods.

**Strengths:**

1. RL4CO addresses the reproducibility problems and the fair comparison among different RL methods in NCO, which I believe is one important contribution to the research in the relevant field.
2. The library provides an easy-to-use interface to implement different RL algorithms, facilitating research in this field.
3. This paper makes a comprehensive empirical comparison of different models, problems, and search algorithms, which are impressive and helpful to future research.
4. The presentation is very good, the whole paper is well-motivated and easy to follow.

**Weaknesses:**

I appreciate the effort of the authors in introducing this easy-to-use library, but I also want to point out that the contribution of this work is not as significant as it claims.
1. This paper quite narrows the idea of RL algorithms for CO. There could be different ways to formulate the CO as an RL problem, not limited to the one outlined in Equation (1) - (3).  For example, neighborhood search first constructs an initial feasible solution and optimizes the current solution at each step [1,2], DIMES predicts a solution directly and optimizes the model with REINFORCE (one-step RL) [3].
2. Most CO problems presented in the paper are routing problems, other classical CO problems like set covering, and maximum independent set are not covered.


[1] Learning to Perform Local Rewriting for Combinatorial Optimization. Xinyun Chen, Yuandong Tian. NeurIPS 2019.

[2] Learning Large Neighborhood Search Policy for Integer Programming. Yaoxin Wu, Wen Song, Zhiguang Cao, Jie Zhang. NeurIPS 2021.

[3] DIMES: A Differentiable Meta Solver for Combinatorial Optimization. Ruizhong Qiu, Zhiqing Sun, Yiming Yang. NeurIPS 2022.

**Questions:**

I note that DIMES is listed as a supervised approach (footnote on page 1), which I think is incorrect. I'm wondering if the authors have any specific criterion for the method classification or it is just a mistake (relevant to the first point in weaknesses).

---

> ### Author Response · Authors · 2023-11-21
> **Reply**
>
> Thank you for your insightful review. We are glad you acknowledge the strengths of our paper and library, including reproducibility, usability, comprehensiveness, and presentation. We will address your concerns in the following.
>
> > This paper quite narrows the idea of RL algorithms for CO. There could be different ways to formulate the CO as an RL problem, not limited to the one outlined in Equation (1) - (3). For example, neighborhood search first constructs an initial feasible solution and optimizes the current solution at each step [1,2].
>
> You are correct. Indeed, RL algorithms for CO can be generally divided into 1) constructive and 2) improvement.
>
> Constructive methods usually generate the solution from scratch, which we focus on in the paper. They are generally faster but the quality of solutions can be lower than improvement methods. In construction methods, inference techniques (such as greedy multistart, sampling and augmentation) focus on getting the best out of several generated solutions, while search methods such as EAS focus on improving the quality of generated solutions efficiently by updating the model at test time.
>
> On the other hand, improvement methods (such as the papers [1,2]) learn to improve existing solutions. Usually obtain a better solution, but are slower and may be less general (i.e. specialized to a class of problems such as routing). The reason why in RL4CO we focus first on construction methods is that they are more generally applicable to various CO problems, such as routing and electronic design automation, while most learned improvement approaches are ad-hoc for a single class of problems, such as integer programming [1] or routing problems [2r].
>
> > DIMES predicts a solution directly and optimizes the model with REINFORCE (one-step RL) [3]  […] I note that DIMES is listed as a supervised approach (footnote on page 1), which I think is incorrect. I'm wondering if the authors have any specific criterion for the method classification or it is just a mistake (relevant to the first point in weaknesses).
>
> About the mistake: yes, you are right, this is a typo on our side; thanks for spotting it! We have removed it from supervised approaches in the revised PDF.
>
> About benchmarking DIMES: we agree with you; we would like to do this! However, we would like to remind that our method is more general regarding solvable problems, i.e., we can deal with more complex constraints. In DIMES, it unclear whether the approach can be extended to problems with more complex constraints; in fact, authors only mention possible extension to other NP-complete problems (i.e., quoting the authors, “extendable to NP-complete problems [1r]”), but not to NP-hard problems such as vehicle routing problems in general. This limits several practical applications of NCO, which involve several more complex constraints than the ones found MIS and TSP.
>
> > Most CO problems presented in the paper are routing problems, other classical CO problems like set covering, and maximum independent set are not covered.
>
> We agree with your insight, it would be useful to have such problems covered as well in the future and we believe it will not be hard thanks to our design choices. Our goal is to include a wide variety of problems with the help of our community. For instance, we recently introduced the library scheduling problems including the Single Machine Total Weighted Tardiness Problem (SMTWTP) and the Flexible Flow Shop Problem (FFSP). Moreover, RL4CO is easily extensible to other problems thanks to its flexibility; for instance, a recent paper [3r] has extended our implementation to the max cover problem and facility location, which we plan to integrate in our library.
>
> ---
>
> [1r] Richard M Karp. Reducibility among combinatorial problems. Complexity of Computer Computations, pages 85–103, 1972.
>
> [2r] Hottung, André, and Kevin Tierney. "Neural large neighborhood search for the capacitated vehicle routing problem." ECAI 2020 (2020).
>
> [3r] Anonymous. “Unsupervised combinatorial optimization under complex conditions: Principled objectives and incremental greedy derandomization”. Under submission at ICLR (2024).

---

> > ### Comment · Reviewer_bdvZ · 2023-11-22
> > **Thanks for Response**
> >
> > Thank you for your detailed response. Most of the clarifications are clear and easy to follow, but I do not fully agree with the point that "the construction methods are generally applicable to various CO problems." Currently, most constructive neural methods work on CO problems where a feasible solution can be guaranteed via a simple heuristic/post-processing. For example, in routing problems, whenever the model selects the next stop, this stop will be removed from the candidate list to ensure a valid route. However, in problems like the generalized assignment problem, there is no simple heuristic/post-processing to ensure the feasibility of each solution.
> >
> > I'm still glad to see lots of other CO problems you mentioned that could be addressed by RL4CO. My expectation for RL4CO is that it could facilitate the research on lots of new CO problems. However, the current version does not reflect this. I think the contribution of this work would be much more significant if you could include the results of those CO problems you mentioned.

---

### Official Review · Reviewer_h7Zj · 2023-11-01

**Soundness:** 3 good
**Presentation:** 3 good
**Contribution:** 3 good
**Rating:** 3
**Confidence:** 3

**Summary:**

This paper proposes a unified toolbox to evaluate and implement auto-regressive methods for combinatorial optimization. The authors demonstrate that some SOTA methods actually underperform under out-of-distribution data and that policy gradient methods seem to outperform value-based methods.

**Strengths:**

The existence of a unified toolbox for implementation and evaluation of ML methods for CO is important as this community expands. The paper also empirically validates various interesting hypotheses regarding out-of-distribution data and policy gradient methods for training when compared to value-based methods.

**Weaknesses:**

While I find the contribution to be useful and the paper to be well-written, there is no novelty. I understand that the paper is supposed to be a unified toolbox for the implementation and evaluation of AR methods for CO, but the research track of ICLR is intended for novel problems and/or solutions. In the spirit of that, I cannot recommend the paper for publication and suggest that the authors submit the work to a benchmarks or tool track.

**Questions:**

In your opinion, why do policy optimization methods seem to outperform value-based methods in neural CO?

---

> ### Author Response · Authors · 2023-11-21
> **Reply**
>
> Thank you for your review. We would like to address the raised points below.
>
> > […] there is no novelty. I understand that the paper is supposed to be a unified toolbox for the implementation and evaluation of AR methods for CO, but the research track of ICLR is intended for novel problems and/or solutions. In the spirit of that, I cannot recommend the paper for publication and suggest that the authors submit the work to a benchmarks or tool track
>
> We believe the novelty of RL4CO does not lie in the fact that we introduce new problems and solutions but in the fact that we are the first to make problems and solutions that have been proposed easily accessible in a unified manner for quick and fair comparison. Our idea is that in such a way, new researchers and practitioners will be able to develop new algorithms or solve new problems in an organized manner, thus speeding up the research in the NCO community.
>
> Further, historically, several libraries and benchmark papers have been accepted in ICLR, whose contribution relies mostly on the software aspect (for example, [1r, 2r, 3r]). While we do agree with you that the novelty in a classical “research track” sense may be limited mainly to pointing out flaws of existing methods and possible solutions, we believe the real novelty and contribution of RL4CO is to provide a tool that ultimately will empower research in this field.
>
> We hope that in the future ICLR will offer a datasets and benchmarks track so that such works may be evaluated accordingly based on their contributions.
>
> > In your opinion, why do policy optimization methods seem to outperform value-based methods in neural CO?
>
> In short: value-based methods have to learn to predict the reward for a given instance accurately, but for CO problems such as in routing, this is not an easy task, where small variations in decision-making can yield huge differences in the final reward. Moreover, the reward is sparse (i.e. we have to wait until the full solution has been decoded to know the value), which further hinders the accuracy of value-based methods.
>
> We can further clarify the theoretical advantage of using policy gradients (PG) as REINFORCE methods over Actor-Critic (AC) methods in our setting. Accurately estimating $R(x,a)$ is important to train the NCO solvers with high performance. According to the estimation methods of $R(x,a)$, we can differentiate AC and PG. The AC methods (AM-PPO, AM-Critic) typically rely on the learned critics $V_\phi(x)$ to estimate $R(x,a)$ in the spirit of temporal-difference (TD) methods. TD methods can be sample-efficient and typically exhibit lower variance on estimating $R(x,a)$compared to PG. However, $V_\phi(x)$ is known to be a biased estimator of $R(x,a)$. On the other hand, the PG method estimates $R(x,a)$ via the Monte-Carlo method, which is known as an unbiased estimator of $R(x,a)$, instead can exhibit higher variance on estimating $R(x,a)$ than AC methods. Typically, our benchmarked algorithms, such as AM, POMO, and Sym-NCO, utilize proper baselines that can reduce the negative effect of larger variance on estimating  $R(x,a)$. Furthermore, in NCO, generating training samples can be simpler than the other RL applications, as the rollout of the environment can be done simply by plugging in the constructed solution into the CO problem. Due to such reasons, in our benchmarked NCO settings, PG with a proper baseline (e.g., AM, POMO, Sym-NCO) tends to perform better than AC methods (e.g., AM-PPO, AM-Critic) while enjoying reduced variance and unbiased on estimating $R(x,a)$.
>
> ---
>
> [1r] Rogozhnikov, Alex. "Einops: Clear and reliable tensor manipulations with einstein-like notation." ICLR2022 (2022)
>
> [2r] Morad, Steven, et al. "POPGym: Benchmarking Partially Observable Reinforcement Learning." ICLR2023 (2023)
>
> [3r] Choe, Sang Keun, et al. "Betty: An automatic differentiation library for multilevel optimization." ICLR2023 (2023)

---

### Meta-Review · Area_Chair_1QSk · 2023-12-03

**Metareview:**

The paper provides a benchmark suite for running RL-based solvers over combinatorial problems. The algorithms are all based on a unified API for generating solutions autoregressively (e.g. via attention methods), and are evaluated over numerous routing problems.

Most reviewers (bdvZ, CdMT, iFP3) have stated the following core weaknesses:
* The methods in the benchmark only consist of autoregressive (AR) methods, despite the fact that non-AR methods have shown to be highly competitive or even outperform AR methods. These non-AR methods were not reflected in the benchmark.
* The benchmark is limited in its coverage of combinatorial problems and mostly deals with only routing problems.

I would have to agree, and suggest that the authors either:
* Provide a more comprehensive benchmarking suite, consisting of more diverse algorithms and problems
* Change the presentation of the paper to focus more on what is currently contributed (autoregressive methods for routing problems)
* Submit to a benchmarking based venue.

As the paper currently stands, it is insufficient for acceptance at ICLR.

**Justification For Why Not Higher Score:**

The reviewers and I all agree that the paper's story (benchmark for general RL over combinatorial optimization) does not match the current limited contributions (autoregressive methods over routing problems).

These limited contributions are not enough for acceptance.

**Justification For Why Not Lower Score:**

N/A

---

### Decision · Program_Chairs · 2024-01-16

Reject